
# Application of local attractor dimension to reduced space strongly coupled data assimilation for chaotic multiscale systems

Courtney Quinn[1], Terence J. O'Kane[1], and Vassili Kitsios[1]

[1]Decadal Climate Forecasting Project, CSIRO Oceans and Atmosphere, Hobart, TAS, AU

**Correspondence:** Courtney Quinn (courtney.quinn@csiro.au)

**Abstract.** The basis and challenge of strongly coupled data assimilation (CDA) is the accurate representation of cross-domain covariances between various coupled subsystems with disparate spatio-temporal scales, where often one or more subsystems are unobserved. In this study, we explore strong CDA using ensemble Kalman filtering methods applied to a conceptual multiscale chaotic model consisting of three coupled Lorenz attractors. We introduce the use of the local attractor dimension

(i.e. the Kaplan-Yorke dimension, $\dim_{KY}$) to determine the rank of the background covariance matrix which we construct using a variable number of weighted covariant Lyapunov vectors (CLVs). Specifically, we consider the ability to track the nonlinear trajectory of each of the subsystems with different variants of sparse observations, relying only on the cross-domain covariance to determine an accurate analysis for tracking the trajectory of the unobserved subdomain. We find that spanning the global unstable and neutral subspaces is not sufficient at times where the nonlinear dynamics and intermittent linear error growth along

a stable direction combine. At such times a subset of the local stable subspace is also needed to be represented in the ensemble. In this regard the local $\dim_{KY}$ provides an accurate estimate of the required rank. Additionally, we show that spanning the full space does not improve performance significantly relative to spanning only the subspace determined by the local dimension. Where weak coupling between subsystems leads to covariance collapse in one or more of the unobserved subsystems, we apply a novel modified Kalman gain where the background covariances are scaled by their Frobenius norm. This modified gain

increases the magnitude of the innovations and the effective dimension of the unobserved domains relative to the strength of the coupling and time-scale separation. We conclude with a discussion on the implications for higher dimensional systems.

## 1 Introduction

As the world of climate modelling has moved towards coupled Earth system models which combine ocean, atmosphere, sea-ice, and biogeochemical effects, it is essential to understand how the respective domains of disparate spatio-temporal scales

covary and influence each other. In the context of state estimation, strongly coupled data assimilation (CDA) in multiscale systems allows the quantification of cross-domain dynamics. Specifically, strong CDA uses the cross-covariances amongst all components to influence the analysis increment, meaning that unobserved subsystems are directly adjusted in the analysis step regardless of observation set and coupling strengths (Laloyaux et al., 2016; O'Kane et al., 2019). Strong CDA has shown the potential to outperform weakly coupled or uncoupled approaches in intermediate complexity atmosphere-ocean systems

(Sluka et al., 2016; Penny et al., 2019), however CDA has the additional requirements of increased ensemble sizes (Han et al.,





2013) and well observed state (prognostic) variables (Kang et al., 2011). Ensemble DA methods rely on properly sampling the variance of the model, implying very large ensemble sizes are needed for high dimensional systems. In practical implementation this is often not possible due to computational costs and limitations. It is therefore necessary to investigate and develop new methods of accurately representing error growth in multiscale systems.

One approach to reduce the requirement to adequately sample the background covariances is to perform CDA in the reduced subspace of the unstable modes, known as assimilation in the unstable subspace (AUS) (Trevisan and Uboldi, 2004; Carrassi et al., 2007; Trevisan et al., 2010; Trevisan and Palatella, 2011; Palatella et al., 2013; Bocquet and Carrassi, 2017). The concept behind AUS methods is that the analysis increment should lie in the unstable and neutral subspaces of the model, therefore retaining the spatial structure of dominant instabilities (Trevisan and Uboldi, 2004). Many implementations of AUS involve

variational data assimilation methods, namely finding the model trajectory which best fits observations through solving an optimization cost function. On the other hand, the widely used ensemble Kalman filtering methods have been shown to best capture the unstable error growth in nonlinear systems (Evensen, 1997). For this reason, we focus on the ability to accurately represent the unstable and neutral subspace within the ensemble Kalman filtering framework.

    The main motivation for this study comes from the conjecture that when applying ensemble Kalman filtering methods to

high dimensional nonlinear systems, the true time-dependent error covariance matrix collapses onto a subspace of the model domain which contains unstable, neutral, and sometimes weakly stable modes. While previous results prove the collapse of the error covariance matrix onto the unstable and neutral subspace for linear systems (Gurumoorthy et al., 2017; Bocquet et al., 2017), nonlinear systems have the additional complication that the unstable subspace is a function of the underlying trajectory and not globally defined (Bocquet et al., 2017). As nonlinearity increases, short-term dynamics can cause some stable modes

of the linearized system to experience significant growth. These additional modes are therefore important when considering local error growth in ensemble DA methods (Ng et al., 2011).

    Such transient error growth has previously been explored in ocean-atmosphere models of varying complexity. One way of quantifying local error growth is through finite-time Lyapunov exponents (FTLEs), *i.e.* rates of expansion and contraction over finite intervals of time. Nese and Dutton (1993) utilised the largest (leading) FTLE to quantify predictability times along

different parts of the attractor of a low-order ocean-atmosphere model. The statistical properties of FTLEs have been studied more recently in a range of atmosphere and ocean models with varying complexity, including low-order models (Vannitsem, 2017) and intermediate-complexity models (Vannitsem, 2017; De Cruz et al., 2018; Kwasniok, 2019). While FTLEs provide local rates of error growth, one can also consider directions of local error growth. Early work in this area considered finite-time normal modes, which are generalised as the eigenvectors of tangent linear propagators over a given time interval, and studied

their relation to blocking events in the atmosphere (Frederiksen, 1997, 2000; Wei and Frederiksen, 2005). More recent studies have focused on covariant Lyapunov vectors (CLVs) which give directions of asymptotic growth and decay in the tangent linear space. While these vectors have an average growth rate corresponding to the asymptotic Lyapunov exponents, their finite-time behaviour may differ. This finite-time behaviour of CLVs has been analysed across a range of quasi-geostrophic atmosphere (Schubert and Lucarini, 2015, 2016; Gritsun and Lucarini, 2017) and coupled atmosphere-ocean (Vannitsem and Lucarini,

2016) models.





In this study we utilise FTLEs and CLVs within the ensemble Kalman filtering framework applied to a low-dimensional chaotic model with spatio-temporal scale separation. The model was designed to represent the interaction between the ocean, extratropical atmosphere, and an ocean-atmosphere interface (referred to as the tropical atmosphere). The idea is that the ocean and extratropical atmosphere are only implicitly coupled through the interface, with the interface being strongly coupled to the

ocean and weakly coupled to the extratropical atmosphere. We consider the performance of strong CDA on this paradigm model with different subsets of observations. We introduce the use of local CLVs to form the forecast error covariance matrix. The idea of AUS is incorporated through the use of a time-varying subspace defined by the local attractor dimension. The dimension is calculated through FTLEs and the error covariance matrix is constructed via a projection of the ensemble members onto a corresponding subset of the CLVs at each analysis step. We compare full rank ensemble transform and square-root filters to

"phase space" variants whose background covariances are defined in terms of either the local or global attractor dimension. Another variant considered includes only the unstable, neutral, and weakest stable CLVs. We consider benchmark calculations comparing to the recent work of Yoshida and Kalnay (2018) and then a comprehensive set of experiments where the various domains are partially or even completely unobserved. We also examine the role of correlated versus random observational errors.

The paper is organized as follows. Section 2 introduces the paradigm model and discusses the dynamical properties of a control simulation. Section 3 describes the method for calculating the CLVs and discusses the possibility of CLV alignment. Section 4 summarizes the Kalman filtering method and introduces the modification to the calculation of the error covariance matrix. The results of the DA experiments are presented in Section 5, along with a novel scheme for adaptive Kalman gain inflation. The implications of the results of this study and future endeavours are discussed in Section 6.

## 20  2  Coupled Lorenz model for global circulation

### 2.1  Peña and Kalnay model

This section describes a low-dimensional chaotic model representing a coupled ocean and atmosphere. It is a system of three coupled Lorenz 63 models introduced by Peña and Kalnay (2004) to study unstable modes with a time scale separation. This model has previously been described in modified form by Norwood et al. (2013) who used it to investigate the instability

properties of related dynamical vectors i.e. bred vectors (BVs), singular vectors (SVs) and CLVs, and by Yoshida and Kalnay





(2018) and O'Kane et al. (2019) in the context of strongly coupled data assimilation. The model is given as follows:

$$\dot{x}_e = \sigma(y_e - x_e) - c_e(Sx_t + k_1), \tag{1a}$$

$$\dot{y}_e = \rho x_e - y_e - x_e z_e + c_e(Sy_t + k_1), \tag{1b}$$

$$\dot{z}_e = x_e y_e - \beta z_e, \tag{1c}$$

$$\dot{x}_t = \sigma(y_t - x_t) - c(SX + k_2) - c_e(Sx_e + k_1), \tag{1d}$$

$$\dot{y}_t = \rho x_t - y_t - x_t z_t + c(SY + k_2) + c_e(Sy_e + k_1), \tag{1e}$$

$$\dot{z}_t = x_t y_t - \beta z_t + c_z Z, \tag{1f}$$

$$\dot{X} = \tau\sigma(Y - X) - c(x_t + k_2), \tag{1g}$$

$$\dot{Y} = \tau\rho X - \tau Y - \tau SXZ + c(y_t + k_2), \tag{1h}$$

$$\dot{Z} = \tau SXY - \tau\beta Z - c_z z_t. \tag{1i}$$

The model is proposed as representing the fast extratropical atmosphere (1a-c), fast tropical atmosphere (1d-f), and slow tropical ocean (1g-i). The standard Lorenz parameter values of $\sigma = 10$, $\rho = 28$, and $\beta = 8/3$ are used. The coupling coefficients

are $c_e = 0.08$ and $c = c_z = 1$, representing a weak coupling of extratropical to tropical atmosphere and a strong coupling of tropical atmosphere and ocean. The scaling parameters are set as $\tau = 0.1$ and $S = 1$, giving an explicit timescale separation (note there is still difference in the spatial scales through the dynamics). The parameters $k_1 = 10$ and $k_2 = -11$ are chosen as uncentering parameters.

More generally, these choices lead to a tropical subsystem that is dominated by changes in the ocean subsystem, but has

an extratropical atmosphere, representative of weather noise, whose behaviour is similar to the original Lorenz model in that the system exhibits chaotic behaviour with two distinct regimes observed in the $x_e$ and $y_e$ coordinates of the extratropical atmosphere. The ocean exhibits significant deviations from its normal trajectory about every 2-8 years where three years corresponds to approximately 10 time units. Figure 1 shows typical trajectories of the $x_e$, $x_t$, and $X$ components of the extratropical, tropical, and ocean subsystems respectively. Figure 2 shows the approximate phase space structure of each of the

respective subsystem's attractor.

## 2.2 Lyapunov exponents and dimension

We are interested in analysing both the local and global dynamics of system (1). We start by considering the asymptotic behaviour of trajectories, which can be understood through the Lyapunov exponents. Chaotic systems are characterized by one or more positive Lyapunov exponents (Benettin et al., 1976; Sano and Sawada, 1985), and the underlying attractor dimension

can be related to the values of the Lyapunov exponents (Young, 1982; Eckmann and Ruelle, 1985).





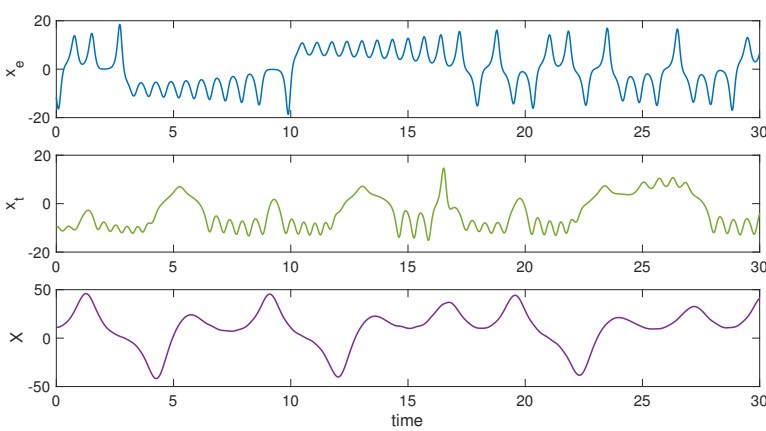

**Figure 1.** Example trajectories of coupled Lorenz model (1) for $x_e$ (top), $x_t$ (middle), and $X$ (bottom).

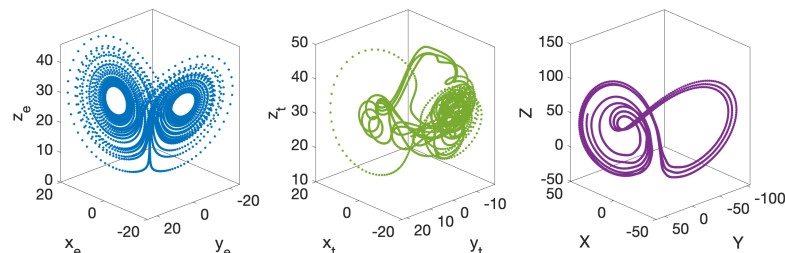

**Figure 2.** Trajectories along attractors of the extratropical (left), tropical (middle) and ocean (right) subsystems of (1).

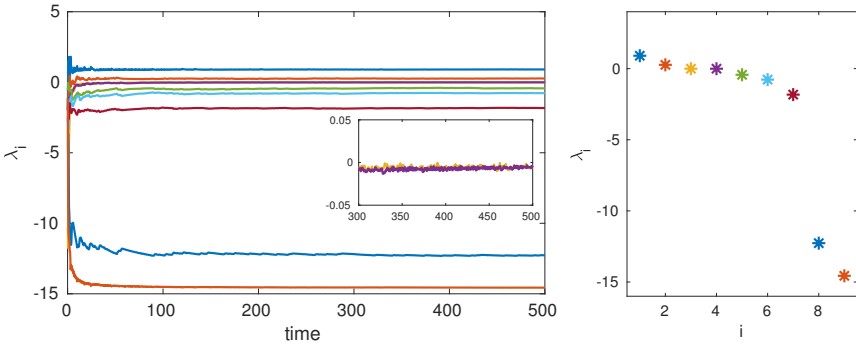

**Figure 3.** (Left) Convergence of Lyapunov exponents of (1) with inset of zoom around zero to show the two neutral exponents. (Right) Asymptotic values of Lyapunov exponents of (1).





We compute the Lyapunov exponents using a QR decomposition method (see *e.g.* Dieci et al. (1997)). For the computation we run the model for 1000 time units to ensure convergence onto the attractor. We use a time step of 0.01 and compute the Lyapunov exponents from the last 500 time units, with an orthonormalization time step of 0.25 for the QR method. The convergence of the Lyapunov exponents is shown in Figure 3. We observe that for these parameter values there are two unstable

and two approximately neutral Lyapunov exponents. We list the values of all nine computed asymptotic exponents in Table 1.

| $\lambda_1$ | $\lambda_2$ | $\lambda_3$ | $\lambda_4$ | $\lambda_5$ | $\lambda_6$ | $\lambda_7$ | $\lambda_8$ | $\lambda_9$ |
|---|---|---|---|---|---|---|---|---|
| 0.9071 | 0.2670 | -0.0056 | -0.0060 | -0.4326 | -0.7706 | -1.8263 | -12.2691 | -14.5640 |

**Table 1.** Asymptotic Lyapunov exponents of (1) computed over 500 time units using a QR decomposition method, time step of 0.01, and orthogonalization step of 0.25.

Given the approximated asymptotic Lyapunov exponents in Table 1, we can compute the global attractor dimension i.e. the number of active degrees of freedom. Notice that there is a large spectral gap between the seventh and eighth Lyapunov exponents. This gives evidence that the effective dimension of the attractor will be less than 8. Throughout this study we will use the Kaplan-Yorke dimension (Kaplan and Yorke, 1979; Frederickson et al., 1983) to calculate the upper bound on attractor

dimension. It is defined as follows:

$$\dim_{KY} := j + \frac{\sum_{i=1}^{j} \lambda_i}{|\lambda_{j+1}|}, \tag{2}$$

where $j$ is the largest integer such that

$$\sum_{i=1}^{j} \lambda_i \geq 0$$

and

$$\sum_{i=1}^{j+1} \lambda_i < 0.$$

In addition we calculate the Kolmogorov–Sinai entropy $\text{ent}_{KS}$ as a measure of the diversity of the trajectories generated by the dynamical system and determined through the Pesin formula

$$\text{ent}_{KS} = \sum_{\lambda > 0} \lambda_i, \tag{3}$$

which provides an upper bound to $\text{ent}_{KS}$ (Eckmann and Ruelle, 1985).

With the values in Table 1 we obtain a value of 5.9473 for the global attractor dimension of the 9-component system. As previously mentioned, asymptotically stable modes can experience transient periods of linear unstable growth. We therefore define local dimension through the computation of (2) using finite-time Lyapunov exponents (FTLEs). The computation of FTLEs is similar to that of the asymptotic Lyapunov exponents, with the difference being a finite window of time over which

the exponents are computed. The FTLEs then serve as a time-dependent measure of the local unstable, neutral, and stable growth rates of the evolving system (Abarbanel et al., 1991; Eckhardt and Yao, 1993; Yoden and Nomura, 1993). The temporal





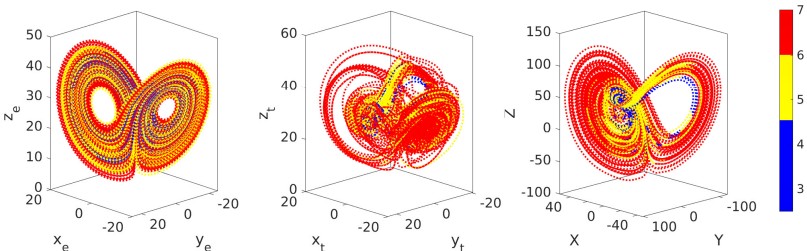

**Figure 4.** Local Kaplan-Yorke dimension plotted along trajectory in phase space for the extratropical atmosphere (left), tropical atmosphere (middle), and ocean (right) subsystems.

variability of FTLEs is highly dependent on the window size, $\tau$, over which they are computed. As $\tau \to \infty$ the variability reduces and the FTLEs approach their corresponding asymptotic values (Yoden and Nomura, 1993). We compute the FTLEs for a window size $\tau = 4$ and the corresponding time-varying Kaplan-Yorke dimension. Figure 4 shows the local dimension plotted along the attractors of each of the subsystems. We see the local dimension is lowest when the model is in the interior
region of the ocean subsystem attractor. In contrast, the extratropical atmosphere subsystem attractor displays periods of low dimension largely uniformly confined to the center of each lobe of the attractor. The tropical atmosphere also shows most of the measures of low dimension confined to the interior of the attractor, reflecting the strong ocean coupling.

## 3   Covariant Lyapunov vectors

The existence of Lyapunov vectors for a large class of dynamical systems was proven by Oseledets (1968). The Multiplicative
Ergodic Theorem states that there exists a relation between Lyapunov exponents, $\lambda_i$, and a (non-unique) set of vectors $\phi$ such that

$$\lambda_i = \lim_{\tau \to \infty} \frac{1}{\tau} \log \|\mathcal{A}(x(t), \tau)\phi\| \qquad \text{iff} \qquad \phi \in \Phi_i(x(t)) \setminus \Phi_{i+1}(x(t)) \qquad (4)$$

Here, $A(x(t), \tau)$ forms the cocycle of the tangent dynamics for a given dynamical system. The subspaces ($\Phi_i$) on which the growth rates ($\lambda_i$) occur are covariant with the tangent dynamics and invariant under time reversal (Ginelli et al., 2007; Froyland
et al., 2013). The covariant Lyapunov vectors (CLVs) are then defined as the set of directions at each point in phase space that satisfy (4) both backwards and forwards in time (Ginelli et al., 2007; Ng et al., 2011). In the last few decades there have been significant advances in the ability to numerically approximate such vectors for chaotic dynamical systems (Ginelli et al., 2007; Wolfe and Samelson, 2007; Froyland et al., 2013). In this work we will employ a numerical algorithm introduced by Froyland et al. (2013). We summarize this algorithm below.
It should be mentioned that the push forward step need not be equal to $N$; $M \neq N$ for $\mathcal{A}(x^{i-N}, M)$. However, for all calculations in this study we consider only the case $M = N$. The trajectory of the system is discretized with time step $\Delta t$ such that $x^i = x(t_i)$ and $x^{i+m} = x(t_i + m\Delta t)$.





---

**Algorithm 1** Summary of numerical algorithm for calculating CLVs introduced by Froyland et al. (2013)

---

1. Construct matrix cocycle $\mathcal{A}(x^{i+m}, 0)$ for every $i \in [-N, ..., N]$.

2. Compute the eigenvectors $\boldsymbol{e}_j^{i-N}$ of $\mathcal{A}(x^{i-N}, N)^* \mathcal{A}(x^{i-N}, N)$ [the right singular vectors of $\mathcal{A}(x^{i-N}, N)$], where $\mathcal{A}(x^{i-N}, N) = \mathcal{A}(x^i, 0) \cdot ... \cdot \mathcal{A}(x^{i-N}, 0)$. Note that $\mathcal{A}(x^{i-N}, N)^*$ is the adjoint of $\mathcal{A}(x^{i-N}, N)$.

3. Push forward by multiplication of matrix cocycle, $\boldsymbol{\phi}_j^i = \mathcal{A}(x^{i-N}, N)\boldsymbol{e}_j^{i-N}$.

4. For each $j$, reorthogonalize $\boldsymbol{\phi}_j^i$ with subspace spanned by eigenvectors $\boldsymbol{e}_k^{i-n}$ for $k = 1, ..., j-1$ of $\mathcal{A}(x^{i-n}, N)^* \mathcal{A}(x^{i-n}, N)$ every $n$ time steps.

5. The vector $\boldsymbol{\phi}_j^i$ is then an approximation of the $j-th$ largest CLV at time $t = t_i$.

---

We examine the effectiveness of this algorithm on the Peña and Kalnay (2004) model introduced in Section 2. By definition, CLVs describe the directions in phase space corresponding to different growth rates. Two or more CLVs can align during a transition to a different regime in phase space. We calculate the alignment through

$$\theta_{i,j} = \frac{|\boldsymbol{\phi}_i \cdot \boldsymbol{\phi}_j|}{||\boldsymbol{\phi}_i|| \cdot ||\boldsymbol{\phi}_j||}. \tag{5}$$

Here, $\theta_{i,j} = |\cos(\Theta_{i,j})|$ where $\Theta_{i,j}$ is the angle between the $i$-th and $j$-th CLV. Larger values of $\theta_{i,j}$ imply alignment of the two CLVs. Figure 5 shows the alignment of the unstable CLVs ($\theta_{1,2}$) and the neutral CLVs ($\theta_{3,4}$) plotted against the $X$ component of system (1), along with the FTLEs and local time-varying Kaplan-Yorke dimension. As the changes in local dimension reflect the structure of the ocean subsystem (Figure 4), we expect alignment of the CLVs during a transition in this subsystem. The CLVs analysed in Figure 5 are calculated from the first 35 time units of the previous model run with a time step $\Delta t = 0.01$.

We start the calculation at $t = 5$ to allow for initialization and a window size of $\tau = 4$. The parameters for the algorithm are set as $N = 400$ and $n = 25$. It can be seen that the algorithm detects near alignment of either the unstable or neutral CLVs during the transitions in the ocean subsystem. The transitions here are between the inner part of the attractor with smaller oscillation amplitudes and the outer, large amplitude excursions. In general, the alignment is most prominent in the outer, large amplitude part of the attractor. This follows the changes in local dimension, shown in the lower panel of Figure 5. Higher local dimension

tends to accompany alignment of the unstable and neutral CLVs.

In the following sections, we will utilise CLVs within the data assimilation framework of ensemble forecasting. The CLVs will be used to construct the forecast error covariance matrix, which informs the increment used on ensemble members to bring them closer to observations. Using CLVs in this context suggests a more accurate method of forming the forecast error covariance matrix when the true covariance is undersampled due to insufficient number of ensemble members.

## 20  4  Data assimilation with the Kalman filter

We now summarize the Kalman filter equations. For detailed derivations we refer the interested reader to the reviews by Evensen (2003), Houtekamer and Zhang (2016) or Carrassi et al. (2018). Here we follow the notation of Carrassi et al. (2018).

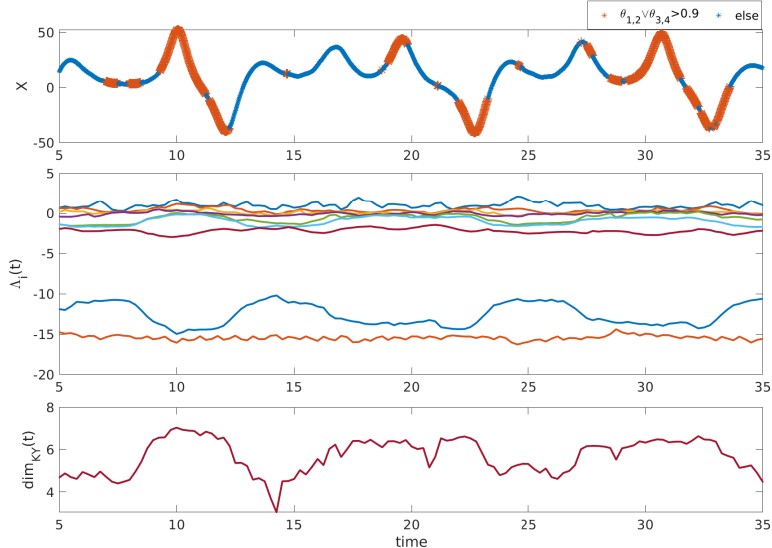

**Figure 5.** Local dynamical properties of a segment of an example model run. (Top) Alignment of CLVs associated with the unstable and neutral subspaces plotted along the x-coordinate of the ocean subsystem. Large orange stars indicate high alignment of unstable or neutral CLVs ($\theta_{1,2} > 0.9$ or $\theta_{3,4} > 0.9$). (Middle) Time-varying FTLEs $\Lambda_i(t)$ computed over window $\tau = 4$. (Bottom) Local Kaplan-Yorke dimension calculated from FTLEs.

Consider a deterministic or stochastic model defined by

$$\boldsymbol{x}_k = \mathcal{M}_{k:k-1}(\boldsymbol{x}_{k-1}, \boldsymbol{p}) + \eta_k, \tag{6}$$

where $\boldsymbol{x}_k \in \mathbb{R}^N$ is the model state at time $t = t_k$, $\boldsymbol{p} \in \mathbb{R}^p$ are the model parameters, $\mathcal{M}_{k:k-1} : \mathbb{R}^N \to \mathbb{R}^N$ is a function taking the model from time $t_{k-1}$ to $t_k$, and $\eta_k$ is the model error at time $t_k$ (for deterministic systems let $\eta_k = 0$). Suppose there exists
5    a time-dependent set of observations $\boldsymbol{y} \in \mathbb{R}^d$ which can be expressed as a function of the model state through

$$\boldsymbol{y}_k = \mathcal{H}_k(\boldsymbol{x}_k) + \epsilon_k. \tag{7}$$

The observation operator $\mathcal{H}_k : \mathbb{R}^N \to \mathbb{R}^d$ can be linear or nonlinear, and $\epsilon_k$ is the observational error.

In the Kalman filter method, equations (6) and (7) are assumed to be linear, resulting in evolution and observation matrices $\mathbf{M}_{k:k-1}$ and $\mathbf{H}_k$ respectively. The model and observation errors, $\eta_k$ and $\epsilon_k$, are taken to be uncorrelated and from a Gaussian
10    distribution with covariance matrices $\mathbf{Q}_k \in \mathbb{R}^{N \times N}$ and $\mathbf{R}_k \in \mathbb{R}^{d \times d}$ respectively. There are two basic steps to the Kalman filter method: forecast and analysis.

– **Forecast equations**

$$\boldsymbol{x}_k^f = \mathbf{M}_{k:k-1} \boldsymbol{x}_{k-1}^a \tag{8a}$$

$$\mathbf{P}_k^f = \mathbf{M}_{k:k-1} \mathbf{P}_{k-1}^a \mathbf{M}_{k:k-1}^{\mathrm{T}} + \mathbf{Q}_k \tag{8b}$$




- **Analysis equations**

$$\mathbf{K}_k = \mathbf{P}_k^f \mathbf{H}_k^{\mathrm{T}} [\mathbf{H}_k \mathbf{P}_k^f \mathbf{H}_k^{\mathrm{T}} + \mathbf{R}_k]^{-1} \tag{9a}$$

$$\boldsymbol{x}_k^a = \boldsymbol{x}_k^f + \mathbf{K}_k (\boldsymbol{y}_k - \mathbf{H}\boldsymbol{x}_k^f) \tag{9b}$$

$$\mathbf{P}_k^a = (\mathbf{I}_k - \mathbf{K}_k \mathbf{H}_k) \mathbf{P}_k^f \tag{9c}$$

There is difficulty in implementing equations (8-9) for realistic systems which have high dimension and are nonlinear (as is the case in weather and climate forecasting). Solving such equations are computationally expensive and produce errors from the assumption of linearity (Evensen, 1994). Within the Kalman filter class, various ensemble filter variants have been applied to tracking trajectories in nonlinear systems. The most popular are the deterministic filters (Tippett et al., 2003; Sakov and Oke, 2008; Sakov et al., 2012).

## 4.1  Ensemble Kalman filtering

Ensemble Kalman filtering methods use Monte Carlo sampling to form the approximate error statistics of a model. An ensemble of model states $\boldsymbol{x}^f \in \mathbb{R}^N$ with a finite number of ensemble members $m$ produces an approximation to the true error covariance matrix as follows. The ensemble forecast anomaly matrix $\mathbf{X}^f \in \mathbb{R}^{N \times m}$ is constructed with respect to the ensemble mean $\overline{\boldsymbol{x}}^f \in \mathbb{R}^N$:

$$\overline{\boldsymbol{x}}^f = \frac{1}{m} \sum_{n=1}^{m} \boldsymbol{x}_n^f, \tag{10a}$$

$$\mathbf{X}^f \equiv \frac{1}{\sqrt{m-1}} \Big[ \boldsymbol{x}_1^f - \overline{\boldsymbol{x}}^f, \ldots, \boldsymbol{x}_m^f - \overline{\boldsymbol{x}}^f \Big]. \tag{10b}$$

Note that we have dropped the time subscript $k$, the subscripts used here refer to individual ensemble members. The forecast error covariance matrices $\mathbf{P}^f$ are then constructed through

$$\mathbf{P}^f = (\mathbf{X}^f)(\mathbf{X}^f)^{\mathrm{T}}. \tag{11}$$

To preserve the variance of the ensemble through the analysis step, square-root (deterministic) schemes for ensemble Kalman filtering are often used. One such scheme is the ensemble transform Kalman filter (ETKF) developed by Bishop et al. (2001). In such schemes, the observations do not need to be perturbed to preserve the analysis covariance in equation (9c). The main idea is that a transform matrix $\mathbf{T}$ can be used to adjust the ensemble analysis anomalies matrix,

$$\mathbf{X}^a \equiv \frac{1}{\sqrt{m-1}} \Big[ \boldsymbol{x}_1^a - \overline{\boldsymbol{x}}^a, \ldots, \boldsymbol{x}_m^a - \overline{\boldsymbol{x}}^a \Big] = \mathbf{X}^f \mathbf{T}, \tag{12}$$

which ultimately forms the analysis error covariance matrix,

$$\mathbf{P}^a = (\mathbf{X}^a)(\mathbf{X}^a)^{\mathrm{T}}. \tag{13}$$

This transform matrix $\mathbf{T}$ is recovered through calculating the ensemble perturbations in normalized observation space,

$$\mathbf{E} = (\mathbf{R}^{-1/2} \mathbf{H} \mathbf{X}^f)^{\mathrm{T}} (\mathbf{R}^{-1/2} \mathbf{H} \mathbf{X}^f), \tag{14}$$





The transform matrix $\mathbf{T}$ is then defined as

$$\mathbf{T} = (\mathbf{I} + \mathbf{E}^{\mathrm{T}}\mathbf{E})^{-1/2}, \tag{15}$$

where $\mathbf{I}$ is the $m \times m$ identity matrix. See Bishop et al. (2001) for the full derivation. This leads to the update of the ensemble mean and the individual ensemble members to their analyzed state through the equations:

$$\overline{\boldsymbol{x}}^a = \overline{\boldsymbol{x}}^f + \mathbf{K}(\boldsymbol{y} - \mathbf{H}\overline{\boldsymbol{x}}^f), \tag{16a}$$

$$\boldsymbol{x}_n^a = \overline{\boldsymbol{x}}^a + (\sqrt{m-1})[\mathbf{X}^f\mathbf{T}]_{*,n}. \tag{16b}$$

The Kalman gain $\mathbf{K}$ is defined through equation (9a), and the notation $[]_{*,n}$ denotes taking the $n$-th column of the matrix.

Another deterministic scheme for ensemble Kalman filtering which uses a left-multiplied transform matrix was shown by Tippett et al. (2003) to be equivalent to ETKF:

$$\mathbf{X}^a = \mathbf{T}\mathbf{X}^f, \tag{17a}$$

$$\mathbf{T} = (\mathbf{I} - \mathbf{K}\mathbf{H})^{1/2}. \tag{17b}$$

We will refer to this left-multiplied transform filter as the ensemble square-root filter (ESRF). The ensemble mean is updated through (16a) and the individual ensemble members are then updated through:

$$\boldsymbol{x}_n^a = \overline{\boldsymbol{x}}^a + (\sqrt{m-1})[\mathbf{T}\mathbf{X}^f]_{*,n}. \tag{18}$$

When using ensemble Kalman filtering methods like the ones introduced here, sampling errors can often occur. For non-linear models in particular, there is a systematic underestimation of analysis error covariances which eventually leads to filter divergence (Anderson and Anderson, 1999). This is commonly avoided through the use of inflation. In other words, after each analysis step the ensemble anomalies are inflated through

$$\boldsymbol{x}_n^a = \overline{\boldsymbol{x}}^a + \lambda(\boldsymbol{x}_n^a - \overline{\boldsymbol{x}}^a), \tag{19}$$

where $(\lambda - 1)/100$ is the percentage inflation.

### 4.2 Ensemble filtering in reduced subspace

Here we define the error covariance matrix $\mathbf{P}^f$ based on the directions of growth and decay of model errors associated with different timescales at the given analysis time. Specifically, we construct $\mathbf{P}^f$ using the local covariant Lyapunov vectors (CLVs) computed at each data assimilation time step where the number of CLVs is determined by the local attractor dimension $\dim_{KY}$. This allows for the inclusion of unstable, neutral and stable directions dependent on the local dynamics of the system. This differs to past approaches where the subspace was determined in terms of the long time averaged (invariant) unstable and neutral CLVs (Trevisan and Uboldi, 2004; Carrassi et al., 2008; Trevisan and Palatella, 2011).





To determine the number of CLVs required to form the basis for $\mathbf{P}^f$, we use the time dependent or local $\dim_{KY}$ rounded up to an integer value. To determine how to weight the individual CLVs, we deconstruct the ensemble anomalies matrix defined in (10b), $\mathbf{X}^f$, into

$$\mathbf{X}^f = \mathbf{\Phi W}, \tag{20}$$

where $\mathbf{\Phi}$ is a matrix with columns equal to the CLVs ($\phi_i$) and $\mathbf{W}$ is a matrix of weights. The columns of $\mathbf{\Phi}$ are ordered according to the corresponding FTLEs in descending order (the first being the direction corresponding to fastest unstable growth). In this formulation $\mathbf{\Phi}$ need not be square, i.e. the CLVs used don't need to span the entire space. We compute the CLVs at the assimilation step using Algorithm 1. Equation (20) can then be solved for $\mathbf{W}$ in a least squares sense through

$$\mathbf{W} = (\mathbf{\Phi}^\mathrm{T} \mathbf{\Phi})^{-1} \mathbf{\Phi}^\mathrm{T} \mathbf{X}^f. \tag{21}$$

The weights in $\mathbf{W}$ combined with the directions in $\mathbf{\Phi}$ now define an object with dimension equal to the chosen number of CLVs whose covariance matrix is defined by

$$\mathbf{P}^f = \mathbf{\Phi W W}^\mathrm{T} \mathbf{\Phi}^\mathrm{T}. \tag{22}$$

We can then use the formulation of $\mathbf{P}^f$ above in conjunction with the ensemble schemes of Section 4.1. We use the modified forecast covariance matrix (22) in the calculation of the Kalman gain (9a) which then also alters any subsequent calculations.

## 5 Results

We perform a collection of data assimilation experiments for system (1) using a control run as observations (computed using a Runga-Kutta 4th order scheme with $\Delta t = 0.01$). Here we emphasize that we are interested in exploring the dynamical attributes of data assimilation across multiple timescales. In all cases we are using standard strong CDA, meaning that the cross-covariances are used amongst all components regardless of observation set (Laloyaux et al., 2016; O'Kane et al., 2019).

This allows for the analysis increment of any unobserved subsystems to be influenced by the observations, even in cases of weak coupling between the subsystems.

The initialisation settings for the DA experiments are as follows, unless otherwise stated. We use the settings from Yoshida and Kalnay (2018): analysis window 0.08, inflation factor 1%, and 10 ensemble members. We run the model for 75000 time steps (9375 analysis windows) and use 50000 time steps (6250 analysis windows) for calculating analysis error statistics. The

model is let spin-up for 400 time steps before starting the assimilation cycles as we are using a window of $\tau = 4$ for the calculation of the FTLEs and CLVs. We note that for the CLV method, the system must be sufficiently tracking the control to accurately calculate the initial CLVs. For this reason we start the assimilation before there is significant ensemble divergence. The ensemble members are initialised as perturbations from the control initial condition, taken from a uniform distribution defined on $[-0.025, 0.025]$.



## 5.1 Constructing the observations

In the Kalman filtering method introduced in Section 4, there is an underlying assumption that the observations have some error with variance $\mathbf{R}$. This error variance is typically unknown and chosen *a priori*. If we consider the observations in a statistical sense, we can deconstruct them at each assimilation step into a mean field and perturbation value: $\boldsymbol{y} = \overline{\boldsymbol{y}} + \hat{\boldsymbol{y}}$. In such a formulation, $\overline{\boldsymbol{y}}$ would be the truth at a given point in time and the observation error variance would be the average of the variance of the perturbations, $\mathbf{R} = \overline{\hat{\boldsymbol{y}}\hat{\boldsymbol{y}}^{\mathrm{T}}}$. To emulate this in deterministic models where the truth $\overline{\boldsymbol{y}}$ is known (*i.e.* from a control run), it is common practice to construct the observations by adding to the truth a random value $\hat{\boldsymbol{y}}$ taken from a normal distribution with variance given by the diagonals of $\mathbf{R}$. However, this produces uncorrelated observation errors which have the same variance at any given point in phase space. We argue here that in reality, the true variance of the observation error can be spatially dependent and errors are often correlated in time. We therefore also consider the case where there is error in the observations but it is consistent with the underlying nonlinear dynamics by constructing a trajectory that 'shadows' the truth. Both types of observational errors, with the additional case of perfectly observing a subset of variables, are explored.

The main differences in our subsequent experiments are in the subset of observations used and their corresponding observational errors. We first present a benchmark test on the CLV method which is identical to an experiment presented in Yoshida and Kalnay (2018) where the $y$-component of each subsystem is observed. The perturbations to the control run are from a normal distribution with error variance $\mathbf{R} = \mathrm{diag}([1, 1, 25])$. This case is referred to as *benchmark observations*. We then consider two experiments where the observations are less sparse within the subsystems, however one subsystem is completely unobserved: *atmosphere observations* $(y_e, z_e, y_t, z_t)$ and *ENSO observations* $(y_t, z_t, Y, Z)$. For the atmosphere observations case we sample the perturbations from error variance $\mathbf{R} = \mathbf{I}_4$, while for the ENSO observations we experiment with different error variances. We then consider correlated observation errors through *shadowed observations*. For these experiments the observation error variances $\mathbf{R}$ are set to the standard values from Yoshida and Kalnay (2018), but the actual perturbations are constructed through a model trajectory which is initialized close to the control run and forced by a relaxation term back to the control. The observation errors then also reflect the local nonlinear growth. We repeat the benchmark, atmosphere, and ENSO observation cases with this type of observation error. Finally, we reduce the observation space to only the extratropical subsystem $(x_e, y_e, z_e)$. This extends upon the work in O'Kane et al. (2019) where the authors considered only ocean observations $(X, Y, Z)$. Due to the difficulty of constraining a system through only fast, weakly coupled dynamics, we decrease the analysis window and do not perturb the control run at all when taking the observations.

## 5.2 Benchmark observations

The first DA experiment we consider is a benchmark case with observations $(y_e, y_t, Y)$. We reproduce the results in Yoshida and Kalnay (2018) using the CLV method introduced in section 4.2. We first perform a DA experiment using a full rank covariance matrix (equivalent to the ETKF method introduced in section 4.1) and then compare to using reduced subspace methods. The first reduced subspace method uses a fixed number of CLVs defined by spanning the asymptotic unstable and neutral subspace plus the first stable mode as in Ng et al. (2011). The second reduced subspace method also uses a fixed number of CLVs, except





the number is defined by the asymptotic Kaplan-Yorke dimension as suggested in Carrassi et al. (2008) (note in that study the authors discuss the number of ensemble members which is equivalent to rank of covariance matrix). These fixed numbers are 5 and 6 CLVs, respectively. Finally we analyse our novel reduced subspace method which uses a variable number of CLVs based on the local Kaplan-Yorke dimension.

The error statistics of all the experiments are listed in Table 2. The analysis RMSE is calculated for each subsystem individually at every analysis window and then averaged over the windows, in line with the error statistics produced in Yoshida and Kalnay (2018). We also calculate the average RMSE of the full system. The RMSE is defined as

$$\text{RMSE} = \sqrt{\frac{1}{N}(\overline{\boldsymbol{x}}^a - \boldsymbol{x})^{\text{T}}(\overline{\boldsymbol{x}}^a - \boldsymbol{x})},\tag{23}$$

where $N$ is the number of states (either 3 or 9) and $\boldsymbol{x}$ is the truth (control run in our case). Additional metrics calculated for
each observed state variable include:

$$\text{spread} = \sqrt{\sum_{n=1}^{m}(\boldsymbol{x}_n^f - \overline{\boldsymbol{x}}^f)^2}, \qquad\qquad \text{average increment} = \frac{1}{m}\sum_{n=1}^{m}(\boldsymbol{x}_n^a - \boldsymbol{x}_n^f),$$

$$\text{MAD} = \frac{1}{m}\sum_{n=1}^{m}|\boldsymbol{y} - \mathbf{H}\boldsymbol{x}_n^f|, \qquad\qquad \text{bias} = \frac{1}{m}\sum_{n=1}^{m}(\boldsymbol{y} - \mathbf{H}\boldsymbol{x}_n^f).$$

| Method | Observations [error variance] | $\langle\text{RMSE}\rangle$ extratropical | $\langle\text{RMSE}\rangle$ tropical | $\langle\text{RMSE}\rangle$ ocean | $\langle\text{RMSE}\rangle$ full | $\langle\dim_{KY}\rangle$ |
|---|---|---|---|---|---|---|
| CLVs - 9 (full rank) | $y_e, y_t, Y$ $[1, 1, 25]$ | 0.3111 | 0.1619 | 0.4951 | 0.4017 | 5.9311 |
| CLVs - 5 (unstable/neutral subspace + 1) | $y_e, y_t, Y$ $[1, 1, 25]$ | 0.3338 | 0.3019 | 1.0080 | 0.6967 | 5.9327 |
| CLVs - 6 (global dimension) | $y_e, y_t, Y$ $[1, 1, 25]$ | 0.3256 | 0.2002 | 0.6623 | 0.4958 | 5.9317 |
| CLVs - variable (local dimension) | $y_e, y_t, Y$ $[1, 1, 25]$ | 0.3196 | 0.1881 | 0.5963 | 0.4573 | 5.9279 |

**Table 2.** Summary metrics of DA experiments using right-transform matrix (15) and benchmark observations $(y_e, y_t, Y)$. The angle brackets $\langle\cdot\rangle$ denote average over analysis steps. Compare to results in Yoshida and Kalnay (2018). Parameters: analysis window 0.08, inflation factor 1%, 10 ensemble members.

We observe that all experiments generally succeed at constraining the full system. The trajectories, spread, increments, and error metrics of the variable CLV experiments are shown in Figure 6; all other experiments behave qualitatively similarly.
When comparing the reduced subspace experiments to the full rank case in Table 2, we find that using 5 CLVs (unstable, neutral, and one stable) performs worse than the other CLV experiments, and there is more than a 100% increase in the average RMSE of the ocean subsystem compared to full rank. While using the asymptotic Kaplan-Yorke dimension (6 CLVs) shows





improvement over 5 CLVs, the most improvement is in the variable CLV case. Although all methods produce a comparable average dimension (last column of Table 2), our experiments show that taking into account local variations in dimension is most effective.

To take a closer inspection of the dynamics during the assimilation, Figure 7 shows some dynamical properties of the variable
CLV experiment in time. The top panel shows the FTLEs computed from the ensemble mean. The first thing to notice is the correlation of the temporal FTLE variations. The most unstable and most stable modes have the same frequency of variability and remain highly correlated throughout the whole experiment. The lower frequency variations seen in the second most stable mode are often correlated with some of the weakly stable modes. These seem to have the biggest impact on the local dimension (shown in panel b). We can compare the changes in FTLEs and local dimension to the rank of the covariance matrix (panel c).
A decrease in local dimension typically occurs when one of the weakly stable modes becomes more stable, and therefore less CLVs are needed to span the local unstable and neutral subspace. On the contrary, when the dimension increases the weakly stable modes have become more unstable, at times even becoming positive. This implies that more CLVs are needed to span the local unstable and neutral subspace, and therefore the rank of the covariance matrix increases. The local Kolmogorov-Sinai entropy is also shown with the local dimension. We see that directly before a full dimension collapse ($\dim_{KY} = 0$), there is
a spike in local entropy. A collapse in dimension here occurs when all the FTLEs become negative. This does not impact the effectiveness of the method, however, as it just enforces the covariance matrix and Kalman gain to be zero for that analysis step. The ensemble members are not adjusted and therefore left to evolve accordingly. The bottom panel of Figure 7 shows the statistics of the FTLEs. The mean values, shown as a full circle, are used to calculate the average dimension in Table 2. These can be compared to the asymptotic values computed in Section 2.2 which are shown as open circles to the right of the FTLE
statistics. We also show the standard deviation for each of the FTLEs. The largest standard deviations are found in the fifth and sixth FTLEs, which correspond to the first two stable modes of the system. This supports the hypothesis that the weakly stable modes are most influential in the variation of local dimension. We also see that the maximum values (squares) are greater than (or equal to) zero. This implies that at a given time the fifth and sixth modes have moved into the unstable/neutral subspace and additional modes are therefore needed to account for nonlinear error growth.

## 5.3 Atmosphere observations

We now consider the case where only the two atmosphere subsystems are observed, however the observations are less sparse in that we take both the $y$ and $z$ components from each subsystem. Yoshida and Kalnay (2018) considered a similar case with only $y$ component observations were used with the cross-covariances in the atmosphere, but the ocean was also observed and assimilated separately. Here we rely only on the cross-covariances to recover the ocean subsystem, as the ocean is strongly
coupled to the tropical atmosphere. For this case we use the following settings: analysis window 0.08, observation error covariance matrix $\mathbf{R} = I_4$, inflation factor 1%, and 10 ensemble members. The model is run for the same amount of time as in the benchmark observations experiments.

Table 3 shows the summary statistics of performing full rank DA (9 CLVs), rank equal to the number of CLVs corresponding to asymptotic dimension, and a variable rank equal to number of CLVs corresponding to local dimension. In such a case where





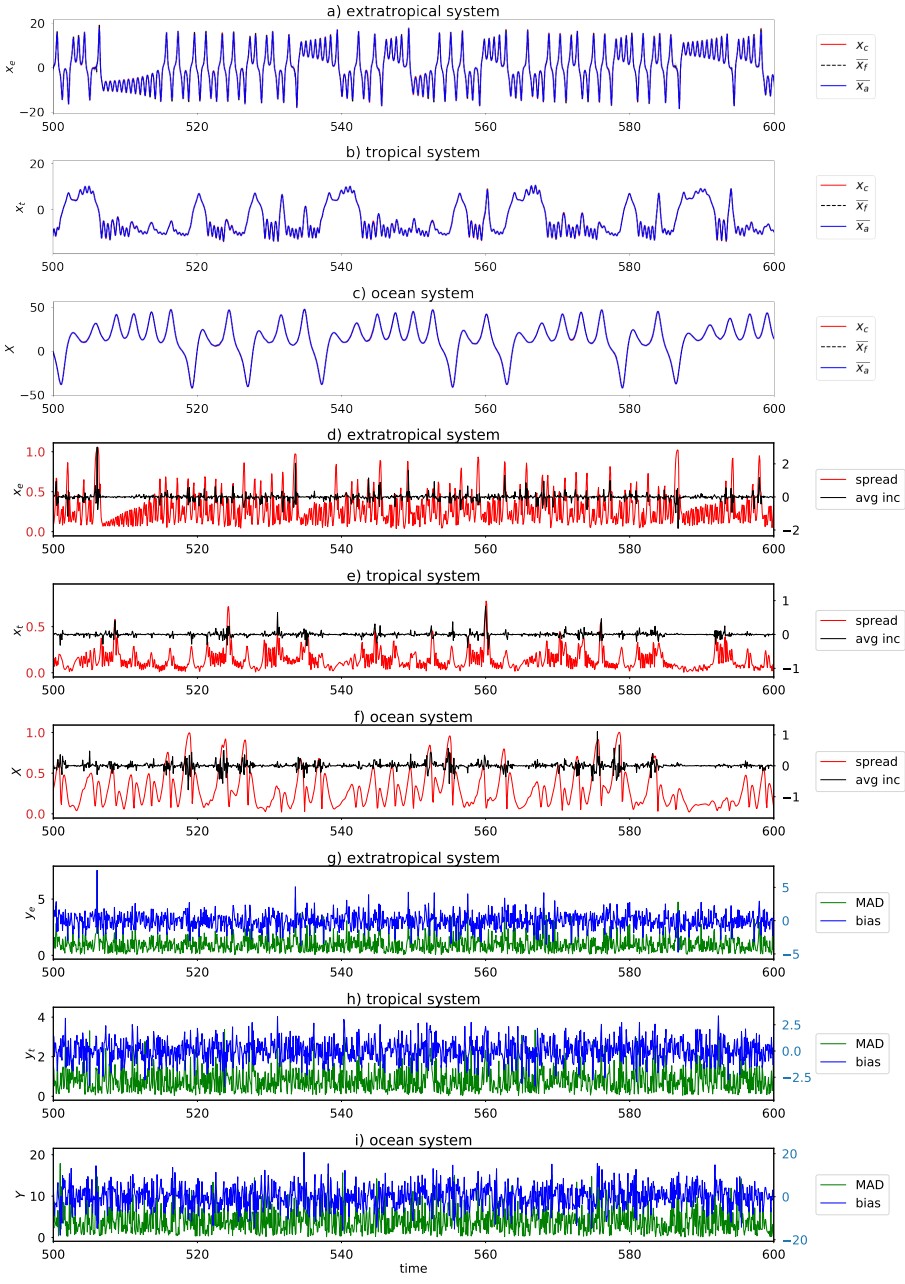

**Figure 6.** Segment of DA using the variable CLV method and benchmark observations $(y_e, y_t, Y)$. (a-c) Trajectories shown are control run (red), ensemble mean (blue), and individual ensemble members (faint). (d-f) Metrics shown are size of ensemble spread (red) and ensemble mean increment (black). (g-i) Metrics shown are mean absolute deviation (MAD, red) and bias (blue). For conciseness we only show the results for one coordinate of each subsystem. The other two coordinates behave similarly. Parameters: analysis window 0.08, inflation factor 1%, 10 ensemble members, observation error covariance matrix $\mathbf{R} = \mathrm{diag}([1, 1, 25])$.

**Figure 7.** Local attractor properties of DA using the variable CLV method and benchmark observations $(y_e, y_t, Y)$. (a) FTLEs calculated from the ensemble mean trajectory over a window of $\tau = 4$. (b) Local Kaplan-Yorke dimension and Kolmogorov-Sinai entropy computed through (2) and (3) using the FTLEs at the given time. (c) Rank of covariance matrix. (d) Statistics of first 7 FTLEs compared to asymptotic values.




| Method | Observations [error variance] | ⟨RMSE⟩ extratropical | ⟨RMSE⟩ tropical | ⟨RMSE⟩ ocean | ⟨RMSE⟩ full | ⟨dim$_{KY}$⟩ |
|---|---|---|---|---|---|---|
| CLVs - 9 | $y_e, z_e, y_t, z_t$ $[1,1,1,1]$ | 0.1757 | 0.1392 | 0.5892 | 0.4598 | 5.9134 |
| CLVs - 6 | $y_e, z_e, y_t, z_t$ $[1,1,1,1]$ | 0.6656 | 3.5193 | 18.7334 | 12.8749 | 5.8591 |
| CLVs - variable | $y_e, z_e, y_t, z_t$ $[1,1,1,1]$ | 0.1890 | 0.1804 | 0.8146 | 0.6062 | 5.9152 |

**Table 3.** Summary metrics of DA experiments using right-transform matrix (15) and atmosphere observations $(y_e, z_e, y_t, z_t)$. Parameters: analysis window 0.08, inflation factor 1%, 10 ensemble members.

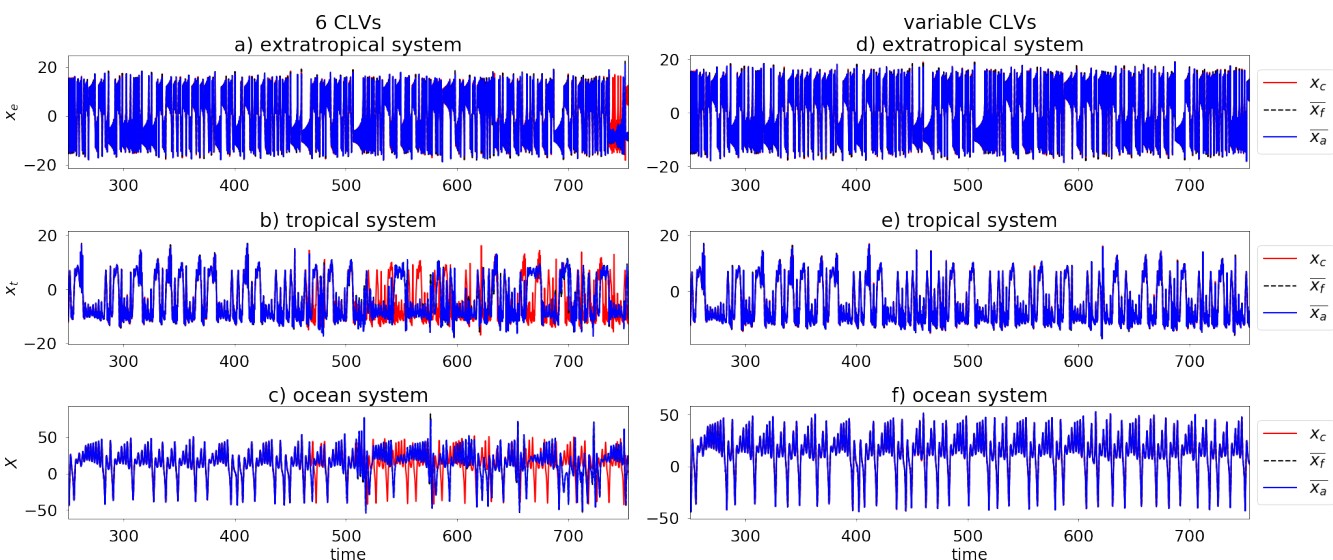

**Figure 8.** Trajectories of DA using 6 CLVs (a-c) and variable CLVs (d-f) with atmosphere observations $(y_e, z_e, y_t, z_t)$. Trajectories shown are control run (red), ensemble mean (blue), and individual ensemble members (faint). For conciseness we only show the results for the x-coordinate of each subsystem. The other two coordinates behave similarly. Parameters: analysis window 0.08, inflation factor 1%, 10 ensemble members, observation error covariance matrix $\mathbf{R} = I_4$.

we rely only on observations from the fast timescale subsystems, the asymptotic dimension is insufficient for tracking the system. When using a set number of 6 CLVs the tropical atmosphere and ocean subsystems begin following a different trajectory than the control run (Figure 8). This eventually also leads to the extratropical subsystem following a different trajectory ($t \approx 730$). On the other hand, the variable CLV method has no difficulty in tracking the control run for all the subsystems, and

5     it has summary statistics comparable to the full rank experiment (Figure 8 and Table 3).





We again analyse the dynamical properties of the variable CLV experiment to understand why the 6 CLV method fails to track the control run. The FTLEs, local Kaplan-Yorke dimension, local Kolmogorov-Sinai entropy, number of CLVs, and FTLE statistics are shown in Figure 9. We see that just before the tropical and ocean subsystems first fail to track the control run in the 6 CLV experiment ($t \approx 465$), there is a spike in the local Kolmogorov-Sinai entropy and the number of CLVs increases to

8. There are other instances where the local Kaplan-Yorke dimension calls for 8 CLVs, however the combination of the spike in entropy and dimension seems to be the driving factor behind the 6 CLV method failing. Note that this is unique to the case where one subsystem is unobserved and assimilated solely based on cross-covariances. There is also a spike in local entropy and dimension in the benchmark observations case (Figure 7), however the 6 CLV method succeeds in tracking the control run when all the subsystems are partially observed. The statistics of the FTLEs are similar to those of the benchmark case, except

the maximum value of the sixth FTLE is clearly larger and the minimum value of the first FTLE is no longer negative (hence the absence of total dimension collapse).

### 5.4 ENSO observations

The strong coupling and low frequency variation in the ocean and tropical atmosphere subsystems represent an ENSO-like variability. We therefore refer to the case of observing the $y$ and $z$ components from the tropical and ocean subsystems as

ENSO observations. Again, this is similar to a case studied in Yoshida and Kalnay (2018) except we don't assimilate the extratropical atmosphere at all and attempt to recover the variability solely through the cross-covariances. We don't expect to track the control trajectory of the extratropical system, but we are interested if we can preserve the variance of the ensemble mean for the extratropical attractor.

Since the extratropical subsystem is likely to be unconstrained with these observations, the ensemble mean will not be

accurately estimated. In such a case, the variable CLV method fails due to the fact that the first CLV (which corresponds to the directions of fastest error growth) is inaccurately calculated. Therefore the true directions of unstable growth are inaccurately sampled in the reduced space experiments. This is amplified by the fact that we are using uncorrelated observation errors; if the observation errors have a temporal correlation, the dominant direction of nonlinear unstable growth can be ascertained even without tracking the extratropical subsystem (see following section on shadowed observations). The inaccurate dimension

reduction leads to exponential growth in the system and numerical instability. For this reason we turn our focus only on the full rank (9 CLV) method and the accuracy of the observations.

We use the following settings for all the DA experiments with ENSO observations: analysis window 0.08, inflation factor 1%, and 10 ensemble members. The model is run for the same amount of time as in the benchmark observations experiments. We study the effect of reducing the observation error variances in $\mathbf{R}$: standard observation error ($\mathbf{R} = \mathrm{diag}([1, 1, 25, 25])$),

reduced tropical atmosphere error ($\mathbf{R} = \mathrm{diag}([0.01, 0.01, 25, 25])$), reduced ocean error ($\mathbf{R} = \mathrm{diag}([1, 1, 0.25, 0.25])$), and reduced overall error ($\mathbf{R} = \mathrm{diag}([0.01, 0.01, 0.25, 0.25])$). The summary statistics are listed in Table 4 for all the experiments.

We find that for the standard observation errors, there is a collapse in the variance of the ensemble mean for the extratropical subsystem (Figure 10). However, when the observation error for the tropical subsystem is reduced, that variance is significantly increased. This can be seen through the increase in the average local Kaplan-Yorke dimension (Table 4). There is also a slight







**Figure 9.** Local attractor properties of DA using variable CLVs and atmosphere observations $(y_e, z_e, y_t, z_t)$. (a) FTLEs calculated from the ensemble mean trajectory over a window of $\tau = 4$. (b) Local Kaplan-Yorke dimension and Kolmogorov-Sinai entropy computed through (2) and (3) using the FTLEs at the given time. (c) Rank of covariance matrix. (d) Statistics of first 7 FTLEs compared to asymptotic values.




| Method | Observations [error variance] | $\langle$RMSE$\rangle$ extratropical | $\langle$RMSE$\rangle$ tropical | $\langle$RMSE$\rangle$ ocean | $\langle$RMSE$\rangle$ full | $\langle\dim_{KY}\rangle$ |
|---|---|---|---|---|---|---|
| CLVs - 9 | $y_t, z_t, Y, Z$ $[1, 1, 25, 25]$ | 6.9336 | 0.1488 | 0.3626 | 4.6428 | 4.5416 |
| CLVs - 9 | $y_t, z_t, Y, Z$ $[0.01, 0.01, 25, 25]$ | 5.5029 | 0.0479 | 0.1742 | 3.6743 | 5.4347 |
| CLVs - 9 | $y_t, z_t, Y, Z$ $[1, 1, 0.25, 0.25]$ | 7.0781 | 0.1277 | 0.0838 | 4.7212 | 4.6362 |
| CLVs - 9 | $y_t, z_t, Y, Z$ $[0.01, 0.01, 0.25, 0.25]$ | 5.1399 | 0.0457 | 0.0901 | 3.4286 | 5.4480 |

**Table 4.** Summary metrics of DA experiments using right-transform matrix (15) and ENSO observations ($y_t, z_t, Y, Z$). Parameters: analysis window 0.08, inflation factor 1%, 10 ensemble members.

increase in ability to track the extratropical subsystem of the control run. We note that decreasing the ocean observation error alone does not provide any improvements over the total error statistics or dimension, actually making them slightly worse. When the observation error of both subsystems is reduced, there is only a small improvement to the overall error statistics and dimension in comparison to the reduced tropical error case. The improvement is most notable for the cases with reduced tropical observation errors due to the tropical system's weak direct coupling to the extratropical system.

## 5.5 Shadowed observations

In this section we explore a different type of observation error. Rather than randomly perturbing the control run to form our observation points, we use a trajectory that shadows the control run which produces correlated observational errors. In other




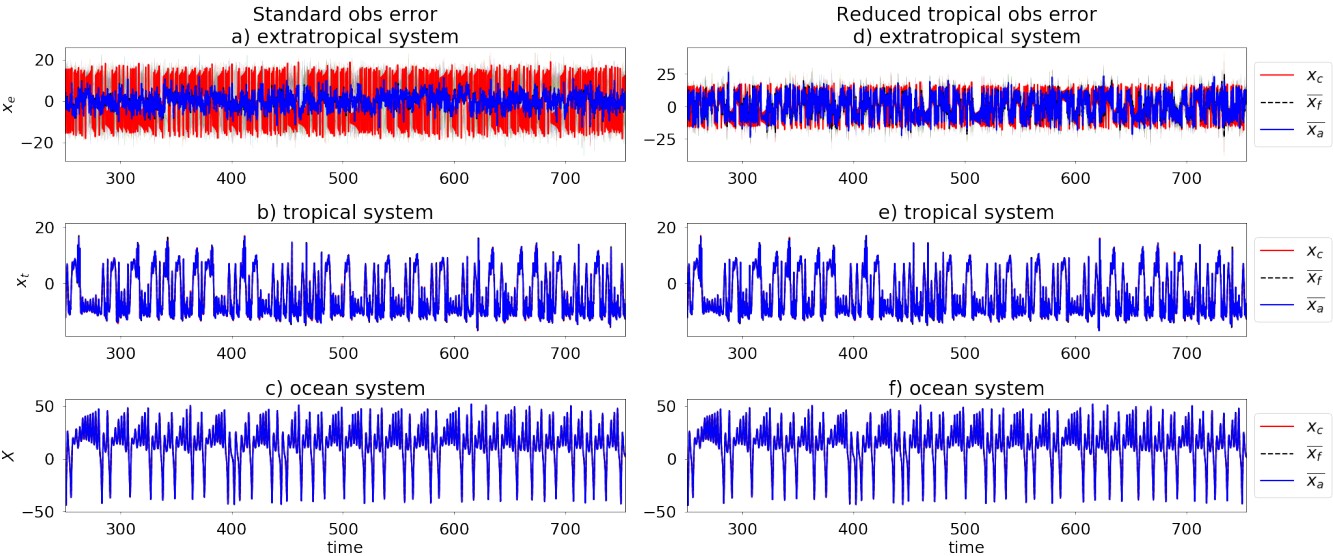

**Figure 10.** Trajectories of DA using 9 CLVs, $\mathbf{R} = \text{diag}([1, 1, 25, 25])$ (a-c) and $\mathbf{R} = \text{diag}([0.01, 0.01, 25, 25])$ (d-f) with ENSO observations $(y_t, z_t, Y, Z)$. Trajectories shown are control run (red), ensemble mean (blue), and individual ensemble members (faint). For conciseness we only show the results for the x-coordinate of each subsystem. The other two coordinates behave similarly. Parameters: analysis window 0.08, inflation factor 1%, 10 ensemble members.

words, we construct a new trajectory with a relaxation to the control run. This is implemented into the model as follows:

$$\dot{\tilde{x}}_e = \sigma(\tilde{y}_e - \tilde{x}_e) - c_e(S\tilde{x}_t + k_1), \tag{24a}$$

$$\dot{\tilde{y}}_e = \rho\tilde{x}_e - \tilde{y}_e - \tilde{x}_e\tilde{z}_e + c_e(S\tilde{y}_t + k_1) + \alpha_1(y_e - \tilde{y}_e), \tag{24b}$$

$$\dot{\tilde{z}}_e = \tilde{x}_e\tilde{y}_e - \beta\tilde{z}_e, \tag{24c}$$

$$\dot{\tilde{x}}_t = \sigma(\tilde{y}_t - \tilde{x}_t) - c(S\tilde{X} + k_2) - c_e(S\tilde{x}_e + k_1), \tag{24d}$$

$$\dot{\tilde{y}}_t = \rho\tilde{x}_t - \tilde{y}_t - \tilde{x}_t\tilde{z}_t + c(S\tilde{Y} + k_2) + c_e(S\tilde{y}_e + k_1) + \alpha_2(y_t - \tilde{y}_t), \tag{24e}$$

$$\dot{\tilde{z}}_t = \tilde{x}_t\tilde{y}_t - \beta\tilde{z}_t + c_z\tilde{Z}, \tag{24f}$$

$$\dot{\tilde{X}} = \tau\sigma(\tilde{Y} - \tilde{X}) - c(\tilde{x}_t + k_2), \tag{24g}$$

$$\dot{\tilde{Y}} = \tau\rho\tilde{X} - \tau\tilde{Y} - \tau S\tilde{X}\tilde{Z} + c(\tilde{y}_t + k_2) + \alpha_3(Y - \tilde{Y}), \tag{24h}$$

$$\dot{\tilde{Z}} = \tau S\tilde{X}\tilde{Y} - \tau\beta\tilde{Z} - c_z\tilde{z}_t. \tag{24i}$$

It is sufficient to constrain the trajectory using the relaxation term only in the $y$-coordinates. The parameters $\alpha_1 = 2.75$, $\alpha_2 = 0.8$, and $\alpha_3 = 0.8$ are the relaxation strengths and $y_e(t)$, $y_t(t)$, and $Y(t)$ are taken from the control trajectory at the given time.

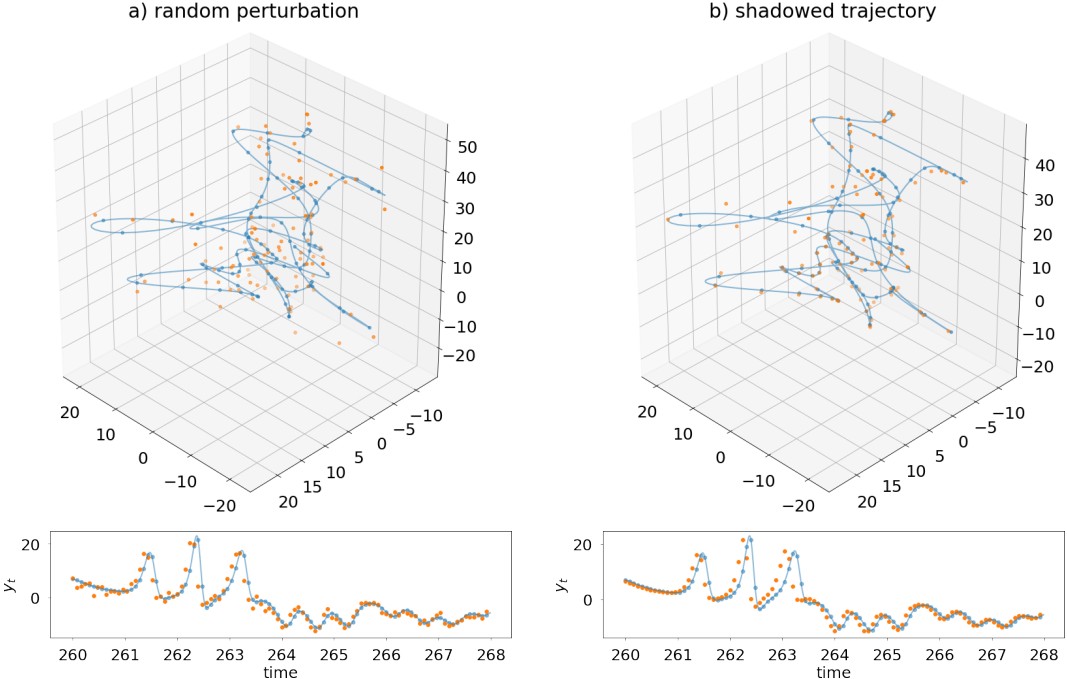

**Figure 11.** Comparison of the two different types of observations used in the data assimilation experiments. We show the observation space for the benchmark observations case $(y_e, y_t, Y)$. A selection of 800 time steps (100 observations) of the control model (blue line) and values at the observation times (blue dots) are shown in both plots, along with the actual observations used (orange dots). Top figures show behaviour along attractor and bottom panels show $y_t$ in time. (a) Observations formed by random perturbations to control run points. (b) Observations taken from a trajectory which shadows the control run.

We initialize the shadowed trajectory with a small perturbation to the control trajectory initial condition. We then propagate the shadowed trajectory along with the control trajectory, taking the observations from the shadowed trajectory at each assimilation cycle. Figure 11 shows the difference between the observations constructed using random perturbations and those from the shadowed trajectory. One benefit of the shadowed trajectory which is clearly visible in these figures is that it much more

5 closely maintains the structure of the attractor. This is not the case when using randomized perturbations, where the structure is not as discernible.

We repeat the observation experiments of the previous three sections: benchmark observations, atmospheric observations, and ENSO observations. We only focus on the full rank and variable CLV methods. When using correlated observational errors in any ensemble Kalman filtering method, a larger inflation value and ensemble size are needed to avoid ensemble collapse.

10 We find that for the standard ETKF method increasing the ensemble size to 11 members is sufficient. To facilitate a direct comparison, we therefore also use this ensemble size for the CLV experiments. The inflation value varies slightly with the different observation cases. The setup and results of the experiments are shown in Table 5.





| Method | Observations [error variance] | $\langle \text{RMSE} \rangle$ extratropical | $\langle \text{RMSE} \rangle$ tropical | $\langle \text{RMSE} \rangle$ ocean | $\langle \text{RMSE} \rangle$ full | $\langle \dim_{KY} \rangle$ |
|---|---|---|---|---|---|---|
| CLVs - 9 3% inflation | $\tilde{y}_e, \tilde{y}_t, \tilde{Y}$ $[1,1,25]$ | 0.4825 | 0.1604 | 0.4303 | 0.4361 | 5.9566 |
| CLVs - variable 3% inflation | $\tilde{y}_e, \tilde{y}_t, \tilde{Y}$ $[1,1,25]$ | 0.4926 | 0.1630 | 0.4158 | 0.4335 | 5.9722 |
| CLVs - 9 4% inflation | $\tilde{y}_e, \tilde{z}_e, \tilde{y}_t, \tilde{z}_t$ $[1,1,1,1]$ | 0.4215 | 0.1368 | 0.4648 | 0.4834 | 5.9602 |
| CLVs - variable 4% inflation | $\tilde{y}_e, \tilde{z}_e, \tilde{y}_t, \tilde{z}_t$ $[1,1,1,1]$ | 0.3773 | 0.1366 | 0.4663 | 0.4653 | 5.9582 |
| CLVs - 9 4% inflation | $\tilde{y}_t, \tilde{z}_t, \tilde{Y}, \tilde{Z}$ $[1,1,25,25]$ | 6.8620 | 0.1440 | 0.3067 | 4.5962 | 4.6264 |
| CLVs - variable 4% inflation | $\tilde{y}_t, \tilde{z}_t, \tilde{Y}, \tilde{Z}$ $[1,1,25,25]$ | 6.9726 | 0.1404 | 0.5528 | 4.7305 | 4.4854 |

**Table 5.** Summary metrics of DA experiments using right-transform matrix (15) and shadowed trajectory as observations. We set the observation error covariances to the standard values as in Yoshida and Kalnay (2018). Parameters: analysis window 0.08, 11 ensemble members, inflation as noted in table.

When using the benchmark observation set from the shadowed trajectory, the full rank and variable CLV methods perform comparably. This shows improvement over the case with random observation errors discussed in Section 5.2 where the variable CLV method performed slightly worse than the full rank. For the atmosphere observation case, we observe an improvement when using the variable CLV method over the full rank with the largest reduction of average RMSE in the extratropical

subsystem. The overall improvement is in contrast to the results when using random observation errors in Section 5.3; in those cases the full rank method outperformed the variable CLV method. The major change in the ENSO observation case from the random observational error experiments in Section 5.4 is that when using the shadowed trajectory observations the variable CLV method no longer fails. There is still difficulty in accurately calculating the first CLV from the extratropical subsystem being unconstrained, however the correlation in the errors of the tropical and ocean subsystems provide additional information

about the underlying nonlinear error growth. We observe that the variable CLV method performs comparably to the full rank, with a slightly larger average RMSE in the ocean subsystem (and subsequently the full). The inability to track the extratropical subsystem can once again be seen through the decrease in average local dimension.

## 5.6 Extratropical observations

We finally consider the case where one subsystem is fully observed and the others are completely unobserved. We choose to

observe the extratropical subsystem, as it is the most extreme case with weakly coupled fast dynamics. Due to the difficulty of constraining the full system with such minimal observations, the assimilation window is reduced to 0.02 and we use *perfect*





*observations*. In other words, we do not add any perturbations to the control run when taking the observation. Having more accurate observations was shown in Section 5.4 to improve the performance of the unobserved subsystems in the assimilation.

It is also clear from the previous experiments that the inability to constrain unobserved subsystems leads to a collapse in dimension, and correspondingly a collapse in the covariances. A collapse in covariance is commonly avoided through the use

of inflation (Anderson and Anderson, 1999; Hamill et al., 2001; Carrassi et al., 2008; Raanes et al., 2019). While a static background inflation avoids full covariance collapse, we are interested here in the covariance collapse related only to specific subsystems which aren't being constrained. For such a case, we argue that the forecast error covariance matrix should be scaled by a factor relating to the ensemble performance at each analysis step. In other words, when an individual subsystem is not being constrained, the covariances should be increased in the calculation of the Kalman gain, analogous to the approach

outlined by Miller et al. (1994) for the strongly nonlinear Lorenz '63 system. Here we introduce a new method for adaptive scaling of the Kalman gain. At each analysis step we multiply the forecast error covariance matrix by a spread-dependent scalar factor. The idea being that larger spread implies we have underestimated the covariances, and vice versa. We note that this adaptive scaling is different to traditional inflation as it does not directly adjust the underlying ensemble spread. We use this in conjunction with the static background inflation of $1\%$ to avoid total ensemble collapse.

We scale the error covariance matrix $\mathbf{P}^f$ by the Frobenius norm of $(\mathbf{X}^f)(\mathbf{X}^f)^{\mathrm{T}}$, where $\mathbf{X}^f$ is the ensemble spread matrix defined by (10b). This leads to a new scaled error covariance matrix $\hat{\mathbf{P}}^f$ defined by

$$\hat{\mathbf{P}}^f = ||(\mathbf{X}^f)(\mathbf{X}^f)^{\mathrm{T}}||(\mathbf{X}^f)(\mathbf{X}^f)^{\mathrm{T}}, \tag{25}$$

in the standard ESRF method or

$$\hat{\mathbf{P}}^f = ||(\mathbf{X}^f)(\mathbf{X}^f)^{\mathrm{T}}||(\mathbf{\Phi}\mathbf{W}\mathbf{W}^{\mathrm{T}}\mathbf{\Phi}^{\mathrm{T}}), \tag{26}$$

when using the CLVs. This rescaling factor is mathematically similar to the K-factor adaptive quality control procedure introduced by Sakov and Sandery (2017) and the $\beta$-factor rescaling of the background error covariances introduced by Bowler et al. (2013). The K-factor method was used to account for inconsistencies in observations and therefore uses an adaptive observation error covariance $\mathbf{R}$ that takes into account innovation size at each analysis step, while the $\beta$-factor is a constant multiplier to the forecast error covariance matrix to avoid the underestimation of the ensemble spread ($0 < \beta < 1$). Both the K-factor procedure

and the $\beta$-factor multiplier can be shown to have the same scaling effect on the Kalman gain $\mathbf{K}$ defined by (9a) as the adaptive scaling presented here, with the difference in that the modified $\hat{\mathbf{P}}^f$ in (25) takes into account both effects: small $||(\mathbf{X}^f)(\mathbf{X}^f)^{\mathrm{T}}||$ behaves like the $\beta$-factor and large $||(\mathbf{X}^f)(\mathbf{X}^f)^{\mathrm{T}}||$ behaves like the K-factor. We discuss the limiting behaviour of this adaptive scaling method in terms of the increment size and analysis error covariance in Appendix A.

Due to the fact that only the Kalman gain is being adjusted, for ease of implementation we use ESRF method introduced in

Section 4.1. This allows for the left-transform matrix to be calculated with the modified Kalman gain,

$$\hat{\mathbf{K}} = \hat{\mathbf{P}}^f \mathbf{H}^{\mathrm{T}}[\mathbf{H}\hat{\mathbf{P}}^f \mathbf{H}^{\mathrm{T}} + \mathbf{R}]^{-1}, \tag{27a}$$

$$\mathbf{T} = (\mathbf{I} - \hat{\mathbf{K}}\mathbf{H})^{1/2}. \tag{27b}$$





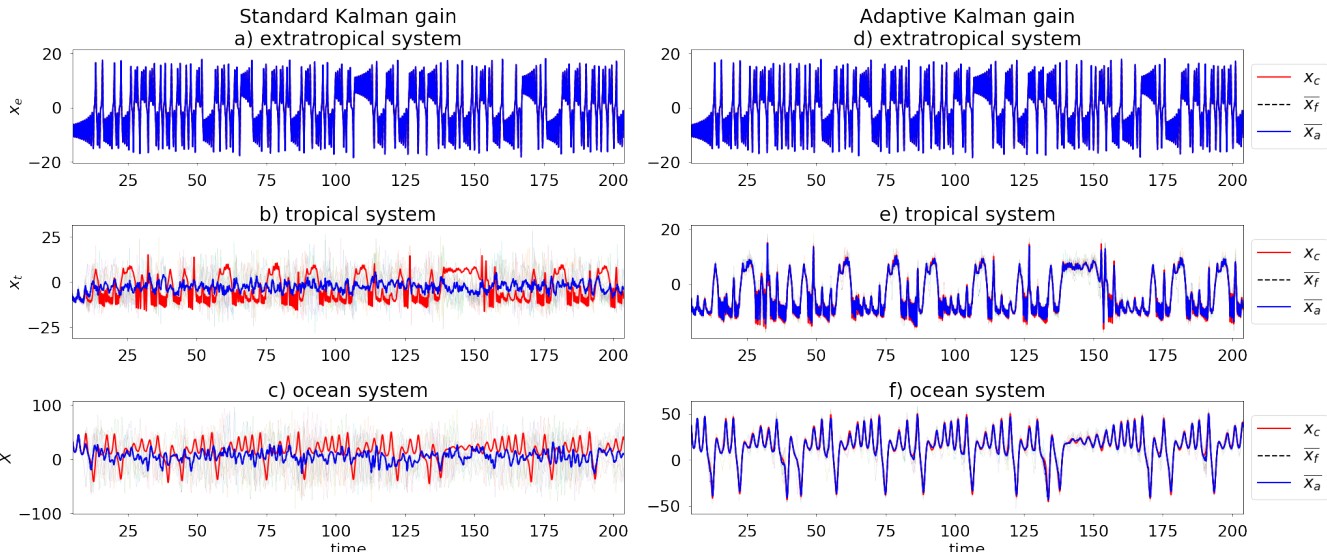

**Figure 12.** Trajectories of DA experiments using 9 CLVs, left-transform matrix (17b), and perfect observations from the extratropical subsystem of a control run $(x_e, y_e, z_e)$, with (a-c) the standard Kalman gain (9a) and (d-f) the adaptive Kalman gain (27a). Trajectories shown are control run (red), ensemble mean (blue), and individual ensemble members (faint). For conciseness we only show the results for the x-coordinate of each subsystem. The other two coordinates behave similarly. Parameters: analysis window 0.02, inflation factor 1%, 10 ensemble members, observations error covariance $\mathbf{R} = I_4$.

We also note that the variable CLV method will be less effective than using the full rank CLV matrix (or equivalent standard ESRF). The reduction in dimension is related to the unobserved subsystems being unconstrained. By using the variable CLV method, we are *a priori* setting the rank of the ensemble member matrix and covariance matrix. This can also have the effect of reducing some cross-covariances further, and thus the scaling implemented here may be ineffective. Therefore, in both methods we use the full rank (9 CLV) formulation of the covariance matrix. The results of both experiments, perfect observations and perfect observations with adaptive gain, are shown in Figure 12 and the error statistics listed in Table 6.

| Method | Observations [error variance] | $\langle\text{RMSE}\rangle$ extratropical | $\langle\text{RMSE}\rangle$ tropical | $\langle\text{RMSE}\rangle$ ocean | $\langle\text{RMSE}\rangle$ full | $\langle\text{dim}_{KY}\rangle$ |
|---|---|---|---|---|---|---|
| CLVs - 9 | $x_e, y_e, z_e$ $[1, 1, 1]$ | 0.0666 | 8.9288 | 38.5408 | 22.9953 | 4.1240 |
| CLVs - 9 adaptive gain | $x_e, y_e, z_e$ $[1, 1, 1]$ | 0.0034 | 0.7648 | 3.7789 | 2.2671 | 6.0229 |

**Table 6.** Summary metrics of DA experiments using left-transform matrix (17b) and the full extratropical subsystem as observations $(x_e, y_e, z_e)$. We use perfect observations (no random error added to the control run) with the observation error covariances set to the standard values as in Yoshida and Kalnay (2018). Parameters: analysis window 0.02, inflation factor 1%, 10 ensemble members.





We see from Figure 12 that there is remarkable improvement when using the adaptive gain method. Not only is the ensemble spread reduced in the unobserved subsystems, but the ensemble mean is also able to track the control run. This improvement in tracking the control run is exemplified in Table 6 with significant reduction in the average RMSE of individual subsystems. As expected, the average dimension is also significantly increased.

## 6    Concluding remarks

This study presents an initial understanding of the transient dynamics associated with the Kalman filter forecast error covariance matrix for nonlinear multiscale coupled systems. We have demonstrated the varying rank of the error covariance matrix related to the transient growth in the stable modes of the system. Additionally, we have shown the large impact of using isolated observations and cross-domain covariances in such a coupled system. The cross-covariances are significantly underestimated when the observed subsystems are weakly coupled to the unobserved, however this can be compensated through either reduced observational error or the use of an adaptive scaling of the Kalman gain.

The dynamical properties of strongly coupled DA in a multiscale system were investigated through a low-dimensional nonlinear chaotic model to represent the interactions between the extratropical atmosphere, ocean, and tropical atmosphere-ocean interface. The model contains significant spatio-temporal scale separations between the subsystems, as well as varying coupling strengths. We introduced a local dimension measure, namely the Kaplan-Yorke dimension calculated using FTLEs, to determine the rank of the forecast error covariance matrix at each analysis step. We have shown that through using time-varying CLVs to form a reduced rank forecast error covariance matrix, you achieve comparable results to the full rank ETKF and ESRF schemes.

We considered a benchmark experiment previously explored in Yoshida and Kalnay (2018) to examine the most effective number of CLVs needed to form the forecast error covariance matrix. We found that when using less than full rank, the variable amount based on local dimension performed the best. We also found there was not significant improvement when increasing to full rank. In particular, we found that spanning the space comprised of the asymptotic unstable, neutral, and first weakly stable mode (5 CLVs in this case) performed much worse than using either dimension measure (asymptotic and local). This suggests that significant growth occurring in more than one weakly stable mode is important when capturing short-term dynamics of highly nonlinear systems. We see improvement when implementing a rank based on local dimension over asymptotic dimension, however both produce successful results in this case where all subsystems are sampled in the observations.

We then tested the effectiveness of the reduced rank forecast error covariance matrix in strong CDA when a subsystem is completely unobserved, *i.e.* using only cross-covariances to determine the increment of the unobserved system. The first set of these experiments used observations from the two atmosphere subsystems, extratropical and tropical, while the ocean was left completely unobserved. In this case we found that the DA failed to constrain the system to the observations when using the asymptotic dimension to determine the rank of the covariance matrix, however both the variable and full rank methods succeeded. The second set of experiments consisted of ENSO observations, or observations from the strongly coupled tropical





and ocean subsystems only. In this case, the observational errors and weak coupling to the extratropical subsystem caused the reduced rank experiment to fail. The full rank experiment succeeded in tracking the tropical and ocean subsystems but left the extratropical subsystem unconstrained. This resulted in a collapse of the variance in the ensemble mean and a subsequent reduction in average dimension. However, we found that reducing the observational error variance of the tropical subsystem

provided an increase in ensemble mean variance of the extratropical subsystem and therefore an increase in average dimension. Reducing the observational error variance of the ocean did not provide a significant improvement since it is only indirectly coupled to the extratropical subsystem.

The effect of correlated observational errors was also explored. We constructed a trajectory which shadowed the control run and used this as our observational set, repeating all the previous experiments with different observation subsets. Since the

correlated errors preserve the underlying dynamical structure of the system, we found that the reduced rank method based on local dimension was more successful in all experiments when compared to those using random observational error. This included the ENSO observations case, where the extratropical subsystem remained unconstrained.

Finally, we showed that when only observing the extratropical subsystem, the unobserved subsystems could not be constrained due to their weak or indirect coupling to the observations. This manifested as an overall reduction in dimension as

well as a collapse in the cross-domain covariances. In order to counter the covariance and dimension collapse, we introduced a novel scheme for adaptive Kalman gain scaling. This adaptive scaling is based on a measure of the overall spread of the system, therefore accounting for unobserved subsystems that have become unconstrained. Through use of the adaptive scaling the weakly coupled unobserved subsystems were able to be relatively constrained, and moreover the ensemble mean of the unobserved subsystems was able to track the control run. The adaptive scaling introduced here can be applied to general systems

with weak coupling, although care may need to be taken in the choice of the norm.

We now turn to the implications on more realistic high dimensional systems. It has been shown that when using a finer model resolution (increasing dimension) there is an increase in near-zero asymptotic Lyapunov exponents (De Cruz et al., 2018). We observed through the examination of the dynamical properties of the coupled Lorenz system that the stable yet near-zero exponents have the largest temporal variability which affect the local dimension. As the number of near-zero exponents

increase, we may expect that the temporal variability in dimension will increase further. This would have strong implications on the necessary rank of the forecast error covariance and the subsequent number of ensemble members. It is not implausible that the number of ensemble members could vary significantly in time. In such a case where the model degrees of freedom is much larger than its effective dimension, the projection onto CLVs becomes even more effective. This would ensure the ensemble perturbations lie in subspaces associated with error growth at the given time, and that the directions of error growth

are sufficiently sampled. Such improvements in modelling error growth of high dimensional atmospheric systems has already been seen through the use of finite-time normal modes (Wei and Frederiksen, 2005). There is still more work to be done on how CLVs relate to meteorological and climatic events in such models, similar to the blocking studies of Schubert and Lucarini (2016). Future work should also consider the numerical cost of CLV calculation and methods to increase efficiency for high dimensional systems.




**Appendix A: Limiting behaviour of adaptive Kalman gain scaling**

We address the implication of the adaptive Kalman gain scaling for the two extreme cases: ensemble collapse ($||(\mathbf{X}^f)(\mathbf{X}^f)^{\mathrm{T}}|| \to$ 0) and ensemble blow-up ($||(\mathbf{X}^f)(\mathbf{X}^f)^{\mathrm{T}}|| \to \infty$).

1. $||(\mathbf{X}^f)(\mathbf{X}^f)^{\mathrm{T}}|| \to 0$:

We consider the Kalman gain in the form

$$\mathbf{K} = \delta\mathbf{P}^f\mathbf{H}^{\mathrm{T}}[\delta\mathbf{H}\mathbf{P}^f\mathbf{H}^{\mathrm{T}} + \mathbf{R}]^{-1}, \tag{A1}$$

where $\delta = ||(\mathbf{X}^f)(\mathbf{X}^f)^{\mathrm{T}}||$ and the subscript from equation (9a) has been dropped for brevity. For $\delta \ll 1$, equation (A1) can be expanded to

$$\mathbf{K} = \delta\mathbf{P}^f\mathbf{H}^{\mathrm{T}}[\mathbf{R}^{-1} - \delta\mathbf{R}^{-1}\mathbf{H}\mathbf{P}^f\mathbf{H}^{\mathrm{T}}\mathbf{R}^{-1} + \mathcal{O}(\delta^2)]$$
$$= \delta\mathbf{P}^f\mathbf{H}^{\mathrm{T}}\mathbf{R}^{-1} - \delta^2\mathbf{P}^f\mathbf{H}^{\mathrm{T}}\mathbf{R}^{-1}\mathbf{H}\mathbf{P}^f\mathbf{H}^{\mathrm{T}}\mathbf{R}^{-1} + \mathcal{O}(\delta^3). \tag{A2}$$

Letting $\delta \to 0$, equation (A2) simplifies to $\mathbf{K} = \mathbf{0}$, the zero matrix. In this case the mean analysis increment and error covariance equations (9b-9c) simplify to

$$\overline{\boldsymbol{x}}^a = \overline{\boldsymbol{x}}^f, \tag{A3}$$
$$\mathbf{P}^a = \mathbf{P}^f. \tag{A4}$$

In other words, in the case of collapsed spread, the analysis is equal to the forecast.

2. $||(\mathbf{X}^f)(\mathbf{X}^f)^{\mathrm{T}}|| \to \infty$:

We consider the Kalman gain in the form

$$\mathbf{K} = \mathbf{P}^f\mathbf{H}^{\mathrm{T}}[\mathbf{H}\mathbf{P}^f\mathbf{H}^{\mathrm{T}} + \delta\mathbf{R}]^{-1}, \tag{A5}$$

where $\delta = \frac{1}{||(\mathbf{X}^f)(\mathbf{X}^f)^{\mathrm{T}}||}$. For $\delta \ll 1$, equation (A5) can be expanded to

$$\mathbf{K} = \mathbf{P}^f\mathbf{H}^{\mathrm{T}}[(\mathbf{H}\mathbf{P}^f\mathbf{H}^{\mathrm{T}})^{-1} - \delta(\mathbf{H}\mathbf{P}^f\mathbf{H}^{\mathrm{T}})^{-1}\mathbf{R}(\mathbf{H}\mathbf{P}^f\mathbf{H}^{\mathrm{T}})^{-1} + \mathcal{O}(\delta^2)]. \tag{A6}$$

Letting $\delta \to 0$, equation (A6) simplifies to

$$\mathbf{K} = \mathbf{P}^f\mathbf{H}^{\mathrm{T}}(\mathbf{H}\mathbf{P}^f\mathbf{H}^{\mathrm{T}})^{-1}. \tag{A7}$$

In this case the mean analysis increment and error covariance equations (9b-9c) become

$$\overline{\boldsymbol{x}}^a = \overline{\boldsymbol{x}}^f + \mathbf{P}^f\mathbf{H}^{\mathrm{T}}(\mathbf{H}\mathbf{P}^f\mathbf{H}^{\mathrm{T}})^{-1}(\boldsymbol{y} - \mathbf{H}\overline{\boldsymbol{x}}^f), \tag{A8}$$
$$\mathbf{P}^a = (\mathbf{I} - \mathbf{P}^f\mathbf{H}^{\mathrm{T}}(\mathbf{H}\mathbf{P}^f\mathbf{H}^{\mathrm{T}})^{-1}\mathbf{H})\mathbf{P}^f. \tag{A9}$$



Recall that $\boldsymbol{y}$ are the observations defined by $\boldsymbol{y} = \mathbf{H}\boldsymbol{x}$ where $\boldsymbol{x}$ is the truth (the observation error has implicitly been set as zero in the case of large spread). For a function $\mathcal{F}$ with a Taylor series expansion operating on two arbitrary matrices $\mathbf{A}$ and $\mathbf{B}$, we have the following identity:

$$\mathcal{F}(\mathbf{AB})\mathbf{A} = \mathbf{A}\mathcal{F}(\mathbf{BA}) \tag{A10}$$

Taking $\mathbf{A} = \mathbf{H}$, $\mathbf{B} = \mathbf{P}^f\mathbf{H}^{\mathrm{T}}$, and $\mathcal{F}$ the inverse function, we can simplify (A8-A9) to

$$\overline{\boldsymbol{x}}^a = \boldsymbol{x}, \tag{A11}$$

$$\mathbf{P}^a = \mathbf{0}. \tag{A12}$$

In such a case, all ensemble members are adjusted to the same value as inferred from the observations.

*Author contributions.* All authors designed the study. The schemes for the calculation of Lyapunov exponents and CLVs were adapted and

implemented by CQ in both Python and Matlab. The Python codes for the ensemble Kalman filtering methods were produced by VK, with modifications by CQ. All figures were produced by CQ. All authors contributed to the direction of the study, discussion of results, and the writing and approval of the manuscript.

*Competing interests.* The authors declare that they have no conflicts of interest.

*Acknowledgements.* The authors would like to thank Dylan Harries, Pavel Sakov, and Paul Sandery for their valuable input throughout the

preparation of this manuscript. The authors were supported by the Australian Commonwealth Scientific and Industrial Research Organisation (CSIRO) Decadal Climate Forecasting Project (https://research.csiro.au/dfp).





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
