# Peer review of "Application of local attractor dimension to reduced space strongly coupled data assimilation for chaotic multiscale systems"

_Nonlinear Processes in Geophysics, 2019_

## Referee Comment (RC1) · Anonymous Referee #1 · 1 Nov 2019

General comment

This manuscript is exploring the use of covariant Lyapunov vectors (CLVs) to build the error covariance matrix in ensemble Kalman filtring methods. The set of vectors is selected based on the computation of a local Kaplan-Yorke dimension based on the finite time Lyapunov exponents. This approach is implemented in the context of a muli-scale system mimicking the (coupled) dynamics of a coupled tropical ocean-atmosphere system and the extra-tropical atmosphere. Different strategies of observation are then evaluated. It is found that observation within the atmosphere is essential, and that the variable number of CLVs to be used in building the error covariance matrix

is a successful strategy for strongly coupled data assimilation. Very interesting results are also obtained with the observation sampling of a shadowing trajectory, leading to measurement correlations.

This is an interesting manuscript exploring many aspects of the strongly coupled data assimilation and I would in principle recommend publication of this work. I have however an important concern on the use of the local Kaplan Yorke dimension and the CLVs that should be addressed before publication. It seems to me that the use of both is inconsistent. Let me clarify my point. When computing the FTLEs, one can use either the QR (associated with the backward Lyapunov vectors) decomposition, the Forward Lyapunov vectors obtained with backward integration in time, or the local amplification along the CLVs. Although all these are giving the appropriate asymptotic Lyapunov exponents, they are not providing similar variability of these quantities as illustrated for instance in Vannitsem and Lucarini (2016) you quoted. So if you use the QR decomposition and then select the CLVs on that basis this is probably not optimal.

Alternatively, if you use an estimate of the FTLE using the amplification along the CLVs then the "dimension" of the subspace of instabilities is not the same and one can wonder what is the signification of this quantity. This is related to your comment at line 14 of page 8 indicating that higher dimension is associated with more important alignment of CLVs. Imagine for instance that several CLVs are pointing in almost the same direction with large amplifications, then the dimension would be large but intuitively (as they point all in the same direction) we would expect a low dimension. The specific way you compute the FTLEs and the local dimension should therefore be clarified, and probably I would not call it "dimension". Furthermore, if the KY dimension is computed with the local amplification rates of the backward Lyapunov exponents, a comparison should be made with the use of the backward Lyapunov vectors in building the error covariance matrix. It would have been my first choice in view of the fact this is much less costly than the CLVs.

More specific comments

1. Figure 3, you mentioned 2 neutral Lyapunov exponents. I am wondering why you have two such exponents. Is there any specific symmetries allowing for that? Is'nt it a numerical artifact?

2. Table 2. The RMSE for the extratropics are very close to each other whatever the experiments. Are the differences significant?

3. Also in Table 2. An average dimension is computed. This average is based on the QR decomposition or some estimate with the CLVs? This is related to my main point. Please clarify how it is computed.

4. FTLEs are computed for 4 time units. What is happening to your analysis when this window is changed? And why choosing this specific window?

5. At page 12, in the two first paragraphs of Section 5, you present how the experiment is done. As far as I understood, the CLVs are computed during a limited period of time during the assimilation period. Am I right? At first reading it was not very clear to me and it would be nice to improve the presentation of that part. In particular, a sketch of the whole process in a figure would be really useful.

6. In the partially observed CDA, you also compute a local dimension. I am wondering how the CLVs are computed there since the trajectory of the model is probably very far from reality. Moreover, it was not clear to me whether you are using the CLVs of the reality or of the model integration. Would you please clarify how you do this? It can maybe be incorporated in the general description of the experiments mentioned at point 5 above.

Minor points

1. Line 7, page 11. Please modify the notation on the brackets. It looks like a vertical rectangle.

2. Line 20, page 11. I suppose that ïĄň should be larger than 1.

3. Line 22, page 11. Please do not use the terminology "model error". It is confusing as model error is used when there is structural uncertainty in the model.

---

## Author Comment (AC1) · 29 Nov 2019

**Response to anonymous reviewer comments**

Dear Reviewer,

We first thank you for your positive and constructive comments. Below we attempt to address the your comments through a more detailed discussion. We will also make adjustments to the manuscript where necessary.

Many thanks,

Courtney Quinn, Terence O'Kane, and Vassili Kitsios
* * *
**General comments**

This manuscript is exploring the use of covariant Lyapunov vectors (CLVs) to build the error covariance matrix in ensemble Kalman filtring methods. The set of vectors is selected based on the computation of a local Kaplan-Yorke dimension based on the finite time Lyapunov exponents. This approach is implemented in the context of a multi-scale system mimicking the (coupled) dynamics of a coupled tropical ocean-atmosphere system and the extra-tropical atmosphere. Different strategies of observation are then evaluated. It is found that observation within the atmosphere is essential,and that the variable number of CLVs to be used in building the error covariance matrix is a successful strategy for strongly coupled data assimilation. Very interesting results are also obtained with the observation sampling of a shadowing trajectory, leading to measurement correlations.

This is an interesting manuscript exploring many aspects of the strongly coupled data assimilation and I would in principle recommend publication of this work.

We thank the reviewer for the positive evaluation.

I have however an important concern on the use of the local Kaplan Yorke dimension and the CLVs that should be addressed before publication. It seems to me that the use of both is inconsistent. Let me clarify my point. When computing the FTLEs, one can use either the QR (associated with the backward Lyapunov vectors) decomposition,the Forward Lyapunov vectors obtained with backward integration in time, or the local amplification along the CLVs. Although all these are giving the appropriate asymptotic Lyapunov exponents, they are not providing similar variability of these quantities as illustrated for instance in Vannitsem and Lucarini (2016) you quoted. So if you use the QR decomposition and then select the CLVs on that basis this is probably not optimal.

We have considered the reviewer's concern and are in agreement that the use of the two quantities could produce inconsistencies. To clarify, we use the QR decomposition method to calculate the FTLEs, which would give amplification rates for the backward Lyapunov vectors rather than the CLVs. However, in this case we argue that the way in which we utilise the FTLEs does not create an inconsistency with the CLVs in that we are not assigning any particular FTLE to a CLV. Rather, the backward FTLEs are used only to produce an adequate estimate of the number of CLVs to retain in constructing the covariance matrix. The true local dimension of the system may be different from the quantity computed, but the Kaplan-Yorke dimension calculated from the backward FTLEs still produces a sufficient upper bound. We expand on this point further below.

Figure 1 shows the finite-time Kaplan-Yorke dimension of a segment of the example model run from Sections 2 and 3 in the manuscript (compare to Figure 5 in manuscript). The top panel is the dimension measure, while the bottom is the ceiling of the dimension measure which is used to select the number of CLVs for constructing the covariance matrix. We compare the two different methods for calculating the CLVs that the reviewer mentions: growth rate along CLVs (FTCLEs) and QR decomposition (FTBLEs). We observe that for our particular nonlinear system, the QR decomposition method consistently gives a larger estimate for the Kaplan-Yorke measure than the amplification along CLVs, and therefore implies retaining more CLVs in the analysis. We maintain that a higher number of retained CLVs will not harm the DA, while too few CLVs potentially could. Specifically, where the positive CLVs are closely aligned, the DA requires the inclusion of most of the additional neutral and stable CLVs to avoid ensemble collapse (further discussion of this point on the following page). Thus, Kaplan-Yorke is an effective choice to determine the rank of the background covariance matrix despite the any inaccuracies due to degeneracy in the unstable directions. Additionally, we mention that in order to obtain the growth rate of the CLVs, one needs the accurate forward model. Since we are calculating the CLVs from the ensemble mean trajectory within the DA experiments, we do not have an accurate future trajectory and therefore no forward model (due to the state dependence of the Jacobian). For these reasons we maintain that the FTLEs from the QR method are a sufficient for our analysis at hand.

[Figure]

Figure 1: Comparison of finite-time Kaplan-Yorke dimension calculated using the growth rates of CLVs (FTCLEs) and the QR method (FTBLEs).

[Figure]

Figure 2: Comparison of variance of the individual FTLEs calculated using the growth rates of CLVs (FTCLEs) and the QR method (FTBLEs).

Alternatively, if you use an estimate of the FTLE using the amplification along the CLVs then the "dimension" of the subspace of instabilities is not the same and one can wonder what is the signification of this quantity. This is related to your comment at line 14 of page 8 indicating that higher dimension is associated with more important alignment of CLVs. Imagine for instance that several CLVs are pointing in almost the same direction with large amplifications, then the dimension would be large but intuitively (as they point all in the same direction) we would expect a low dimension. The specific way you compute the FTLEs and the local dimension should therefore be clarified, and probably I would not call it "dimension". Furthermore, if the KY dimension is computed with the local amplification rates of the backward Lyapunov exponents, a comparison should be made with the use of the backward Lyapunov vectors in building the error covariance matrix. It would have been my first choice in view of the fact this is much less costly than the CLVs.

The reviewer is correct in that the finite-time Kaplan-Yorke measure is not necessarily the true local dimension. However, this measure is giving an indication of local dimension combined with alignment. We explain this through an examination of two difference time steps in the benchmark DA experiment (Section 5.2 in the manuscript) where a high Kaplan-Yorke measure is recorded and the FTLE behaviour is similar. The alignment of the CLVs and the FTLEs (computed using the QR decomposition method) are shown in Figure 3. In both cases we see that there are 2 equally strong leading unstable FTLEs, 3 positive but near zero FTLEs, and one weakly stable FTLE. In the first column ($t = 306.24$) we see that the 5 leading CLVs are strongly aligned as well as the two most stable (8 and 9). The dimension based on alignment would then intuitively be around 4, and one would need to select the set of CLVs that are not aligned. In our method since $\dim_{KY} > 7$ we would retain 8 CLVs, therefore accounting for the strong alignment of the leading CLVs and retaining all necessary directions. In the second column ($t = 705.04$) the case is quite different in that the leading CLVs are not strongly aligned, but there is strong alignment of CLVs 4-6 and pairs 3,7 and 8,9. This would give an alignment-based dimension around 5, but again we need to retain up to the 8th CLV to span the local manifold. While one could create a method based on alignment for selecting directions, we point out that the actual criteria for "strong alignment" is arbitrary and one could risk excluding a significant direction.

With regards to using the backwards Lyapunov vectors (BLVs), if one is using the QR method then the backwards Lyapunov vectors computed are orthogonal by construction. We therefore do not see the same behaviour of alignment of the vectors. We ran the benchmark DA experiment (Section 5.2 in the manuscript) using a variable number of BLVs based on the Kaplan-Yorke measure computed using the QR decomposition method (statistics shown in Table 2 of next section). We see that the variable BLVs and variable CLVs perform comparably. While all experiments could be run with the BLVs as suggested by the reviewer, we aim to show the functionality of using the CLVs as they provide additional information about the phase-space dynamics which can be

[Figure]

Figure 3: We compare the alignment of the CLVs at two time steps of the benchmark DA experiment (section 5.2 in the manuscript). The two time steps have similar Kaplan-Yorke measure and distribution of leading FTBLEs, shown in the bottom panels. We observe that the behaviour of the alignment can be vastly different for similar FTBLE behaviour, however the method of retaining CLVs based on the Kaplan-Yorke measure gives a reliable way to reflect the true local dimension regardless of alignment.

analysed *a posteriori*.

We have made the following additions to the manuscript to address all the above concerns and clarify our methods:

- We have added a paragraph and figure to Section 3 of the manuscript to discuss the differences in calculating the FTLEs.

- We have added Figure 3 along with a discussion of alignment, dimension, and relevance to ensemble DA in Section 4.2.

- We have added the statistics for using BLVs with variable Kaplan-Yorke measure in the benchmark DA case of Section 5.2.

**Specific comments:**

1. Figure 3, you mentioned 2 neutral Lyapunov exponents. I am wondering why you have two such exponents. Is there any specific symmetries allowing for that? Isn't it a numerical artifact?

   For clarification, the model has a neutral and a near-neutral Lyapunov exponent. The near-neutral exponent comes from the weak coupling to the extratropical subsystem ($c_e = 0.08$). If one were to set $c_e = 0$, the extratropical subsystem would be completely uncoupled from the tropical and ocean subsystems, and it would retain its symmetry $[x_e, y_e, z_e] = [-x_e, -y_e, z_e]$. This implies an additional neutral Lyapunov exponent. However, we retain that the near-neutral exponent is important to consider within the neutral subspace in this study as the timescales of the neutral and near-neutral exponents are indistinguishable and changes in both can significantly affect the local Kaplan-Yorke measure. To improve and assess numerical precision, we have computed the Lyapunov exponents over a longer window (500,000 time steps) and show the results for the original parameter values as well as different combinations of coupling strengths set to zero in Table 1.

|  | $\lambda_1$ | $\lambda_2$ | $\lambda_3$ | $\lambda_4$ | $\lambda_5$ | $\lambda_6$ | $\lambda_7$ | $\lambda_8$ | $\lambda_9$ |
|---|---|---|---|---|---|---|---|---|---|
| as in manuscript | 0.9043 | 0.3052 | 0.0007 | -0.0032 | -0.4829 | -0.8008 | -1.8149 | -12.2359 | -14.5726 |
| $c_e = 0$ | 0.9083 | 0.3029 | 0.0001 | -0.0006 | -0.4814 | -0.7962 | -1.8172 | -12.2415 | -14.5744 |
| $c = 0$ | 0.9042 | 0.3491 | 0.0597 | -0.0002 | -0.0151 | -0.3186 | -1.6222 | -13.4793 | -14.5777 |
| $c_z = 0$ | 0.9081 | -0.0004 | -0.0723 | -0.0728 | -0.1283 | -0.1289 | -1.1599 | -13.4702 | -14.5753 |
| $c = c_z = 0$ | 0.9069 | 0.8886 | 0.0902 | 0.0001 | -0.0004 | -0.0741 | -1.4569 | -14.4801 | -14.5743 |
| $c_e = c = 0$ | 0.9083 | 0.3581 | 0.0692 | -0.0006 | -0.0007 | -0.3227 | -1.6360 | -13.5012 | -14.5744 |
| $c_e = c = c_z = 0$ | 0.9083 | 0.9083 | 0.0902 | 0.0001 | -0.0006 | -0.0006 | -1.4569 | -14.5744 | -14.5744 |

Table 1: Asymptotic Lyapunov exponents of coupled Lorenz model for different coupling coefficients set to 0. Lyapunov exponents computed over 5000 time units using a QR decomposition method, time step of 0.01, and orthogonalization step of 0.25.

2. Table 2. The RMSE for the extratropics are very close to each other whatever the experiments. Are the differences significant?

   We preface this response with pointing out that the statistics of Table 2 have slightly changed on fixing a numerical error in the code (see following section for revised statistics). In order to discuss significance of differences in the average RMSE we have to compute error bounds on the means. To do this, we perform a bootstrapping of the time series of extratropical RMSE for each of the five experiments listed in Table 2 of the next section. We resample the data (with replacement) 10,000 times

[Figure]

Figure 4: Histograms for the means of the resampled time-dependent extratropical RMSEs for each respective experiment. The black lines show the means of the histograms, which are approximately the values of extratropical average RMSE in Table 2 of the manuscript.

and then compute the average RMSE for each resample. This produces a distribution of average RMSE for each experiment, shown in Figure 4. Each distribution has standard deviation $\sigma \approx 0.004$, and we see that the mean of the distributions shift by less than $\sigma$ between experiments with the exception of the variable CLVs experiment. The mean of the variable CLVs experiment still lies within $2\sigma$ of all the experiment distributions. In this sense we can conclude that the values listed in Table 2 are not significantly different. Additionally, we note that in the sense of quantifying the DA performance, all five methods constrain the extratropical system to a similar degree.

3. Also in Table 2. An average dimension is computed. This average is based on the QR decomposition or some estimate with the CLVs? This is related to my main point. Please clarify how it is computed.

   The average dimension comes from the FTLEs which are calculated using the QR decomposition. We have clarified this in the manuscript.

4. FTLEs are computed for 4 time units. What is happening to your analysis when this window is changed? And why choosing this specific window?

   The window of 4 time units was chosen in order to capture dimension changes related to the ENSO-like excursions in the ocean subsystem (occurring anywhere between approximately 6.5 and 27 time units). The window should therefore be long enough to filter out the intrinsic oscillatory behaviour of the ocean subsystem but short enough to capture the transition to the excursion state. The period of intrinsic variability is approximately 3 time units, therefore any window between 3 and 6.5 time units should perform comparably. Decreasing and increasing the window leads to higher and lower temporal variability of the FTLEs, respectively.

5. At page 12, in the two first paragraphs of Section 5, you present how the experiment is done. As far as I understood, the CLVs are computed during a limited period of time during the assimilation period. Am I right? At first reading it was not very clear to me and it would be nice to improve the presentation of that part. In particular, a sketch of the whole process in a figure would be really useful.

    We have clarified the paragraph where we explain the computation of the CLVs within the DA framework.

6. In the partially observed CDA, you also compute a local dimension. I am wondering how the CLVs are computed there since the trajectory of the model is probably very far from reality. Moreover, it was not clear to me whether you are using the CLVs of the reality or of the model integration. Would you please clarify how you do this? It can maybe be incorporated in the general description of the experiments mentioned at point 5 above.

    The CLVs are calculated from the ensemble mean trajectory. In the cases where the full system is constrained (atmosphere-only observations), the ensemble mean remains close enough to the truth such that the CLVs can still be calculated to a sufficient accuracy. On the other hand, this is not the case when some of the subsystems are unconstrained (i.e. ENSO and extratropical-only observations). In such cases we cannot use the variable dimension method, and instead we use all 9 computed CLVs. This is valid since the forecast error covariance matrix has rank 9 in this set-up and we are projecting onto a basis of equivalent rank, regardless of the accuracy of the computed CLVs. The local dimension we are calculating comes from the QR decomposition as mentioned in the response to the main comment. While the CLVs (or BLVs) may be quite inaccurate at a given point in time, the average dimension can give an idea of variance in the unobserved subsystems. In cases where an unobserved subsystem is unconstrained, it is common to observe variance collapse in the ensemble mean of that subsystem and we subsequently record a lower average local dimension. We therefore explore methods to maintain the variance in the unobserved subsystems (i.e. increase the average local dimension), with the caveat that the instantaneous dimension will not necessarily be reflective of the true dynamics of the system. We have provided additional clarification in the manuscript about the calculation of the dynamical properties within the DA experiments.

**Minor points:**

1. Line 7, page 11. Please modify the notation on the brackets. It looks like a vertical rectangle.

   We have changed the text to read "...and the **subscript** $*, n$ denotes taking ...".

2. Line 20, page 11. I suppose that $\lambda$ should be larger than 1.

   We have added a condition to Equation 19 specifying $\lambda > 1$.

3. Line 22, page 11. Please do not use the terminology "model error". It is confusing as model error is used when there is structural uncertainty in the model.

   We have removed the term "model" as we agree that the usage is confusing in this instance.
* * *
**Minor corrections to DA experiments**

After submission we discovered a small error in the orthogonalisation step of the CLV calculation within the DA experiments. In correcting we found that using the CLV without orthogonalisation to the backwards subspaces (Algorithm 2.1 in [1]) was more appropriate due to the lack of an accurate forward model within the DA implementation. This has minor impact on the results of the experiments, however we include the tables with the updated statistics below which we will also update in the new manuscript.

**References**

[1] Gary Froyland, Thorsten Hüls, Gary P Morriss, and Thomas M Watson. Computing covariant lyapunov vectors, oseledets vectors, and dichotomy projectors: A comparative numerical study. *Physica D: Nonlinear Phenomena*, 247(1):18–39, 2013.

[2] Takuma Yoshida and Eugenia Kalnay. Correlation-cutoff method for covariance localization in strongly coupled data assimilation. *Monthly Weather Review*, 146(9):2881–2889, 2018.

| Method | Observations [error variance] | $\langle$RMSE$\rangle$ extratropical | $\langle$RMSE$\rangle$ tropical | $\langle$RMSE$\rangle$ ocean | $\langle$RMSE$\rangle$ full | $\langle\dim_{KY}\rangle$ |
|---|---|---|---|---|---|---|
| CLVs - 9 (full rank) | $y_e, y_t, Y$ $[1, 1, 25]$ | 0.3142 | 0.1598 | 0.4948 | 0.4027 | 5.8928 |
| CLVs - 5 (unstable/neutral subspace + 1) | $y_e, y_t, Y$ $[1, 1, 25]$ | 0.3123 | 0.1843 | 0.5920 | 0.4550 | 5.8870 |
| CLVs - 6 (global dimension) | $y_e, y_t, Y$ $[1, 1, 25]$ | 0.3156 | 0.1674 | 0.5513 | 0.4310 | 5.8892 |
| CLVs - variable (local dimension) | $y_e, y_t, Y$ $[1, 1, 25]$ | 0.3215 | 0.1688 | 0.5346 | 0.4272 | 5.8863 |
| BLVs - variable (local dimension) | $y_e, y_t, Y$ $[1, 1, 25]$ | 0.3149 | 0.1658 | 0.5122 | 0.4141 | 5.8895 |

Table 2: Summary metrics of DA experiments using right-transform matrix (ETKF) and benchmark observations $(y_e, y_t, Y)$. The angle brackets $\langle\cdot\rangle$ denote average over analysis steps. Compare to results in [2]. Parameters: analysis window 0.08, inflation factor 1%, 10 ensemble members.

| Method | Observations [error variance] | $\langle$RMSE$\rangle$ extratropical | $\langle$RMSE$\rangle$ tropical | $\langle$RMSE$\rangle$ ocean | $\langle$RMSE$\rangle$ full | $\langle\dim_{KY}\rangle$ |
|---|---|---|---|---|---|---|
| CLVs - 9 | $y_e, z_e, y_t, z_t$ $[1, 1, 1, 1]$ | 0.1734 | 0.1332 | 0.5782 | 0.4515 | 5.8672 |
| CLVs - 6 | $y_e, z_e, y_t, z_t$ $[1, 1, 1, 1]$ | 0.1715 | 0.1386 | 0.5807 | 0.4524 | 5.8718 |
| CLVs - variable | $y_e, z_e, y_t, z_t$ $[1, 1, 1, 1]$ | 0.1697 | 0.1383 | 0.5659 | 0.4433 | 5.8681 |

Table 3: Summary metrics of DA experiments using right-transform matrix (ETKF) and atmosphere observations $(y_e, z_e, y_t, z_t)$. Parameters: analysis window 0.08, inflation factor 1%, 10 ensemble members.

| Method | Observations [error variance] | $\langle$RMSE$\rangle$ extratropical | $\langle$RMSE$\rangle$ tropical | $\langle$RMSE$\rangle$ ocean | $\langle$RMSE$\rangle$ full | $\langle\dim_{KY}\rangle$ |
|---|---|---|---|---|---|---|
| CLVs - 9 | $y_t, z_t, Y, Z$ $[1, 1, 25, 25]$ | 7.0467 | 0.1571 | 0.3675 | 4.7182 | 4.5449 |
| CLVs - 9 | $y_t, z_t, Y, Z$ $[0.01, 0.01, 25, 25]$ | 5.5203 | 0.0480 | 0.1764 | 3.6860 | 5.4052 |
| CLVs - 9 | $y_t, z_t, Y, Z$ $[1, 1, 0.25, 0.25]$ | 7.1308 | 0.1386 | 0.0873 | 4.7566 | 4.4959 |
| CLVs - 9 | $y_t, z_t, Y, Z$ $[0.01, 0.01, 0.25, 0.25]$ | 5.1697 | 0.0431 | 0.0822 | 3.42478 | 5.4899 |

Table 4: Summary metrics of DA experiments using right-transform matrix (ETKF) and ENSO observations $(y_t, z_t, Y, Z)$. Parameters: analysis window 0.08, inflation factor 1%, 10 ensemble members.

| Method | Observations [error variance] | $\langle\text{RMSE}\rangle$ extratropical | $\langle\text{RMSE}\rangle$ tropical | $\langle\text{RMSE}\rangle$ ocean | $\langle\text{RMSE}\rangle$ full | $\langle\dim_{KY}\rangle$ |
|---|---|---|---|---|---|---|
| CLVs - 9 3% inflation | $\tilde{y}_e, \tilde{y}_t, \tilde{Y}$ $[1,1,25]$ | 0.5200 | 0.1635 | 0.4255 | 0.4552 | 5.9576 |
| CLVs - variable 3% inflation | $\tilde{y}_e, \tilde{y}_t, \tilde{Y}$ $[1,1,25]$ | 0.4848 | 0.1568 | 0.4095 | 0.4263 | 5.9371 |
| CLVs - 9 3% inflation | $\tilde{y}_e, \tilde{z}_e, \tilde{y}_t, \tilde{z}_t$ $[1,1,1,1]$ | 0.4258 | 0.1342 | 0.4300 | 0.4687 | 5.9237 |
| CLVs - variable 3% inflation | $\tilde{y}_e, \tilde{z}_e, \tilde{y}_t, \tilde{z}_t$ $[1,1,1,1]$ | 0.4337 | 0.1373 | 0.4027 | 0.4649 | 5.9191 |
| CLVs - 9 4% inflation | $\tilde{y}_t, \tilde{z}_t, \tilde{Y}, \tilde{Z}$ $[1,1,25,25]$ | 6.8613 | 0.1431 | 0.3025 | 4.5961 | 4.5309 |
| CLVs - variable 4% inflation | $\tilde{y}_t, \tilde{z}_t, \tilde{Y}, \tilde{Z}$ $[1,1,25,25]$ | 6.8722 | 0.1463 | 0.3370 | 4.6054 | 4.5541 |

Table 5: Summary metrics of DA experiments using right-transform matrix (ETKF) and shadowed trajectory as observations. We set the observation error covariances to the standard values as in [2]. Parameters: analysis window 0.08, 11 ensemble members, inflation as noted in table.

| Method | Observations [error variance] | $\langle\text{RMSE}\rangle$ extratropical | $\langle\text{RMSE}\rangle$ tropical | $\langle\text{RMSE}\rangle$ ocean | $\langle\text{RMSE}\rangle$ full | $\langle\dim_{KY}\rangle$ |
|---|---|---|---|---|---|---|
| CLVs - 9 | $x_e, y_e, z_e$ $[1,1,1]$ | 0.0640 | 8.4752 | 36.3662 | 21.7108 | 4.1332 |
| CLVs - 9 adaptive gain | $x_e, y_e, z_e$ $[1,1,1]$ | 0.0032 | 0.7241 | 3.5757 | 2.1504 | 5.9991 |

Table 6: Summary metrics of DA experiments using left-transform matrix (ESRF) and the full extratropical subsystem as observations ($x_e, y_e, z_e$). We use perfect observations (no random error added to the control run) with the observation error covariances set to the standard values as in [2]. Parameters: analysis window 0.02, inflation factor 1%, 10 ensemble members.

---

## Referee Comment (RC2) · Anonymous Referee #2 · 8 Dec 2019

This a very interesting paper. It is extremely well written, with very few typos and it is a pleasure to read it. Nonetheless, the study and the manuscript have in my opinion a few flaws that need amending.

1. Although the bibliography is rather dense (and better done than usual), there are still a few (very) relevant references such as Palatella and Trevisan (2015); Grudzien et al. (2018a,b) that are missing.

2. Even though the number of CLVs incorporated in the DA algorithm depends on time, you still need to compute a number of CLVs corresponding to the maximum

local KY, do you? So that the gain is computationally limited. If I am wrong, please explain.

3. In at least a few experiments, you need to use optimally tuned inflation (section 5.3 for instance), since it is already known that the lack of span of the unstable modes can be compensated with by a stronger multiplicative inflation. Otherwise several of your claims are undermined.

4. The new inflation scheme is not justified enough. Beware that it has been tested in a very specific case and does not warrant generality.

5. Because of the above points, there are too strong statements in the conclusion regarding the novelty and performance of the proposed method.

I believe that the paper only requires minor revisions before acceptance. But the above points should be seriously addressed and perhaps a few experiments re-run with multiple inflation values.

You will find below a list of minor suggestions or related to the points above, which could help improve the manuscript:

1. p.1, l.5: "to determine" is a bit ambiguous as it could also need "to infer" which would be a bold statement. I guess you meant "to prescribe", right?

2. p.2, l.2: "implying very large ensemble sizes are needed" $\longrightarrow$ "implying that very large ensemble sizes are needed"

3. p.3, l.6: What are "local CLVs"? I know local LEs, but not local CLVs.

4. p.3: Even though the beginning of the paper is very good and enjoyable, reading "We also examine the role of correlated versus random observational errors.",

"along with a novel scheme for adaptive Kalman gain inflation." at the end of the introduction is odd as these two subjects do not seem directly connected to the main objective of the paper, and they seem, at this stage of the reading, unnecessary.

5. p.4, l.18: "uncentering parameters": please explain what this means.

6. p.4, l.27: "We are interested in analysing both the local and global dynamics of system (1).": I guess you mean the short-term and asymptotics dynamics – your words lack accuracy here.

7. p.6, l.16: Can you be sure/prove that the last digits of $5.9473$ are relevant?

8. p.7, l.2: "approach their corresponding asymptotic values" $\longrightarrow$ "approach their corresponding LEs asymptotic values"

9. p.7, l.3: You have to discuss/justify more the concept of local KY dimension. This is the main idea of your paper.

10. p.7, l.4: "We see the local dimension" $\longrightarrow$ "We see that the local dimension"

11. p.7, l.13: "the cocycle of": Please explain what a cocycle is. How is it connected to (1)?

12. p.8, l.7: "local time-varying Kaplan-Yorke dimension.": what is its interpretation? This is key to your paper.

13. p.8, l.16-19: Palatella and Trevisan (2015) could be mentioned here.

14. p.9, l.4: "Suppose there exists" $\longrightarrow$ "Suppose that there exists"

15. p.9, Fig.5, legend: replace the "v" symbol by "or" for the sake of clarity.

16. p.9, l.9-11: The errors are also supposed uncorrelated in time (white).

17. p.9, Eqs.(8,9): Missing punctuation.

18. p.10, l.5: "There is difficulty" $\longrightarrow$ "There is a difficulty"

19. p.10, l.7: "assumption of linearity": this is confusing here since you have not introduced the extended Kalman filter yet.

20. p.10, l.21: Even though quoting Bishop et al. (2001) is certainly adequate, a reference to Hunt et al. (2007) is also missing as it is equally relevant.

21. p.10, Eq.(14) is wrong, is it? It should be

$$\mathbf{E} = \mathbf{R}^{-1/2}\mathbf{H}\mathbf{X}^{\mathrm{f}}. \tag{1}$$

Also it is not recommended to use $\mathbf{E}$ as it is usually used for the full ensemble matrix. Authors often use $\mathbf{S}$ instead.

22. p.11, l.7: "The Kalman gain K is defined through equation (9a)": No. Not in the classical ETKF (see Hunt et al. (2007)).

23. p.11, l.15-20: In this context, sampling errors are actually due to nonlinearity, as it was explained and proven by Bocquet et al. (2015); Raanes et al. (2019).

24. p.11, l.25-27: "This differs to past approaches where the subspace was determined in terms of the long time averaged (invariant) unstable and neutral CLVs (Trevisan and Uboldi, 2004; Carrassi et al., 2008; Trevisan and Palatella, 2011).": This statement is misleading. You just mean that the number of retained CLVs is kept fixed. Did you?

25. p.12, l.7-8: "We compute CLVs at the assimilation step using Algorithm 1.": The number of CLVs is fixed over the full time span of the algorithm, is it?

26. p.12, l.19: "regardless of observation set" $\longrightarrow$ "regardless of the observation set"

27. p.12, l.23: The term "Analysis window" is unfortunate as it usually refers to the time range over which asynchronous observations are assimilated in 4D-Var or with an ensemble smoother. I guess you mean the time interval between updates. You could denote it $\Delta t$ for instance. Please change it throughout the manuscript.

28. p.13, section 5.1: Please explain better what changing the observation set has to do with the main goal of the manuscript.

29. p.13, l.9-10: "We argue here that in reality, the true variance of the observation error can be spatially dependent and errors are often correlated in time.": this is a bit too much, since there are quite a few DA papers dealing with at least spatially correlated errors.

30. p.13, l.27: "we decrease the analysis window and do not perturb the control run at all when taking the observations.": You mean that the synthetic observations are not perturbed, do you? The sentence seems a bit twisted.

31. p.15, l.15: "Finally we analyse our novel reduced subspace method which uses a variable number of CLVs based on the local Kaplan-Yorke dimension.": yes, but I guess you need to compute a number of CLVs corresponding to the maximum local KY dimension, so that even though it is theoretically interesting, it is, in practice, of limited interest.

32. p.17, Fig. 7 (a-c): please plot over a smaller range, typically [500-600] as in Fig. 16.

33. p.18, Table 3, and discussion around: This experiment does not account for what is actually known in the literature. You should have made an experiment with the 6 CLVs but with optimally tuned inflation, or you could have used the finite-size EnKF (Bocquet et al., 2015, and references therein). It is by now well known that the gap between the second and the third experiment might be compensated by optimally tuned inflation.

34. p.19, l.16-18: "but we are interested if we can preserve": I don't really understand the phrase.

35. p.21, l.1: "in ability" $\longrightarrow$ "in the ability"

36. p.25, l.5-10: In this paragraph, you argue but you don't give a strong rationale for the inflation scheme you propose. You need a stronger case to convince the reader. All the more since the inflation scheme is tested with a toy model in a very specific configuration.

37. p.25, Eq.(25): The modified $\mathbf{P}^f$ does no have the good engineering dimensional (cube in the anomalies instead of square). What do you make of this?

38. p.25, l.23-24: It seems like the $\beta$-factor approach is implementing deflation. Is it so? If yes, please use the term deflation instead of inflation, which is customary.

39. p.27, l.1: "there is remarkable" $\longrightarrow$ "there is a remarkable"

40. p.27, l.7-8: "We have demonstrated the varying rank of the error covariance matrix related to the transient growth in the stable modes of the system.": true but this was already emphasised in the literature, so that your implicit statement of novelty should be tuned down here.

41. p.27, l.16-17: "to determine the rank": unclear and confusing; I believe you mean "to specify the appropriate rank"; you don't discover the rank, you set it.

42. p.27, l.22-24: "In particular, we found that spanning the space comprised of the asymptotic unstable, neutral, and first weakly stable mode (5 CLVs in this case) performed much worse than using either dimension measure (asymptotic and local).": I don't think so. This is one weak point of your study. It is known that in this case, the inflation must be adjusted to account for the error upscaling from the the region of the spectrum (Grudzien et al., 2018a,b). I don't believe that

you have tuned the inflation, have you? If true, your statement appears to be too strong.

43. p.28, l.19-20: "The adaptive scaling introduced here can be applied to general systems with weak coupling, although care may need to be taken in the choice of the norm.": No, you have not proven anything like that. Please remove the statement.

44. p.28, l.33-34: "Future work should also consider the numerical cost of CLV calculation and methods to increase efficiency for high dimensional systems": At the very end, you raise what experts familiar with AUS have in mind reading your paper: what you propose is certainly interesting and of theoretical interest but of lesser practical value since (i) one needs to compute the CLVs alongside (ii) with the variable CLV context, you need to compute the maximum number of CLVs. You should mentioned this point way earlier in the paper, unless I am mistaken.

**References**

Bishop, C.H., Etherton, B.J., Majumdar, S.J., 2001. Adaptive sampling with the ensemble transform Kalman filter. Part I: Theoretical aspects. Mon. Wea. Rev. 129, 420–436. doi:`10.1175/1520-0493(2001)129<0420:ASWTET>2.0.CO;2`.

Bocquet, M., Raanes, P.N., Hannart, A., 2015. Expanding the validity of the ensemble Kalman filter without the intrinsic need for inflation. Nonlin. Processes Geophys. 22, 645–662. doi:`10.5194/npg-22-645-2015`.

Grudzien, C., Carrassi, A., Bocquet, M., 2018a. Asymptotic forecast uncertainty and the unstable subspace in the presence of additive model error. SIAM/ASA J. Uncertainty Quantification 6, 1335–1363. doi:`10.1137/17M114073X`.

Grudzien, C., Carrassi, A., Bocquet, M., 2018b. Chaotic dynamics and the role of covariance inflation for reduced rank Kalman filters with model error. Nonlin. Processes Geophys. 25, 633–648. doi:`10.5194/npg-25-633-2018`.

Hunt, B.R., Kostelich, E.J., Szunyogh, I., 2007. Efficient data assimilation for spatiotemporal chaos: A local ensemble transform Kalman filter. Physica D 230, 112–126. doi:`10.1016/j.physd.2006.11.008`.

Palatella, L., Trevisan, A., 2015. Interaction of Lyapunov vectors in the formulation of the nonlinear extension of the Kalman filter. Phys. Rev. E 91, 042905. doi:`10.1103/PhysRevE.91.042905`.

Raanes, P.N., Bocquet, M., Carrassi, A., 2019. Adaptive covariance inflation in the ensemble Kalman filter by Gaussian scale mixtures. Q. J. R. Meteorol. Soc. 145, 53–75. doi:`10.1002/qj.3386`.

---

## Author Comment (AC2) · 17 Dec 2019

**Response to anonymous reviewer comments**

Dear Reviewer,

We first thank you for your general positive evaluation of the manuscript. We agree that the points of concern that you raised add valuable discussion to the work. Below we address your comments in detail. We also note where we will make adjustments to the manuscript accordingly.

Many thanks,

Courtney Quinn, Terence O'Kane, and Vassili Kitsios
* * *
**Main points:**

1. Although the bibliography is rather dense (and better done than usual), there are still a few (very) relevant references such as Palatella and Trevisan (2015) [9]; Grudzien et al. (2018a,b) [4, 5] that are missing.

   We thank the reviewer for pointing out the additional relevant references and we will include them in accordance with the specific suggestions below.

2. Even though the number of CLVs incorporated in the DA algorithm depends on time, you still need to compute a number of CLVs corresponding to the maximum local KY, do you? So that the gain is computationally limited. If I am wrong, please explain.

   In the way that we have implemented the variable dimension, one could compute the local KY measure first and then only compute the number of CLVs corresponding to that given value at each assimilation time step. (This is due to the fact that we compute dimension with the QR-factorization approach and the CLVs with a separate algorithm - clarification of this and further discussion has been added to the modified manuscript.) With respect to computational gain, there will be an additional cost associated with computing the unstable manifold regardless of the method. Whether or not this is realizable in high dimensional systems, or how the overall cost compares to increasing ensemble size in order to fully sample the variance are not questions we attempt to answer in this study. We rather focused on the applicability of assimilation in a variable unstable subspace for different situations of strongly coupled data assimilation (*i.e.* different types of observational subsets). We discuss the reviewer's comments regarding computational cost more within the specific points below.

3. In at least a few experiments, you need to use optimally tuned inflation (section 5.3 for instance), since it is already known that the lack of span of the unstable modes can be compensated with by a stronger multiplicative inflation. Otherwise several of your claims are undermined.

   We first preface our response with bringing to the reviewer's attention that there was a small error in the calculation of the CLVs for the results shown in the manuscript submission. We have since corrected this error and updated all of our experiments. The updated tables are included at the end of our responses below. Due to these corrected results, we now can conclude for the benchmark and atmosphere observations that all experiments are successful. The emphasis of the discussion is now regarding the variable CLV full RMSE being closest to the full rank experiments for each of the observation sets. These conclusions hold for a range of inflation values. We show in Table 1 below the results for the benchmark observations using full rank, 5 CLVs, and variable CLVs with inflation values of 1% (as in manuscript - revised values), 2%, 5% and 10%. As the lowest RMSEs occur for the 1% inflation cases, we conclude this is the optimal inflation value and leave these results in the manuscript. We have added a statement noting this to the beginning of the results section (Section 5 in the manuscript). The optimization was already done for the shadowed observations (Section 5.5), and the optimal inflation values are given in the table. All other experiments have an optimal inflation of 1%.

4. The new inflation scheme is not justified enough. Beware that it has been tested in a very specific case and does not warrant generality.

   We agree with the reviewer that our language when discussing the utility of the adaptive scaling for the Kalman gain is too strong. We have changed the language to emphasize that the adaptive scaling works in this particular case, but needs to be tested on more models to form a a more general applicability. We also have added additional justification for the proposed scheme and relate it to the work of Miller *et al* [7]. More detailed discussion of our justification can be found in response to the specific points of the following section.

5. Because of the above points, there are too strong statements in the conclusion regarding the novelty and performance of the proposed method.

   Again, we agree with the reviewer that too strong of language is used in regards to some of the conclusions. We note that due to our corrected results we have already adjusted some of our conclusions accordingly. We will additionally adjust the specific statements mentioned in the following section.

| Method | Observations [error variance] | ⟨RMSE⟩ extratropical | ⟨RMSE⟩ tropical | ⟨RMSE⟩ ocean | ⟨RMSE⟩ full | ⟨dim$_{KY}$⟩ |
|---|---|---|---|---|---|---|
| CLVs - 9 1% inflation | $y_e, y_t, Y$ [1, 1, 25] | 0.3142 | 0.1598 | 0.4948 | 0.4027 | 5.8928 |
| CLVs - 5 1% inflation | $y_e, y_t, Y$ [1, 1, 25] | 0.3123 | 0.1843 | 0.5920 | 0.4550 | 5.8870 |
| CLVs - variable 1% inflation | $y_e, y_t, Y$ [1, 1, 25] | 0.3215 | 0.1688 | 0.5346 | 0.4272 | 5.8863 |
| CLVs - 9 2% inflation | $y_e, y_t, Y$ [1, 1, 25] | 0.3286 | 0.1893 | 0.5985 | 0.4625 | 5.8881 |
| CLVs - 5 2% inflation | $y_e, y_t, Y$ [1, 1, 25] | 0.3274 | 0.1950 | 0.6083 | 0.4681 | 5.8871 |
| CLVs - variable 2% inflation | $y_e, y_t, Y$ [1, 1, 25] | 0.3219 | 0.1916 | 0.5966 | 0.4594 | 5.8915 |
| CLVs - 9 5% inflation | $y_e, y_t, Y$ [1, 1, 25] | 0.3564 | 0.2627 | 0.8858 | 0.6237 | 5.8832 |
| CLVs - 5 5% inflation | $y_e, y_t, Y$ [1, 1, 25] | 0.3600 | 0.2665 | 0.8880 | 0.6264 | 5.8836 |
| CLVs - variable 5% inflation | $y_e, y_t, Y$ [1, 1, 25] | 0.3552 | 0.2647 | 0.8747 | 0.6180 | 5.8848 |
| CLVs - 9 10% inflation | $y_e, y_t, Y$ [1, 1, 25] | 0.4117 | 0.3719 | 1.3278 | 0.8799 | 5.8763 |
| CLVs - 5 10% inflation | $y_e, y_t, Y$ [1, 1, 25] | 0.4187 | 0.3873 | 1.3168 | 0.8787 | 5.8824 |
| CLVs - variable 10% inflation | $y_e, y_t, Y$ [1, 1, 25] | 0.4123 | 0.3731 | 1.3153 | 0.8736 | 5.8775 |

Table 1: Summary metrics of DA experiments using right-transform matrix (ETKF) and benchmark observations $(y_e, y_t, Y)$. The angle brackets ⟨·⟩ denote average over analysis steps. Compare to results in [12]. Parameters: analysis window 0.08, 10 ensemble members.

**Specific suggestions:**

1. p.1, l.5: "to determine" is a bit ambiguous as it could also need "to infer" which would be a bold statement. I guess you meant "to prescribe", right?

   We have changed the statement to read "to prescribe" instead, as this was our intended connotation.

2. p.2, l.2: "implying very large ensemble sizes are needed" → "implying that very large ensemble sizes are needed

   We have amended the sentence.

3. p.3, l.6: What are "local CLVs"? I know local LEs, but not local CLVs.

This was a typo. We were referring to the CLVs calculated at a given time. We have removed the term "local".

4. p.3: Even though the beginning of the paper is very good and enjoyable, reading "We also examine the role of correlated versus random observational errors.", "along with a novel scheme for adaptive Kalman gain inflation." at the end of the introduction is odd as these two subjects do not seem directly connected to the main objective of the paper, and they seem, at this stage of the reading, unnecessary.

We can agree that the statements are superfluous. We will save the discussion of these two topics for when they arise in the study.

5. p.4, l.18: "uncentering parameters": please explain what this means.

The terminology "uncentering" was originally used in the Peña and Kalnay study [10] and refers to the uncentering of the unstable zero equilibrium in the classical Lorenz model. Dynamically, the $k_1$ and $k_2$ values used in the manuscript allow for the second unstable asymptotic Lyapunov exponent (for $k_1 = k_2 = 0$: $\lambda_1 = 0.91, \lambda_2 = 0, \lambda_i < 0$ for $i > 2$).

6. p.4, l.27: "We are interested in analysing both the local and global dynamics of system (1).": I guess you mean the short-term and asymptotics dynamics - your words lack accuracy here.

That is precisely what we meant. We will change the language as suggested.

7. p.6, l.16: Can you be sure/prove that the last digits of 5.9473 are relevant?

We do not claim that the last digits are relevant, they are included only as the Lyapunov exponents were given up to four digits.

8. p.7, l.2: "approach their corresponding asymptotic values" → "approach their corresponding LEs asymptotic values"

We have changed the sentence to read "approach the asymptotic Lyapunov exponent values".

9. p.7, l.3: You have to discuss/justify more the concept of local KY dimension. This is the main idea of your paper.

We have added some additional discussion around the concept of local dimension used in our manuscript as suggested.

10. p.7, l.4: "We see the local dimension"→ "We see that the local dimension"

We have amended the sentence.

11. p.7, l.13: "the cocycle of": Please explain what a cocycle is. How is it connected to (1)?

   In relation to system (1) in the manuscript, a cocycle is the forward and backward mapping of solutions under the tangent dynamics. The cocycle is then given as $\mathcal{A}(x(t), \tau) = e^{\tau J f(x(t))}$ where $Jf$ denotes the Jacobian of $f$, the right-hand-side of system (1). We have clarified this in the manuscript and have restructured the sentence such that *a prior* knowledge of the term cocycle is not assumed.

12. p.8, l.7: "local time-varying Kaplan-Yorke dimension.": what is its interpretation? This is key to your paper.

   We have added a more descriptive explanation of the local Kaplan-Yorke dimension here, particularly relating to our implementation and its relation to the CLV behaviour.

13. p.8, l.16-19: Palatella and Trevisan (2015) [9] could be mentioned here.

   We have added the reference.

14. p.9, l.4: "Suppose there exists"→ "Suppose that there exists"

   We have amended the sentence.

15. p.9, Fig.5, legend: replace the "v" symbol by "or" for the sake of clarity.

   We have changed the legend accordingly.

16. p.9, l.9-11: The errors are also supposed uncorrelated in time (white).

   We have clarified the assumption of temporally uncorrelated errors in the description of the Kalman filter method.

17. p.9, Eqs.(8,9): Missing punctuation.

   We have added punctuation to the noted equations.

18. p.10, l.5: "There is difficulty"→"There is a difficulty"

   We have amended the sentence.

19. p.10, l.7: "assumption of linearity": this is confusing here since you have not introduced the extended Kalman filter yet.

   We have removed the statement in which this phrase appears as it is not critical to the reader.

20. p.10, l.21: Even though quoting Bishop et al. (2001) [1] is certainly adequate, a reference to Hunt et al. (2007) [6] is also missing as it is equally relevant.

   We have added the reference as suggested.

21. p.10, Eq.(14) is wrong, is it? It should be

$$\mathbf{E} = \mathbf{R}^{-1/2}\mathbf{H}\mathbf{X}^f. \tag{1}$$

   Also it is not recommended to use $\mathbf{E}$ as it is usually used for the full ensemble matrix. Authors often use $\mathbf{S}$ instead.

   We thank the reviewer for catching this typo. We did in fact mean to define the matrix as above, as the left multiplication with its transpose is included in Eq (15). We have also changed the notation from $\mathbf{E}$ to $\mathbf{S}$ as suggested.

22. p.11, l.7: "The Kalman gain K is defined through equation (9a)": No. Not in the classical ETKF (see Hunt et al. (2007) [6]).

   We have changed the text to specify that we are following the definition of the Kalman filter as given in Bishop et al [1].

23. p.11, l.15-20: In this context, sampling errors are actually due to nonlinearity, as it was explained and proven by Bocquet et al. (2015) [2]; Raanes et al. (2019) [11].

   We have included the additional references here where we discuss the sampling errors due to nonlinearity.

24. p.11, l.25-27: "This differs to past approaches where the subspace was determined in terms of the long time averaged (invariant) unstable and neutral CLVs (Trevisan and Uboldi, 2004; Carrassi et al., 2008; Trevisan and Palatella, 2011).": This statement is misleading. You just mean that the number of retained CLVs is kept fixed. Did you?

   This is badly phrased. We meant "asymptotic Lyapunov exponents", and therefore the rank of the error covariance matrix is kept fixed. We have updated the manuscript accordingly.

25. p.12, l.7-8: "We compute CLVs at the assimilation step using Algorithm 1.": The number of CLVs is fixed over the full time span of the algorithm, is it?

   We always compute all 9 CLVs in this case, and then only use the number corresponding to the experiment set-up in constructing the covariance matrix. In this toy model we were not as concerned with optimizing computation time. However, it is easily implemented to only compute a subset of

the CLVs either specified as constant or variable based on the calculated dimension for the assimilation step. (Note, the dimension is calculated from the FTLEs computed using the QR method so this is a separate process to the CLV calculation.) We discuss more about computation time in response to the final comment (44) below.

26. p.12, l.19: "regardless of observation set" → "regardless of the observation set"

    We have amended the sentence.

27. p.12, l.23: The term "Analysis window" is unfortunate as it usually refers to the time range over which asynchronous observations are assimilated in 4D-Var or with an ensemble smoother. I guess you mean the time interval between updates. You could denote it $\Delta t$ for instance. Please change it throughout the manuscript.

    We have changed "analysis window" to "assimilation window" throughout the manuscript ($\Delta t$ conflicts with our integration time step notation).

28. p.13, section 5.1: Please explain better what changing the observation set has to do with the main goal of the manuscript.

    The different observation sets relate to the exploration of strongly coupled data assimilation (strong CDA), namely the use of cross-covariances to update unobserved states. We were interested in whether the use of the reduced space method (either fixed or variable) is applicable in different observation scenarios, including the cases where some subsystems are left unobserved or where there are temporally correlated errors. We have emphasized this in the introduction to the results section (Section 5) and in Section 5.1.

29. p.13, l.9-10: "We argue here that in reality, the true variance of the observation error can be spatially dependent and errors are often correlated in time.": this is a bit too much, since there are quite a few DA papers dealing with at least spatially correlated errors.

    We have revised the tone of the sentence which now reads: "In many applications the true variance ..."

30. p.13, l.27: "we decrease the analysis window and do not perturb the control run at all when taking the observations.": You mean that the synthetic observations are not perturbed, do you? The sentence seems a bit twisted.

    We have amended the sentence to read: "we decrease the assimilation window and assume perfect observations."

31. p.15, l.15: "Finally we analyse our novel reduced subspace method which uses a variable number of CLVs based on the local Kaplan-Yorke dimension.": yes, but I guess you need to compute a number of CLVs corresponding to the maximum local KY dimension, so that even though it is theoretically interesting, it is, in practice, of limited interest.

    We take this as a comment. To clarify, as mentioned in our response to point 25, one only needs to compute the number of CLVs corresponding to the local Kaplan-Yorke dimension for a given assimilation step. In our experiments this varies between 0 and 8 for different assimilation steps. We discuss practical implementation in response to the reviewer's final point below.

32. p.17, Fig. 7 (a-c): please plot over a smaller range, typically [500-600] as in Fig.16.

    We have changed the range of both figures (16 and 17 in previous manuscript) to [450-550] in accordance with the discussion around the dynamical properties.

33. p.18, Table 3, and discussion around: This experiment does not account for what is actually known in the literature. You should have made an experiment with the 6 CLVs but with optimally tuned inflation, or you could have used the finite-size EnKF (Bocquet et al., 2015, and references therein). It is by now well known that the gap between the second and the third experiment might be compensated by optimally tuned inflation.

    This point was addressed in the main comments section above. With our correction to the CLV calculation, we no longer have such a discrepancy between experiments. The corrected table can be found in the following section. We have additionally done an optimization on inflation and find that the 1% is in fact optimal for these experiments.

34. p.19, l.16-18: "but we are interested if we can preserve": I don't really understand the phrase.

    We have changed the sentence to read: "we are interested to see if we can avoid collapse (*i.e.* the loss of variability) in the ensemble mean of the extratropical attractor".

35. p.21, l.1: "in ability" → "in the ability"

    We have amended the sentence.

36. p.25, l.5-10: In this paragraph, you argue but you don't give a strong rationale for the inflation scheme you propose. You need a stronger case to convince the reader. All the more since the inflation scheme is tested with a toy model in a very specific configuration.

We have added further discussion of the Miller *et al* [7] study to motivate our proposed modified Kalman gain. Through the study of data assimilation schemes on the Lorenz '63 model, they find that the forecast error covariance is often underestimated in highly nonlinear systems, particularly when the model is in a region of phase space subject to transitions. This leads to the Kalman gain being underestimated. The authors account for this by including the third and fourth moments of anomalies within the Kalman gain calculation, however this is done for the Extended Kalman Filter (EKF) method (see eq. 4.3 of [7]). O'Kane and Frederiksen [8] also derive a higher order gain for a closure-based statistical dynamical Kalman filter applied in spectral space (see eq. 27 of [8]). Here we use the notion that increased spread in a subsystem represents the inability of ensemble members to track the same transitions in phase space. In such a situation the forecast error covariance is likely to be underestimated (presumably due to emerging importance of higher moments), therefore we scale the forecast error covariance within the Kalman gain calculation by a factor that represents a measure of overall spread in the system. This then provides an adaptive scaling of the Kalman gain at every assimilation time step based on the background performance of the system (large spread implies an increase in Kalman gain, small spread implies a decrease). The scaling can equivalently be written as an observation error variance scaling, and its overall behaviour is a balancing between the forecast error covariance and the observation error variance within the calculation of the Kalman gain. We have expanded upon this discussion within the manuscript.

37. p.25, Eq.(25): The modified $\mathbf{P}^f$ does no have the good engineering dimensional (cube in the anomalies instead of square). What do you make of this?

    Our introduction of the scaled Kalman gain was written poorly here. We actually do not modify the forecast error covariance matrix as worded in the original manuscript. We add an additional scaling term to the forecast error covariance matrix only within the calculation of the Kalman gain. The scaling factor ($||\mathbf{P}^f||$) itself is a measure of the anomalies squared, therefore making the quantity used in the Kalman gain a quartic measure of anomalies. In a crude way one could consider this an incorporation of higher moments into the Kalman gain, however since this is not a true estimate of the fourth moment, we used the spread-based argument for its motivation (as given above).

38. p.25, l.23-24: It seems like the $\beta$-factor approach is implementing deflation. Is it so? If yes, please use the term deflation instead of inflation, which is customary.

We have changed the language to refer to the $\beta$-factor approach as deflation.

39. p.27, l.1: "there is remarkable"→"there is a remarkable"

    We amended the sentence.

40. p.27, l.7-8: "We have demonstrated the varying rank of the error covariance matrix related to the transient growth in the stable modes of the system.": true but this was already emphasised in the literature, so that your implicit statement of novelty should be tuned down here.

    We agree with the reviewer that our statement was too strongly worded. We have amended the sentence to read: "We have explored the varying rank of the error covariance matrix related to the transient growth in the stable modes of the system, and in particular the applicability of this varying rank on different configurations of strong CDA."

41. p.27, l.16-17: "to determine the rank": unclear and confusing; I believe you mean "to specify the appropriate rank"; you don't discover the rank, you set it.

    That is correct, we meant the connotation "to specify". We have revised the sentence accordingly.

42. p.27, l.22-24: "In particular, we found that spanning the space comprised of the asymptotic unstable, neutral, and first weakly stable mode (5 CLVs in this case) performed much worse than using either dimension measure (asymptotic and local).": I don't think so. This is one weak point of your study. It is known that in this case, the inflation must be adjusted to account for the error upscaling from the the region of the spectrum (Grudzien et al., 2018a,b). I don't believe that you have tuned the inflation, have you? If true, your statement appears to be too strong.

    We have toned down the language around the discussion of these results after the correction to the calculation. We no longer claim the 5 CLV case performs "much" worse, as all methods now perform comparably. We point out, however, that the 6 CLV and variable CLV both show a reduction in the full RMSE which can be related to significant linear growth in more than one stable mode. We also have included the reference [5] within our discussion of inflation in Section 4.1, and the reference [4] in discussing the impact of transient growth of asymptotically stable modes on model errors in the introduction.

43. p.28, l.19-20: "The adaptive scaling introduced here can be applied to general systems with weak coupling, although care may need to be taken in the choice of the norm.": No, you have not proven anything like that. Please remove the statement.

Instead of removing, we have amended the statement to one which reflects our original sentiment. We wanted to suggest that the method be tested on other systems of weak coupling and then perhaps a more general statement can be made under deeper analysis. We have changed the statement to read "The adaptive scaling introduced here should be tested on additional systems with weak coupling in order to assess its general applicability, although care may need to be taken in the choice of the norm."

44. p.28, l.33-34: "Future work should also consider the numerical cost of CLV calculation and methods to increase efficiency for high dimensional systems": At the very end, you raise what experts familiar with AUS have in mind reading your paper: what you propose is certainly interesting and of theoretical interest but of lesser practical value since (i) one needs to compute the CLVs alongside (ii) with the variable CLV context, you need to compute the maximum number of CLVs. You should mentioned this point way earlier in the paper, unless I am mistaken.

The reviewer is correct in that we do not address computational cost throughout the paper. This was mainly intentional, as we did not aim to optimize computational performance, but rather explore a method that allows for the reduction of phase space and/or better representation of errors in that reduced subspace. Whether or not there can be a computational cost reduction by using this method over spanning the space needed to fully sample the variance is a question we do not attempt to answer as it will depend on the system one wishes to assimilate. In accordance with the reviewer's request, we have added a footnote within the introduction at the point where we discuss assimilation in the unstable subspace (AUS). The footnote acknowledges the additional cost that comes with computing the unstable subspace which may or may not be less than the cost of sampling the full model variance, and it also states that we will leave the exploration of numerical efficiency in high dimensional systems for future study.

We additionally note that the projection onto CLVs is an example of one specific projection that works for the variable local dimension, but it is not the only one. One could also project onto the backwards Lyapunov vectors (BLVs) which are computationally less expensive to compute and attain similar results. Table 2 below includes the results of variable dimension and projection onto BLVs. We have added a discussion of this alternate projection into the manuscript, as well as the additional statistics of the variable BLV experiment. However, we focus on the CLVs in this study as the *a posteriori* analysis of CLV alignment can provide more information of the time-varying phase space behaviour. (The BLVs are orthogonal by construction.) The revised manuscript has an additional discussion of CLV alignment, variable dimension, and specifically the relationship to

ensemble data assimilation which justifies the utility of CLVs.

We imagine that it is possible to find a computationally efficient projection for high dimensional systems that incorporates the variable dimension aspect. Our main goal in this manuscript though was to introduce the implementation of time-varying dimension in this toy model with different observational sets to understand in what configurations one might expect successful results. It would be of interest for practical applications if future studies could explore numerically efficient projections that span the time-varying dimension and therefore utilise the ideas of AUS in high dimensional systems.
* * *
**Minor corrections to DA experiments**

After submission we discovered a small error in the orthogonalisation step of the CLV calculation within the DA experiments. In correcting we found that using the CLV without orthogonalisation to the backwards subspaces (Algorithm 2.1 in [3]) was more appropriate due to the lack of an accurate forward model within the DA implementation. This has minor impact on the results of the experiments, however we include the tables with the updated statistics below which we will also update in the new manuscript.

| Method | Observations [error variance] | $\langle$RMSE$\rangle$ extratropical | $\langle$RMSE$\rangle$ tropical | $\langle$RMSE$\rangle$ ocean | $\langle$RMSE$\rangle$ full | $\langle$dim$_{KY}\rangle$ |
|---|---|---|---|---|---|---|
| CLVs - 9 (full rank) | $y_e, y_t, Y$ $[1, 1, 25]$ | 0.3142 | 0.1598 | 0.4948 | 0.4027 | 5.8928 |
| CLVs - 5 (unstable/neutral subspace + 1) | $y_e, y_t, Y$ $[1, 1, 25]$ | 0.3123 | 0.1843 | 0.5920 | 0.4550 | 5.8870 |
| CLVs - 6 (global dimension) | $y_e, y_t, Y$ $[1, 1, 25]$ | 0.3156 | 0.1674 | 0.5513 | 0.4310 | 5.8892 |
| CLVs - variable (local dimension) | $y_e, y_t, Y$ $[1, 1, 25]$ | 0.3215 | 0.1688 | 0.5346 | 0.4272 | 5.8863 |
| BLVs - variable (local dimension) | $y_e, y_t, Y$ $[1, 1, 25]$ | 0.3149 | 0.1658 | 0.5122 | 0.4141 | 5.8895 |

Table 2: Summary metrics of DA experiments using right-transform matrix (ETKF) and benchmark observations $(y_e, y_t, Y)$. The angle brackets $\langle \cdot \rangle$ denote average over analysis steps. Compare to results in [12]. Parameters: assimilation window 0.08, inflation factor 1%, 10 ensemble members.

| Method | Observations [error variance] | $\langle$RMSE$\rangle$ extratropical | $\langle$RMSE$\rangle$ tropical | $\langle$RMSE$\rangle$ ocean | $\langle$RMSE$\rangle$ full | $\langle$dim$_{KY}\rangle$ |
|---|---|---|---|---|---|---|
| CLVs - 9 | $y_e, z_e, y_t, z_t$ $[1,1,1,1]$ | 0.1734 | 0.1332 | 0.5782 | 0.4515 | 5.8672 |
| CLVs - 6 | $y_e, z_e, y_t, z_t$ $[1,1,1,1]$ | 0.1715 | 0.1386 | 0.5807 | 0.4524 | 5.8718 |
| CLVs - variable | $y_e, z_e, y_t, z_t$ $[1,1,1,1]$ | 0.1697 | 0.1383 | 0.5659 | 0.4433 | 5.8681 |

Table 3: Summary metrics of DA experiments using right-transform matrix (ETKF) and atmosphere observations ($y_e, z_e, y_t, z_t$). Parameters: assimilation window 0.08, inflation factor 1%, 10 ensemble members.

| Method | Observations [error variance] | $\langle$RMSE$\rangle$ extratropical | $\langle$RMSE$\rangle$ tropical | $\langle$RMSE$\rangle$ ocean | $\langle$RMSE$\rangle$ full | $\langle$dim$_{KY}\rangle$ |
|---|---|---|---|---|---|---|
| CLVs - 9 | $y_t, z_t, Y, Z$ $[1,1,25,25]$ | 7.0467 | 0.1571 | 0.3675 | 4.7182 | 4.5449 |
| CLVs - 9 | $y_t, z_t, Y, Z$ $[0.01, 0.01, 25, 25]$ | 5.5203 | 0.0480 | 0.1764 | 3.6860 | 5.4052 |
| CLVs - 9 | $y_t, z_t, Y, Z$ $[1, 1, 0.25, 0.25]$ | 7.1308 | 0.1386 | 0.0873 | 4.7566 | 4.4959 |
| CLVs - 9 | $y_t, z_t, Y, Z$ $[0.01, 0.01, 0.25, 0.25]$ | 5.1697 | 0.0431 | 0.0822 | 3.42478 | 5.4899 |

Table 4: Summary metrics of DA experiments using right-transform matrix (ETKF) and ENSO observations ($y_t, z_t, Y, Z$). Parameters: assimilation window 0.08, inflation factor 1%, 10 ensemble members.

**References**

[1] Craig H Bishop, Brian J Etherton, and Sharanya J Majumdar. Adaptive sampling with the ensemble transform kalman filter. part i: Theoretical aspects. *Monthly weather review*, 129(3):420–436, 2001.

[2] Marc Bocquet, Patrick N Raanes, and Alexis Hannart. Expanding the validity of the ensemble kalman filter without the intrinsic need for inflation. *Nonlinear Processes in Geophysics*, 22(6):645–662, 2015.

[3] Gary Froyland, Thorsten Hüls, Gary P Morriss, and Thomas M Watson. Computing covariant lyapunov vectors, oseledets vectors, and dichotomy projectors: A comparative numerical study. *Physica D: Nonlinear Phenomena*, 247(1):18–39, 2013.

[4] Colin Grudzien, Alberto Carrassi, and Marc Bocquet. Asymptotic forecast uncertainty and the unstable subspace in the presence of additive model

| Method | Observations [error variance] | $\langle\text{RMSE}\rangle$ extratropical | $\langle\text{RMSE}\rangle$ tropical | $\langle\text{RMSE}\rangle$ ocean | $\langle\text{RMSE}\rangle$ full | $\langle\dim_{KY}\rangle$ |
|---|---|---|---|---|---|---|
| CLVs - 9 3% inflation | $\tilde{y}_e, \tilde{y}_t, \tilde{Y}$ $[1, 1, 25]$ | 0.5200 | 0.1635 | 0.4255 | 0.4552 | 5.9576 |
| CLVs - variable 3% inflation | $\tilde{y}_e, \tilde{y}_t, \tilde{Y}$ $[1, 1, 25]$ | 0.4848 | 0.1568 | 0.4095 | 0.4263 | 5.9371 |
| CLVs - 9 3% inflation | $\tilde{y}_e, \tilde{z}_e, \tilde{y}_t, \tilde{z}_t$ $[1, 1, 1, 1]$ | 0.4258 | 0.1342 | 0.4300 | 0.4687 | 5.9237 |
| CLVs - variable 3% inflation | $\tilde{y}_e, \tilde{z}_e, \tilde{y}_t, \tilde{z}_t$ $[1, 1, 1, 1]$ | 0.4337 | 0.1373 | 0.4027 | 0.4649 | 5.9191 |
| CLVs - 9 4% inflation | $\tilde{y}_t, \tilde{z}_t, \tilde{Y}, \tilde{Z}$ $[1, 1, 25, 25]$ | 6.8613 | 0.1431 | 0.3025 | 4.5961 | 4.5309 |
| CLVs - variable 4% inflation | $\tilde{y}_t, \tilde{z}_t, \tilde{Y}, \tilde{Z}$ $[1, 1, 25, 25]$ | 6.8722 | 0.1463 | 0.3370 | 4.6054 | 4.5541 |

Table 5: Summary metrics of DA experiments using right-transform matrix (ETKF) and shadowed trajectory as observations. We set the observation error covariances to the standard values as in [12]. Parameters: assimilation window 0.08, 11 ensemble members, inflation as noted in table.

error. *SIAM/ASA Journal on Uncertainty Quantification*, 6(4):1335–1363, 2018.

[5] Colin Grudzien, Alberto Carrassi, and Marc Bocquet. Chaotic dynamics and the role of covariance inflation for reduced rank kalman filters with model error. *Nonlinear Processes in Geophysics*, 25(3):633–648, 2018.

[6] Brian R Hunt, Eric J Kostelich, and Istvan Szunyogh. Efficient data assimilation for spatiotemporal chaos: A local ensemble transform kalman filter. *Physica D: Nonlinear Phenomena*, 230(1-2):112–126, 2007.

[7] Robert N Miller, Michael Ghil, and Francois Gauthiez. Advanced data assimilation in strongly nonlinear dynamical systems. *Journal of the atmospheric sciences*, 51(8):1037–1056, 1994.

[8] Terence O'Kane and Jorgen Frederiksen. Comparison of statistical dynamical, square root and ensemble kalman filters. *Entropy*, 10(4):684–721, 2008.

[9] Luigi Palatella and Anna Trevisan. Interaction of lyapunov vectors in the formulation of the nonlinear extension of the kalman filter. *Physical Review E*, 91(4):042905, 2015.

[10] M Peña and E Kalnay. Separating fast and slow modes in coupled chaotic systems. *Nonlinear Processes in Geophysics*, 11(3):319–327, 2004.

| Method | Observations [error variance] | $\langle$RMSE$\rangle$ extratropical | $\langle$RMSE$\rangle$ tropical | $\langle$RMSE$\rangle$ ocean | $\langle$RMSE$\rangle$ full | $\langle$dim$_{KY}\rangle$ |
|---|---|---|---|---|---|---|
| CLVs - 9 | $x_e, y_e, z_e$ $[1, 1, 1]$ | 0.0640 | 8.4752 | 36.3662 | 21.7108 | 4.1332 |
| CLVs - var | $x_e, y_e, z_e$ $[1, 1, 1]$ | 0.0670 | 9.4872 | 40.4894 | 24.1528 | 4.1152 |
| CLVs - 9 adaptive gain | $x_e, y_e, z_e$ $[1, 1, 1]$ | 0.0032 | 0.7241 | 3.5757 | 2.1504 | 5.9991 |
| CLVs - var adaptive gain | $x_e, y_e, z_e$ $[1, 1, 1]$ | 0.0034 | 0.8350 | 4.0632 | 2.4501 | 5.8888 |

Table 6: Summary metrics of DA experiments using left-transform matrix (ESRF) and the full extratropical subsystem as observations ($x_e, y_e, z_e$). We use perfect observations (no random error added to the control run) with the observation error covariances set to the standard values as in [12]. Parameters: assimilation window 0.02, inflation factor 1%, 10 ensemble members.

[11] Patrick N Raanes, Marc Bocquet, and Alberto Carrassi. Adaptive co-variance inflation in the ensemble kalman filter by gaussian scale mixtures. *Quarterly Journal of the Royal Meteorological Society*, 145(718):53–75, 2019.

[12] Takuma Yoshida and Eugenia Kalnay. Correlation-cutoff method for covariance localization in strongly coupled data assimilation. *Monthly Weather Review*, 146(9):2881–2889, 2018.

---

## Author Response (AR1)

**Response to anonymous reviewer comments**

Dear Reviewers,

We would like to thank you both for your thorough reviews of our manuscript. We feel that your comments prompted insightful discussion that has greatly improved our manuscript. Below we include our responses to your comments in blue. We have also made adjustments to the manuscript where necessary and have included our revised manuscript with tracked changes.

Many thanks,

Courtney Quinn, Terence O'Kane, and Vassili Kitsios
* * *
**REVIEW 1**

**General comments**

This manuscript is exploring the use of covariant Lyapunov vectors (CLVs) to build the error covariance matrix in ensemble Kalman filtring methods. The set of vectors is selected based on the computation of a local Kaplan-Yorke dimension based on the finite time Lyapunov exponents. This approach is implemented in the context of a multi-scale system mimicking the (coupled) dynamics of a coupled tropical ocean-atmosphere system and the extra-tropical atmosphere. Different strategies of observation are then evaluated. It is found that observation within the atmosphere is essential,and that the variable number of CLVs to be used in building the error covariance matrix is a successful strategy for strongly coupled data assimilation. Very interesting results are also obtained with the observation sampling of a shadowing trajectory, leading to measurement correlations.

This is an interesting manuscript exploring many aspects of the strongly coupled data assimilation and I would in principle recommend publication of this work.

We thank the reviewer for the positive evaluation.

I have however an important concern on the use of the local Kaplan Yorke dimension and the CLVs that should be addressed before publication. It seems to me that the use of both is inconsistent. Let me clarify my point. When computing the FTLEs, one can use either the QR (associated with the backward Lyapunov vectors) decomposition,the Forward Lyapunov vectors obtained with backward integration in time, or the local amplification along the CLVs.

Although all these are giving the appropriate asymptotic Lyapunov exponents, they are not providing similar variability of these quantities as illustrated for instance in Vannitsem and Lucarini (2016) you quoted. So if you use the QR decomposition and then select the CLVs on that basis this is probably not optimal.

We have considered the reviewer's concern and are in agreement that the use of the two quantities could produce inconsistencies. To clarify, we use the QR decomposition method to calculate the FTLEs, which would give amplification rates for the backward Lyapunov vectors rather than the CLVs. However, in this case we argue that the way in which we utilise the FTLEs does not create an inconsistency with the CLVs in that we are not assigning any particular FTLE to a CLV. Rather, the backward FTLEs are used only to produce an adequate estimate of the number of CLVs to retain in constructing the covariance matrix. The true local dimension of the system may be different from the quantity computed, but the Kaplan-Yorke dimension calculated from the backward FTLEs still produces a sufficient upper bound. We expand on this point further below.

Figure 1 shows the finite-time Kaplan-Yorke dimension of a segment of the example model run from Sections 2 and 3 in the manuscript (compare to Figure 5 in manuscript). The top panel is the dimension measure, while the bottom is the ceiling of the dimension measure which is used to select the number of CLVs for constructing the covariance matrix. We compare the two different methods for calculating the CLVs that the reviewer mentions: growth rate along CLVs (FTCLEs) and QR decomposition (FTBLEs). We observe that for our particular nonlinear system, the QR decomposition method consistently gives a larger estimate for the Kaplan-Yorke measure than the amplification along CLVs, and therefore implies retaining more CLVs in the analysis. We maintain that a higher number of retained CLVs will not harm the DA, while too few CLVs potentially could. Specifically, where the positive CLVs are closely aligned, the DA requires the inclusion of most of the additional neutral and stable CLVs to avoid ensemble collapse (further discussion of this point on the following page). Thus, Kaplan-Yorke is an effective choice to determine the rank of the background covariance matrix despite the any inaccuracies due to degeneracy in the unstable directions. Additionally, we mention that in order to obtain the growth rate of the CLVs, one needs the accurate forward model. Since we are calculating the CLVs from the ensemble mean trajectory within the DA experiments, we do not have an accurate future trajectory and therefore no forward model (due to the state dependence of the Jacobian). For these reasons we maintain that the FTLEs from the QR method are a sufficient for our analysis at hand.

[Figure]

Figure 1: Comparison of finite-time Kaplan-Yorke dimension calculated using the growth rates of CLVs (FTCLEs) and the QR method (FTBLEs).

[Figure]

Figure 2: Comparison of variance of the individual FTLEs calculated using the growth rates of CLVs (FTCLEs) and the QR method (FTBLEs).

Alternatively, if you use an estimate of the FTLE using the amplification along the CLVs then the "dimension" of the subspace of instabilities is not the same and one can wonder what is the signification of this quantity. This is related to your comment at line 14 of page 8 indicating that higher dimension is associated with more important alignment of CLVs. Imagine for instance that several CLVs are pointing in almost the same direction with large amplifications, then the dimension would be large but intuitively (as they point all in the same direction) we would expect a low dimension. The specific way you compute the FTLEs and the local dimension should therefore be clarified, and probably I would not call it "dimension". Furthermore, if the KY dimension is computed with the local amplification rates of the backward Lyapunov exponents, a comparison should be made with the use of the backward Lyapunov vectors in building the error covariance matrix. It would have been my first choice in view of the fact this is much less costly than the CLVs.

The reviewer is correct in that the finite-time Kaplan-Yorke measure is not necessarily the true local dimension. However, this measure is giving an indication of local dimension combined with alignment. We explain this through an examination of two difference time steps in the benchmark DA experiment (Section 5.2 in the manuscript) where a high Kaplan-Yorke measure is recorded and the FTLE behaviour is similar. The alignment of the CLVs and the FTLEs (computed using the QR decomposition method) are shown in Figure 3. In both cases we see that there are 2 equally strong leading unstable FTLEs, 3 positive but near zero FTLEs, and one weakly stable FTLE. In the first column ($t = 306.24$) we see that the 5 leading CLVs are strongly aligned as well as the two most stable (8 and 9). The dimension based on alignment would then intuitively be around 4, and one would need to select the set of CLVs that are not aligned. In our method since $\dim_{KY} > 7$ we would retain 8 CLVs, therefore accounting for the strong alignment of the leading CLVs and retaining all necessary directions. In the second column ($t = 705.04$) the case is quite different in that the leading CLVs are not strongly aligned, but there is strong alignment of CLVs 4-6 and pairs 3,7 and 8,9. This would give an alignment-based dimension around 5, but again we need to retain up to the 8th CLV to span the local manifold. While one could create a method based on alignment for selecting directions, we point out that the actual criteria for "strong alignment" is arbitrary and one could risk excluding a significant direction.

With regards to using the backwards Lyapunov vectors (BLVs), if one is using the QR method then the backwards Lyapunov vectors computed are orthogonal by construction. We therefore do not see the same behaviour of alignment of the vectors. We ran the benchmark DA experiment (Section 5.2 in the manuscript) using a variable number of BLVs based on the Kaplan-Yorke measure computed using the QR decomposition method (statistics shown in Table 2 of revised manuscript). We see that the variable BLVs and variable CLVs perform comparably. While all experiments could be run with the BLVs as suggested by the reviewer, we aim to show the functionality of using the CLVs as they provide additional information about the phase-space dynamics which can

[Figure]

Figure 3: We compare the alignment of the CLVs at two time steps of the benchmark DA experiment (section 5.2 in the manuscript). The two time steps have similar Kaplan-Yorke measure and distribution of leading FTBLEs, shown in the bottom panels. We observe that the behaviour of the alignment can be vastly different for similar FTBLE behaviour, however the method of retaining CLVs based on the Kaplan-Yorke measure gives a reliable way to reflect the true local dimension regardless of alignment.

be analysed *a posteriori*.

We have made the following additions to the manuscript to address all the above concerns and clarify our methods:

- We have added a paragraph and figure to Section 3 of the manuscript to discuss the differences in calculating the FTLEs.

- We have added Figure 3 along with a discussion of alignment, dimension, and relevance to ensemble DA in Section 4.2.

- We have added the statistics for using BLVs with variable Kaplan-Yorke measure in the benchmark DA case of Section 5.2.

**Specific comments:**

1. Figure 3, you mentioned 2 neutral Lyapunov exponents. I am wondering why you have two such exponents. Is there any specific symmetries allowing for that? Isn't it a numerical artifact?

   For clarification, the model has a neutral and a near-neutral Lyapunov exponent. The near-neutral exponent comes from the weak coupling to the extratropical subsystem ($c_e = 0.08$). If one were to set $c_e = 0$, the extratropical subsystem would be completely uncoupled from the tropical and ocean subsystems, and it would retain its symmetry $[x_e, y_e, z_e] = [-x_e, -y_e, z_e]$. This implies an additional neutral Lyapunov exponent. However, we retain that the near-neutral exponent is important to consider within the neutral subspace in this study as the timescales of the neutral and near-neutral exponents are indistinguishable and changes in both can significantly affect the local Kaplan-Yorke measure. To improve and assess numerical precision, we have computed the Lyapunov exponents over a longer window (500,000 time steps) and show the results for the original parameter values as well as different combinations of coupling strengths set to zero in Table 1.

| | $\lambda_1$ | $\lambda_2$ | $\lambda_3$ | $\lambda_4$ | $\lambda_5$ | $\lambda_6$ | $\lambda_7$ | $\lambda_8$ | $\lambda_9$ |
|---|---|---|---|---|---|---|---|---|---|
| as in manuscript | 0.9043 | 0.3052 | 0.0007 | -0.0032 | -0.4829 | -0.8008 | -1.8149 | -12.2359 | -14.5726 |
| $c_e = 0$ | 0.9083 | 0.3029 | 0.0001 | -0.0006 | -0.4814 | -0.7962 | -1.8172 | -12.2415 | -14.5744 |
| $c = 0$ | 0.9042 | 0.3491 | 0.0597 | -0.0002 | -0.0151 | -0.3186 | -1.6222 | -13.4793 | -14.5777 |
| $c_z = 0$ | 0.9081 | -0.0004 | -0.0723 | -0.0728 | -0.1283 | -0.1289 | -1.1599 | -13.4702 | -14.5753 |
| $c = c_z = 0$ | 0.9069 | 0.8886 | 0.0902 | 0.0001 | -0.0004 | -0.0741 | -1.4569 | -14.4801 | -14.5743 |
| $c_e = c = 0$ | 0.9083 | 0.3581 | 0.0692 | -0.0006 | -0.0007 | -0.3227 | -1.6360 | -13.5012 | -14.5744 |
| $c_e = c = c_z = 0$ | 0.9083 | 0.9083 | 0.0902 | 0.0001 | -0.0006 | -0.0006 | -1.4569 | -14.5744 | -14.5744 |

Table 1: Asymptotic Lyapunov exponents of coupled Lorenz model for different coupling coefficients set to 0. Lyapunov exponents computed over 5000 time units using a QR decomposition method, time step of 0.01, and orthogonalization step of 0.25.

2. Table 2. The RMSE for the extratropics are very close to each other whatever the experiments. Are the differences significant?

   We preface this response with pointing out that the statistics of Table 2 have slightly changed on fixing a numerical error in the code (see revised manuscript). In order to discuss significance of differences in the average RMSE we have to compute error bounds on the means. To do this, we perform a bootstrapping of the time series of extratropical RMSE for each of the five experiments listed in Table 2 of the revised manuscript. We resample the data (with replacement) 10,000 times and then compute

[Figure]

Figure 4: Histograms for the means of the resampled time-dependent extratropical RMSEs for each respective experiment. The black lines show the means of the histograms, which are approximately the values of extratropical average RMSE in Table 2 of the manuscript.

the average RMSE for each resample. This produces a distribution of average RMSE for each experiment, shown in Figure 4. Each distribution has standard deviation $\sigma \approx 0.004$, and we see that the mean of the distributions shift by less than $\sigma$ between experiments with the exception of the variable CLVs experiment. The mean of the variable CLVs experiment still lies within $2\sigma$ of all the experiment distributions. In this sense we can conclude that the values listed in Table 2 of the revised manuscript are not significantly different. Additionally, we note that in the sense of quantifying the DA performance, all five methods constrain the extratropical system to a similar degree.

3. Also in Table 2. An average dimension is computed. This average is based on the QR decomposition or some estimate with the CLVs? This is related to my main point. Please clarify how it is computed.

The average dimension comes from the FTLEs which are calculated using the QR decomposition. We have clarified this in the manuscript.

4. FTLEs are computed for 4 time units. What is happening to your analysis when this window is changed? And why choosing this specific window?

The window of 4 time units was chosen in order to capture dimension changes related to the ENSO-like excursions in the ocean subsystem (occurring anywhere between approximately 6.5 and 27 time units). The window should therefore be long enough to filter out the intrinsic oscillatory behaviour of the ocean subsystem but short enough to capture the transition to the excursion state. The period of intrinsic variability is approximately 3 time units, therefore any window between 3 and 6.5 time units should perform comparably. Decreasing and increasing the window leads to higher and lower temporal variability of the FTLEs, respectively.

5. At page 12, in the two first paragraphs of Section 5, you present how the experiment is done. As far as I understood, the CLVs are computed during a limited period of time during the assimilation period. Am I right? At first reading it was not very clear to me and it would be nice to improve the presentation of that part. In particular, a sketch of the whole process in a figure would be really useful.

   We have clarified the paragraph where we explain the computation of the CLVs within the DA framework.

6. In the partially observed CDA, you also compute a local dimension. I am wondering how the CLVs are computed there since the trajectory of the model is probably very far from reality. Moreover, it was not clear to me whether you are using the CLVs of the reality or of the model integration. Would you please clarify how you do this? It can maybe be incorporated in the general description of the experiments mentioned at point 5 above.

   The CLVs are calculated from the ensemble mean trajectory. In the cases where the full system is constrained (atmosphere-only observations), the ensemble mean remains close enough to the truth such that the CLVs can still be calculated to a sufficient accuracy. On the other hand, this is not the case when some of the subsystems are unconstrained (i.e. ENSO and extratropical-only observations). In such cases we cannot use the variable dimension method, and instead we use all 9 computed CLVs. This is valid since the forecast error covariance matrix has rank 9 in this set-up and we are projecting onto a basis of equivalent rank, regardless of the accuracy of the computed CLVs. The local dimension we are calculating comes from the QR decomposition as mentioned in the response to the main comment. While the CLVs (or BLVs) may be quite inaccurate at a given point in time, the average dimension can give an idea of variance in the unobserved subsystems. In cases where an unobserved subsystem is unconstrained, it is common to observe variance collapse in the ensemble mean of that subsystem and we subsequently record a lower average local dimension. We therefore explore methods to maintain the variance in the unobserved subsystems (i.e. increase the average local dimension), with the caveat that the instantaneous dimension will not necessarily be reflective of the true dynamics of the system. We have provided additional clarification in the manuscript about the calculation of the dynamical properties within the DA experiments.

**Minor points:**

1. Line 7, page 11. Please modify the notation on the brackets. It looks like a vertical rectangle.

   We have changed the text to read "...and the **subscript** $*, n$ denotes taking ...".

2. Line 20, page 11. I suppose that $\lambda$ should be larger than 1.

   We have added a condition to Equation 19 specifying $\lambda > 1$.

3. Line 22, page 11. Please do not use the terminology "model error". It is confusing as model error is used when there is structural uncertainty in the model.

   We have removed the term "model" as we agree that the usage is confusing in this instance.
* * *
**REVIEW 2**

**Main points:**

1. Although the bibliography is rather dense (and better done than usual), there are still a few (very) relevant references such as Palatella and Trevisan (2015) [8]; Grudzien et al. (2018a,b) [3, 4] that are missing.

   We thank the reviewer for pointing out the additional relevant references and we will include them in accordance with the specific suggestions below.

2. Even though the number of CLVs incorporated in the DA algorithm depends on time, you still need to compute a number of CLVs corresponding to the maximum local KY, do you? So that the gain is computationally limited. If I am wrong, please explain.

   In the way that we have implemented the variable dimension, one could compute the local KY measure first and then only compute the number of CLVs corresponding to that given value at each assimilation time step. (This is due to the fact that we compute dimension with the QR-factorization approach and the CLVs with a separate algorithm - clarification of this and further discussion has been added to the modified manuscript.) With respect to computational gain, there will be an additional cost associated with computing the unstable manifold regardless of

the method. Whether or not this is realizable in high dimensional systems, or how the overall cost compares to increasing ensemble size in order to fully sample the variance are not questions we attempt to answer in this study. We rather focused on the applicability of assimilation in a variable unstable subspace for different situations of strongly coupled data assimilation (*i.e.* different types of observational subsets). We discuss the reviewer's comments regarding computational cost more within the specific points below.

3. In at least a few experiments, you need to use optimally tuned inflation (section 5.3 for instance), since it is already known that the lack of span of the unstable modes can be compensated with by a stronger multiplicative inflation. Otherwise several of your claims are undermined.

We first preface our response with bringing to the reviewer's attention that there was a small error in the calculation of the CLVs for the results shown in the manuscript submission. We have since corrected this error and updated all of our experiments. The updated tables are included in the revised manuscript. Due to these corrected results, we now can conclude for the benchmark and atmosphere observations that all experiments are successful. The emphasis of the discussion is now regarding the variable CLV full RMSE being closest to the full rank experiments for each of the observation sets. These conclusions hold for a range of inflation values. We show in Table 2 below the results for the benchmark observations using full rank, 5 CLVs, and variable CLVs with inflation values of 1% (as in manuscript - revised values), 2%, 5% and 10%. As the lowest RMSEs occur for the 1% inflation cases, we conclude this is the optimal inflation value and leave these results in the manuscript. We have added a statement noting this to the beginning of the results section (Section 5 in the manuscript). The optimization was already done for the shadowed observations (Section 5.5), and the optimal inflation values are given in the table. All other experiments have an optimal inflation of 1%.

4. The new inflation scheme is not justified enough. Beware that it has been tested in a very specific case and does not warrant generality.

We agree with the reviewer that our language when discussing the utility of the adaptive scaling for the Kalman gain is too strong. We have changed the language to emphasize that the adaptive scaling works in this particular case, but needs to be tested on more models to form a a more general applicability. We also have added additional justification for the proposed scheme and relate it to the work of Miller *et al* [6]. More detailed discussion of our justification can be found in response to the specific points of the following section.

5. Because of the above points, there are too strong statements in the conclusion regarding the novelty and performance of the proposed method.

| Method | Observations [error variance] | $\langle$RMSE$\rangle$ extratropical | $\langle$RMSE$\rangle$ tropical | $\langle$RMSE$\rangle$ ocean | $\langle$RMSE$\rangle$ full | $\langle\dim_{KY}\rangle$ |
|---|---|---|---|---|---|---|
| CLVs - 9 1% inflation | $y_e, y_t, Y$ $[1, 1, 25]$ | 0.3142 | 0.1598 | 0.4948 | 0.4027 | 5.8928 |
| CLVs - 5 1% inflation | $y_e, y_t, Y$ $[1, 1, 25]$ | 0.3123 | 0.1843 | 0.5920 | 0.4550 | 5.8870 |
| CLVs - variable 1% inflation | $y_e, y_t, Y$ $[1, 1, 25]$ | 0.3215 | 0.1688 | 0.5346 | 0.4272 | 5.8863 |
| CLVs - 9 2% inflation | $y_e, y_t, Y$ $[1, 1, 25]$ | 0.3286 | 0.1893 | 0.5985 | 0.4625 | 5.8881 |
| CLVs - 5 2% inflation | $y_e, y_t, Y$ $[1, 1, 25]$ | 0.3274 | 0.1950 | 0.6083 | 0.4681 | 5.8871 |
| CLVs - variable 2% inflation | $y_e, y_t, Y$ $[1, 1, 25]$ | 0.3219 | 0.1916 | 0.5966 | 0.4594 | 5.8915 |
| CLVs - 9 5% inflation | $y_e, y_t, Y$ $[1, 1, 25]$ | 0.3564 | 0.2627 | 0.8858 | 0.6237 | 5.8832 |
| CLVs - 5 5% inflation | $y_e, y_t, Y$ $[1, 1, 25]$ | 0.3600 | 0.2665 | 0.8880 | 0.6264 | 5.8836 |
| CLVs - variable 5% inflation | $y_e, y_t, Y$ $[1, 1, 25]$ | 0.3552 | 0.2647 | 0.8747 | 0.6180 | 5.8848 |
| CLVs - 9 10% inflation | $y_e, y_t, Y$ $[1, 1, 25]$ | 0.4117 | 0.3719 | 1.3278 | 0.8799 | 5.8763 |
| CLVs - 5 10% inflation | $y_e, y_t, Y$ $[1, 1, 25]$ | 0.4187 | 0.3873 | 1.3168 | 0.8787 | 5.8824 |
| CLVs - variable 10% inflation | $y_e, y_t, Y$ $[1, 1, 25]$ | 0.4123 | 0.3731 | 1.3153 | 0.8736 | 5.8775 |

Table 2: Summary metrics of DA experiments using right-transform matrix (ETKF) and benchmark observations $(y_e, y_t, Y)$. The angle brackets $\langle\cdot\rangle$ denote average over analysis steps. Compare to results in [11]. Parameters: analysis window 0.08, 10 ensemble members.

Again, we agree with the reviewer that too strong of language is used in regards to some of the conclusions. We note that due to our corrected results we have already adjusted some of our conclusions accordingly. We will additionally adjust the specific statements mentioned in the following section.

**Specific suggestions:**

1. p.1, l.5: "to determine" is a bit ambiguous as it could also need "to infer" which would be a bold statement. I guess you meant "to prescribe", right?

We have changed the statement to read "to prescribe" instead, as this was our intended connotation.

2. p.2, l.2: "implying very large ensemble sizes are needed" → "implying that very large ensemble sizes are needed

   We have amended the sentence.

3. p.3, l.6: What are "local CLVs"? I know local LEs, but not local CLVs.

   This was a typo. We were referring to the CLVs calculated at a given time. We have removed the term "local".

4. p.3: Even though the beginning of the paper is very good and enjoyable, reading "We also examine the role of correlated versus random observational errors.", "along with a novel scheme for adaptive Kalman gain inflation." at the end of the introduction is odd as these two subjects do not seem directly connected to the main objective of the paper, and they seem, at this stage of the reading, unnecessary.

   We can agree that the statements are superfluous. We will save the discussion of these two topics for when they arise in the study.

5. p.4, l.18: "uncentering parameters": please explain what this means.

   The terminology "uncentering" was originally used in the Peña and Kalnay study [9] and refers to the uncentering of the unstable zero equilibrium in the classical Lorenz model. Dynamically, the $k_1$ and $k_2$ values used in the manuscript allow for the second unstable asymptotic Lyapunov exponent (for $k_1 = k_2 = 0$: $\lambda_1 = 0.91, \lambda_2 = 0, \lambda_i < 0$ for $i > 2$).

6. p.4, l.27: "We are interested in analysing both the local and global dynamics of system (1).": I guess you mean the short-term and asymptotics dynamics - your words lack accuracy here.

   That is precisely what we meant. We will change the language as suggested.

7. p.6, l.16: Can you be sure/prove that the last digits of 5.9473 are relevant?

   We do not claim that the last digits are relevant, they are included only as the Lyapunov exponents were given up to four digits.

8. p.7, l.2: "approach their corresponding asymptotic values" → "approach their corresponding LEs asymptotic values"

   We have changed the sentence to read "approach the asymptotic Lyapunov exponent values".

9. p.7, l.3: You have to discuss/justify more the concept of local KY dimension. This is the main idea of your paper.

   We have added some additional discussion around the concept of local dimension used in our manuscript as suggested.

10. p.7, l.4: "We see the local dimension"→ "We see that the local dimension"

    We have amended the sentence.

11. p.7, l.13: "the cocycle of": Please explain what a cocycle is. How is it connected to (1)?

    In relation to system (1) in the manuscript, a cocycle is the forward and backward mapping of solutions under the tangent dynamics. The cocycle is then given as $\mathcal{A}(x(t), \tau) = e^{\tau Jf(x(t))}$ where $Jf$ denotes the Jacobian of $f$, the right-hand-side of system (1). We have clarified this in the manuscript and have restructured the sentence such that *a prior* knowledge of the term cocycle is not assumed.

12. p.8, l.7: "local time-varying Kaplan-Yorke dimension.": what is its interpretation? This is key to your paper.

    We have added a more descriptive explanation of the local Kaplan-Yorke dimension here, particularly relating to our implementation and its relation to the CLV behaviour.

13. p.8, l.16-19: Palatella and Trevisan (2015) [8] could be mentioned here.

    We have added the reference.

14. p.9, l.4: "Suppose there exists"→ "Suppose that there exists"

    We have amended the sentence.

15. p.9, Fig.5, legend: replace the "v" symbol by "or" for the sake of clarity.

    We have changed the legend accordingly.

16. p.9, l.9-11: The errors are also supposed uncorrelated in time (white).

    We have clarified the assumption of temporally uncorrelated errors in the description of the Kalman filter method.

17. p.9, Eqs.(8,9): Missing punctuation.

    We have added punctuation to the noted equations.

18. p.10, l.5: "There is difficulty"→"There is a difficulty"

    We have amended the sentence.

19. p.10, l.7: "assumption of linearity": this is confusing here since you have not introduced the extended Kalman filter yet.

    We have removed the statement in which this phrase appears as it is not critical to the reader.

20. p.10, l.21: Even though quoting Bishop et al. (2001) [1] is certainly adequate, a reference to Hunt et al. (2007) [5] is also missing as it is equally relevant.

    We have added the reference as suggested.

21. p.10, Eq.(14) is wrong, is it? It should be

$$\mathbf{E} = \mathbf{R}^{-1/2}\mathbf{H}\mathbf{X}^f. \tag{1}$$

    Also it is not recommended to use $\mathbf{E}$ as it is usually used for the full ensemble matrix. Authors often use $\mathbf{S}$ instead.

    We thank the reviewer for catching this typo. We did in fact mean to define the matrix as above, as the left multiplication with its transpose is included in Eq (15). We have also changed the notation from $\mathbf{E}$ to $\mathbf{S}$ as suggested.

22. p.11, l.7: "The Kalman gain K is defined through equation (9a)": No. Not in the classical ETKF (see Hunt et al. (2007) [5]).

    We have changed the text to specify that we are following the definition of the Kalman filter as given in Bishop et al [1].

23. p.11, l.15-20: In this context, sampling errors are actually due to nonlinearity, as it was explained and proven by Bocquet et al. (2015) [2]; Raanes et al. (2019) [10].

    We have included the additional references here where we discuss the sampling errors due to nonlinearity.

24. p.11, l.25-27: "This differs to past approaches where the subspace was determined in terms of the long time averaged (invariant) unstable and neutral CLVs (Trevisan and Uboldi, 2004; Carrassi et al., 2008; Trevisan and Palatella, 2011).": This statement is misleading. You just mean that the number of retained CLVs is kept fixed. Did you?

    This is badly phrased. We meant "asymptotic Lyapunov exponents", and

therefore the rank of the error covariance matrix is kept fixed. We have updated the manuscript accordingly.

25. p.12, l.7-8: "We compute CLVs at the assimilation step using Algorithm 1.": The number of CLVs is fixed over the full time span of the algorithm, is it?

    We always compute all 9 CLVs in this case, and then only use the number corresponding to the experiment set-up in constructing the covariance matrix. In this toy model we were not as concerned with optimizing computation time. However, it is easily implemented to only compute a subset of the CLVs either specified as constant or variable based on the calculated dimension for the assimilation step. (Note, the dimension is calculated from the FTLEs computed using the QR method so this is a separate process to the CLV calculation.) We discuss more about computation time in response to the final comment (44) below.

26. p.12, l.19: "regardless of observation set" → "regardless of the observation set"

    We have amended the sentence.

27. p.12, l.23: The term "Analysis window" is unfortunate as it usually refers to the time range over which asynchronous observations are assimilated in 4D-Var or with an ensemble smoother. I guess you mean the time interval between updates. You could denote it $\Delta t$ for instance. Please change it throughout the manuscript.

    We have changed "analysis window" to "assimilation window" throughout the manuscript ($\Delta t$ conflicts with our integration time step notation).

28. p.13, section 5.1: Please explain better what changing the observation set has to do with the main goal of the manuscript.

    The different observation sets relate to the exploration of strongly coupled data assimilation (strong CDA), namely the use of cross-covariances to update unobserved states. We were interested in whether the use of the reduced space method (either fixed or variable) is applicable in different observation scenarios, including the cases where some subsystems are left unobserved or where there are temporally correlated errors. We have emphasized this in the introduction to the results section (Section 5) and in Section 5.1.

29. p.13, l.9-10: "We argue here that in reality, the true variance of the observation error can be spatially dependent and errors are often correlated in time.": this is a bit too much, since there are quite a few DA papers dealing with at least spatially correlated errors.

We have revised the tone of the sentence which now reads: "In many applications the true variance ..."

30. p.13, l.27: "we decrease the analysis window and do not perturb the control run at all when taking the observations.": You mean that the synthetic observations are not perturbed, do you? The sentence seems a bit twisted.

We have amended the sentence to read: "we decrease the assimilation window and assume perfect observations."

31. p.15, l.15: "Finally we analyse our novel reduced subspace method which uses a variable number of CLVs based on the local Kaplan-Yorke dimension.": yes, but I guess you need to compute a number of CLVs corresponding to the maximum local KY dimension, so that even though it is theoretically interesting, it is, in practice, of limited interest.

We take this as a comment. To clarify, as mentioned in our response to point 25, one only needs to compute the number of CLVs corresponding to the local Kaplan-Yorke dimension for a given assimilation step. In our experiments this varies between 0 and 8 for different assimilation steps. We discuss practical implementation in response to the reviewer's final point below.

32. p.17, Fig. 7 (a-c): please plot over a smaller range, typically [500-600] as in Fig.16.

We have changed the range of both figures (16 and 17 in previous manuscript) to [450-550] in accordance with the discussion around the dynamical properties.

33. p.18, Table 3, and discussion around: This experiment does not account for what is actually known in the literature. You should have made an experiment with the 6 CLVs but with optimally tuned inflation, or you could have used the finite-size EnKF (Bocquet et al., 2015, and references therein). It is by now well known that the gap between the second and the third experiment might be compensated by optimally tuned inflation.

This point was addressed in the main comments section above. With our correction to the CLV calculation, we no longer have such a discrepancy between experiments. The corrected table can be found in the following section. We have additionally done an optimization on inflation and find that the 1% is in fact optimal for these experiments.

34. p.19, l.16-18: "but we are interested if we can preserve": I don't really understand the phrase.

We have changed the sentence to read: "we are interested to see if we

can avoid collapse (*i.e.* the loss of variability) in the ensemble mean of the extratropical attractor".

35. p.21, l.1: "in ability" → "in the ability"

    We have amended the sentence.

36. p.25, l.5-10: In this paragraph, you argue but you don't give a strong rationale for the inflation scheme you propose. You need a stronger case to convince the reader. All the more since the inflation scheme is tested with a toy model in a very specific configuration.

    We have added further discussion of the Miller *et al* [6] study to motivate our proposed modified Kalman gain. Through the study of data assimilation schemes on the Lorenz '63 model, they find that the forecast error covariance is often underestimated in highly nonlinear systems, particularly when the model is in a region of phase space subject to transitions. This leads to the Kalman gain being underestimated. The authors account for this by including the third and fourth moments of anomalies within the Kalman gain calculation, however this is done for the Extended Kalman Filter (EKF) method (see eq. 4.3 of [6]). O'Kane and Frederiksen [7] also derive a higher order gain for a closure-based statistical dynamical Kalman filter applied in spectral space (see eq. 27 of [7]). Here we use the notion that increased spread in a subsystem represents the inability of ensemble members to track the same transitions in phase space. In such a situation the forecast error covariance is likely to be underestimated (presumably due to emerging importance of higher moments), therefore we scale the forecast error covariance within the Kalman gain calculation by a factor that represents a measure of overall spread in the system. This then provides an adaptive scaling of the Kalman gain at every assimilation time step based on the background performance of the system (large spread implies an increase in Kalman gain, small spread implies a decrease). The scaling can equivalently be written as an observation error variance scaling, and its overall behaviour is a balancing between the forecast error covariance and the observation error variance within the calculation of the Kalman gain. We have expanded upon this discussion within the manuscript.

37. p.25, Eq.(25): The modified $\mathbf{P}^f$ does no have the good engineering dimensional (cube in the anomalies instead of square). What do you make of this?

    Our introduction of the scaled Kalman gain was written poorly here. We actually do not modify the forecast error covariance matrix as worded in the original manuscript. We add an additional scaling term to the forecast error covariance matrix only within the calculation of the Kalman gain. The scaling factor ($||\mathbf{P}^f||$) itself is a measure of the anomalies squared,

therefore making the quantity used in the Kalman gain a quartic measure of anomalies. In a crude way one could consider this an incorporation of higher moments into the Kalman gain, however since this is not a true estimate of the fourth moment, we used the spread-based argument for its motivation (as given above).

38. p.25, l.23-24: It seems like the $\beta$-factor approach is implementing deflation. Is it so? If yes, please use the term deflation instead of inflation, which is customary.

We have changed the language to refer to the $\beta$-factor approach as deflation.

39. p.27, l.1: "there is remarkable"→"there is a remarkable"

We amended the sentence.

40. p.27, l.7-8: "We have demonstrated the varying rank of the error covariance matrix related to the transient growth in the stable modes of the system.": true but this was already emphasised in the literature, so that your implicit statement of novelty should be tuned down here.

We agree with the reviewer that our statement was too strongly worded. We have amended the sentence to read: "We have explored the varying rank of the error covariance matrix related to the transient growth in the stable modes of the system, and in particular the applicability of this varying rank on different configurations of strong CDA."

41. p.27, l.16-17: "to determine the rank": unclear and confusing; I believe you mean "to specify the appropriate rank"; you don't discover the rank, you set it.

That is correct, we meant the connotation "to specify". We have revised the sentence accordingly.

42. p.27, l.22-24: "In particular, we found that spanning the space comprised of the asymptotic unstable, neutral, and first weakly stable mode (5 CLVs in this case) performed much worse than using either dimension measure (asymptotic and local).": I don't think so. This is one weak point of your study. It is known that in this case, the inflation must be adjusted to account for the error upscaling from the the region of the spectrum (Grudzien et al., 2018a,b). I don't believe that you have tuned the inflation, have you? If true, your statement appears to be too strong.

We have toned down the language around the discussion of these results after the correction to the calculation. We no longer claim the 5 CLV case performs "much" worse, as all methods now perform comparably. We

point out, however, that the 6 CLV and variable CLV both show a reduction in the full RMSE which can be related to significant linear growth in more than one stable mode. We also have included the reference [4] within our discussion of inflation in Section 4.1, and the reference [3] in discussing the impact of transient growth of asymptotically stable modes on model errors in the introduction.

43. p.28, l.19-20: "The adaptive scaling introduced here can be applied to general systems with weak coupling, although care may need to be taken in the choice of the norm.": No, you have not proven anything like that. Please remove the statement.

Instead of removing, we have amended the statement to one which reflects our original sentiment. We wanted to suggest that the method be tested on other systems of weak coupling and then perhaps a more general statement can be made under deeper analysis. We have changed the statement to read "The adaptive scaling introduced here should be tested on additional systems with weak coupling in order to assess its general applicability, although care may need to be taken in the choice of the norm."

44. p.28, l.33-34: "Future work should also consider the numerical cost of CLV calculation and methods to increase efficiency for high dimensional systems": At the very end, you raise what experts familiar with AUS have in mind reading your paper: what you propose is certainly interesting and of theoretical interest but of lesser practical value since (i) one needs to compute the CLVs alongside (ii) with the variable CLV context, you need to compute the maximum number of CLVs. You should mentioned this point way earlier in the paper, unless I am mistaken.

The reviewer is correct in that we do not address computational cost throughout the paper. This was mainly intentional, as we did not aim to optimize computational performance, but rather explore a method that allows for the reduction of phase space and/or better representation of errors in that reduced subspace. Whether or not there can be a computational cost reduction by using this method over spanning the space needed to fully sample the variance is a question we do not attempt to answer as it will depend on the system one wishes to assimilate. In accordance with the reviewer's request, we have added a footnote within the introduction at the point where we discuss assimilation in the unstable subspace (AUS). The footnote acknowledges the additional cost that comes with computing the unstable subspace which may or may not be less than the cost of sampling the full model variance, and it also states that we will leave the exploration of numerical efficiency in high dimensional systems for future study.

We additionally note that the projection onto CLVs is an example of one

specific projection that works for the variable local dimension, but it is not the only one. One could also project onto the backwards Lyapunov vectors (BLVs) which are computationally less expensive to compute and attain similar results. Table 2 in the revised manuscript includes the results of variable dimension and projection onto BLVs. We have added a discussion of this alternate projection into the manuscript, as well as the additional statistics of the variable BLV experiment. However, we focus on the CLVs in this study as the *a posteriori* analysis of CLV alignment can provide more information of the time-varying phase space behaviour. (The BLVs are orthogonal by construction.) The revised manuscript has an additional discussion of CLV alignment, variable dimension, and specifically the relationship to ensemble data assimilation which justifies the utility of CLVs.

We imagine that it could be possible to find a computationally efficient projection for high dimensional systems that incorporates the variable dimension aspect. Our main goal in this manuscript though was to introduce the implementation of time-varying dimension in this toy model with different observational sets to understand in what configurations one might expect successful results. It would be of interest for practical applications if future studies could explore numerically efficient projections that span the time-varying dimension and therefore utilise the ideas of AUS in high dimensional systems.

**SUMMARY OF MAJOR MANUSCRIPT CHANGES**

"pg" refers to page number in following latexdiff document

| Section 1 | pg 2 | Added footnote discussing cost of computing unstable/neutral manifold |
| | pg 2 | Additional relevant references [3, 4] |
| Section 2.2 | pg 7 | Introduction and discussion of concept of local dimension used throughout manuscript |
| Section 3 | pg 7 | Explanation of cocycle and mathematical formulation used |
| | pg 9 | Comparison of calculating FTLEs as growth rates of backwards or covariant Lyapunov vectors |
| | pg 10 | Additional figure comparing local KY measure calculated from FTBLEs or FTCLEs (Fig 6) |
| Sections 4.1 | pg 12 | Corrected equation 14 |
| | pg 12 | Additional references and discussion regarding use of inflation in DA of nonlinear systems |
| Section 4.2 | pg 13 | Discussion of CLV alignment and local dimension measure in context of DA |
| | pg 14 | Additional figure comparing CLV alignment at two different times (Fig 7) |
| Section 5 | pg 14 | Justification for considering different observational sets |
| | pg 15 | More detailed description of how the dynamical properties are calculated within the DA experiments |
| Section 5.2 | pg 16 | Discussion of using BLVs in place of CLVs |
| | pg 17 | Clarified definitions of metrics and removed MAD as it is not discussed |
| | pg 17 | Updated RMSE values in Table 2 from corrected experiments |
| | pg 17 | Added statistics for BLV experiment |
| Section 5.3 | pg 21 | Removed figure of DA experiments with atmosphere observations |
| | pg 21 | Corrected Table 3 and the discussion of the experiments |
| Section 5.4 | pg 23 | Updated RMSE values in Table 2 from corrected experiments |
| Section 5.5 | pg 26 | Updated RMSE and inflation values in Table 2 from corrected experiments |
| Section 5.6 | pg 26 | Additional justification for adaptive Kalman gain scheme |
| | pg 27 | Defined scheme in terms of Kalman gain scaling rather than forecast covariance scaling |
| | pg 28 | Showed the variable CLV experiment in Figure 12 rather than the 9 CLV |
| | pg 29 | Added statistics of variable CLV experiments to Table 6 |
| Section 6 | pg 28-30 | Generally softened our tone in discussion of results |
| | pg 31 | Additional paragraph discussing the adaptive Kalman gain scheme and the need for an analysis of general applicability |

[revised manuscript text omitted]
  which we will refer to as a "local" dimension. This will give a measure which reflects variations in the finite-time growth rates, with higher (lower) dimension signifying increased instability (stability) in the FTLEs. More specifically this is an upper bound on the true local dimension - the measure does not take into account the geometric degeneracy which can occur when many of the leading CLVs can become aligned. In practice this overestimation of dimension is actually beneficial within the DA framework (more discussion on this can be found in Section 4.2).

Figure 4 shows the local dimension plotted along the attractors of each of the subsystems. We see that the local dimension is lowest when the model is in the interior region of the ocean subsystem attractor. In contrast, the extratropical atmosphere subsystem attractor displays periods of low dimension largely uniformly confined to the center of each lobe of the attractor. The tropical atmosphere also shows most of the measures of low dimension confined to the interior of the attractor, reflecting the strong ocean coupling.

**3 Covariant Lyapunov vectors**

The existence of Lyapunov vectors for a large class of dynamical systems was proven by Oseledets (1968). The Multiplicative Ergodic Theorem states that there exists a relation between Lyapunov exponents, $\lambda_i$, and a (non-unique) set of vectors $\phi$ such that

$$\lambda_i = \lim_{\tau \to \infty} \frac{1}{\tau} \log \|\mathcal{A}(x(t), \tau)\phi\| \qquad \text{iff} \qquad \phi \in \Phi_i(x(t)) \setminus \Phi_{i+1}(x(t)) \tag{4}$$

Here,  $\mathcal{A}(x(t), \tau)$ is the forward and backward mapping of solutions under the tangent dynamics  of a given dynamical system  (also referred to as the cocycle). For system (1), $\mathcal{A}(x(t), \tau) = e^{\tau Jf(x(t))}$ where $Jf$ denotes the Jacobian of $f$, the right-hand-side of system (1). The subspaces ($\Phi_i$) on which the growth rates ($\lambda_i$) occur are covariant with the tangent dynamics and invariant under time reversal (Ginelli et al., 2007; Froyland et al., 2013). The covariant Lyapunov vectors (CLVs) are then defined as the set of directions at each point in phase space that satisfy (4) both backwards and forwards in time

(Ginelli et al., 2007; Ng et al., 2011). In the last few decades there have been significant advances in the ability to numerically approximate such vectors for chaotic dynamical systems (Ginelli et al., 2007; Wolfe and Samelson, 2007; Froyland et al., 2013). In this work we will employ a numerical algorithm introduced by Froyland et al. (2013) (Algorithm 2.2 in the aforementioned study). We summarize this algorithm below.
* * *
**Algorithm 1** Summary of numerical algorithm for calculating CLVs introduced by Froyland et al. (2013)
* * *
1. Construct matrix cocycle $\mathcal{A}(x^{i+m}, 0)$ for every  $m \in [-N,...,N]$.
2. Compute the eigenvectors $\boldsymbol{e}_j^{i-N}$ of $\mathcal{A}(x^{i-N}, N)^* \mathcal{A}(x^{i-N}, N)$ [the right singular vectors of $\mathcal{A}(x^{i-N}, N)$], where $\mathcal{A}(x^{i-N}, N) = \mathcal{A}(x^i, 0) \cdot ... \cdot \mathcal{A}(x^{i-N}, 0)$. Note that $\mathcal{A}(x^{i-N}, N)^*$ is the adjoint of $\mathcal{A}(x^{i-N}, N)$.
3. Push forward by multiplication of matrix cocycle, $\boldsymbol{\phi}_j^i = \mathcal{A}(x^{i-N}, N) \boldsymbol{e}_j^{i-N}$.
4. For each $j$, reorthogonalize $\boldsymbol{\phi}_j^i$ with subspace spanned by eigenvectors $\boldsymbol{e}_k^{i-n}$ for $k = 1,...,j-1$ of $\mathcal{A}(x^{i-n}, N)^* \mathcal{A}(x^{i-n}, N)$ every $n$ time steps.
5. The vector $\boldsymbol{\phi}_j^i$ is then an approximation of the $j-th$ largest CLV at time $t = t_i$.
* * *
It should be mentioned that the push forward step need not be equal to $N$; $M \neq N$ for $\mathcal{A}(x^{i-N}, M)$. However, for all calculations in this study we consider only the case $M = N$. The trajectory of the system is discretized with time step $\Delta t$ such that $x^i = x(t_i)$ and $x^{i+m} = x(t_i + m\Delta t)$.

We examine the effectiveness of this algorithm on the Peña and Kalnay (2004) model introduced in Section 2. By definition, CLVs describe the directions in phase space corresponding to different growth rates. Two or more CLVs can align during a transition to a different regime in phase space. We calculate the alignment through

$$\theta_{i,j} = \frac{|\boldsymbol{\phi}_i \cdot \boldsymbol{\phi}_j|}{||\boldsymbol{\phi}_i|| \cdot ||\boldsymbol{\phi}_j||}. \tag{5}$$

Here, $\theta_{i,j} = |\cos(\Theta_{i,j})|$ where $\Theta_{i,j}$ is the angle between the $i$-th and $j$-th CLV. Larger values of $\theta_{i,j}$ imply alignment of the two CLVs. Figure 5 shows the alignment of the unstable CLVs ($\theta_{1,2}$) and the neutral CLVs ($\theta_{3,4}$) plotted against the $X$ component of system (1), along with the FTLEs and corresponding local time-varying Kaplan-Yorke dimension.  Since the window to calculate the FTLEs was chosen based on the variability timescales of the ocean subsystem, the subsequent local dimension measure should reflect areas of increased or decreased instabilty along the ocean attractor (Figure 4). We also expect alignment of the CLVs during a transition in this subsystem. The CLVs analysed in Figure 5 are calculated from the first 35 time units of the previous model run with a time step $\Delta t = 0.01$. We start the calculation at $t = 5$ to allow for initialization and a window size of $\tau = 4$. The parameters for the algorithm are set as $N = 400$ and $n = 25$. It can be seen that the algorithm detects near alignment of either the unstable or neutral CLVs during the transitions in the ocean subsystem. The transitions here are between the inner part of the attractor with smaller oscillation amplitudes and the outer, large amplitude excursions. In general, the alignment is most prominent in the outer, large amplitude part of the attractor. This follows the changes in local dimension, shown in the lower panel of Figure 5. Higher local dimension tends to accompany alignment of the unstable and neutral CLVs. This relates to the inability of the Kaplan-Yorke dimension measure to account for finite-time geometric degeneracy.

[Figure]

**Figure 5.** Local dynamical properties of a segment of an example model run. (Top) Alignment of CLVs associated with the unstable and neutral subspaces plotted along the x-coordinate of the ocean subsystem. Large orange stars indicate high alignment of unstable or neutral CLVs ($\theta_{1,2} > 0.9$ or $\theta_{3,4} > 0.9$). (Middle) Time-varying FTLEs $\Lambda_i(t)$ computed over window $\tau = 4$. (Bottom) Local Kaplan-Yorke dimension calculated from FTLEs.

At this point we will comment on the non-trivial relationship between the CLVs and FTLEs calculated here. As discussed in Vannitsem and Lucarini (2016), there are three different types of FTLEs one can compute: backward (FTBLEs), forward (FTFLEs), and covariant (FTCLEs). Each type of FTLE gives the local growth rate of the corresponding Lyapunov vectors. Although all three converge to the asymptotic Lyapunov exponents as the computation window increases, the temporal variability

5    for finite window size can be different depending on the model at hand. Vannitsem and Lucarini found that when calculating the growth rates of the CLVs, higher variability in the FTCLEs corresponding to neutral or near-zero modes occurred compared to the other two methods. This could have implications on the local Kaplan-Yorke dimension if one were to use the FTCLEs rather than the FTBLEs in the calculation. We remark here that the QR method produces backward Lyapunov vectors (BLVs) and their corresponding FTBLEs. We therefore compare the Kaplan-Yorke dimension as computed from FTCLEs and FTBLEs

10    for the coupled Lorenz system (1) in Figure 6. We see that for this model the dimension calculated from FTBLEs approximately bounds that calculated from FTCLEs. Since we will be using the local Kaplan-Yorke dimension as a lower bound within the framework of our experiments, the FTBLEs give a conservative estimate of dimension that is varying with our dynamics (see section 4.2 for discussion of the implementation of dimension into our experiments).

In the following sections, we will utilise CLVs within the data assimilation framework of ensemble forecasting. The CLVs

15    will be used to construct the forecast error covariance matrix, which informs the increment used on ensemble members to bring them closer to observations. Using CLVs in this context suggests a more accurate method of forming the forecast error covariance matrix when the true covariance is undersampled due to insufficient number of ensemble members (see *e.g.*

[Figure]

**Figure 6.** Comparison of finite-time Kaplan-Yorke dimension calculated using the growth rates of CLVs (FTCLEs) and the QR method (FTBLEs).

Palatella and Trevisan (2015) where the authors applied a similar approach using BLVs on the classical Lorenz (1963) and Lorenz (1996) systems).

**4 Data assimilation with the Kalman filter**

We now summarize the Kalman filter equations. For detailed derivations we refer the interested reader to the reviews by
Evensen (2003), Houtekamer and Zhang (2016) or Carrassi et al. (2018). Here we follow the notation of Carrassi et al. (2018).
Consider a deterministic or stochastic model defined by

$$\boldsymbol{x}_k = \mathcal{M}_{k:k-1}(\boldsymbol{x}_{k-1}, \boldsymbol{p}) + \eta_k, \tag{6}$$

where $\boldsymbol{x}_k \in \mathbb{R}^N$ is the model state at time $t = t_k$, $\boldsymbol{p} \in \mathbb{R}^p$ are the model parameters, $\mathcal{M}_{k:k-1} : \mathbb{R}^N \to \mathbb{R}^N$ is a function taking
the model from time $t_{k-1}$ to $t_k$, and $\eta_k$ is the model error at time $t_k$ (for deterministic systems let $\eta_k = 0$). Suppose that there
exists a time-dependent set of observations $\boldsymbol{y} \in \mathbb{R}^d$ which can be expressed as a function of the model state through

$$\boldsymbol{y}_k = \mathcal{H}_k(\boldsymbol{x}_k) + \epsilon_k. \tag{7}$$

The observation operator $\mathcal{H}_k : \mathbb{R}^N \to \mathbb{R}^d$ can be linear or nonlinear, and $\epsilon_k$ is the observational error.

In the Kalman filter method, equations (6) and (7) are assumed to be linear, resulting in evolution and observation matrices
$\mathbf{M}_{k:k-1}$ and $\mathbf{H}_k$ respectively. The model and observation errors, $\eta_k$ and $\epsilon_k$, are taken to be uncorrelated in time (white noise)
and from a Gaussian distribution with covariance matrices $\mathbf{Q}_k \in \mathbb{R}^{N \times N}$ and $\mathbf{R}_k \in \mathbb{R}^{d \times d}$ respectively. There are two basic
steps to the Kalman filter method: forecast and analysis.

   – **Forecast equations**

$$\boldsymbol{x}_k^f = \mathbf{M}_{k:k-1}\boldsymbol{x}_{k-1}^a, \tag{8a}$$

$$\mathbf{P}_k^f = \mathbf{M}_{k:k-1}\mathbf{P}_{k-1}^a\mathbf{M}_{k:k-1}^{\mathrm{T}} + \mathbf{Q}_k. \tag{8b}$$

– **Analysis equations**

$$\mathbf{K}_k = \mathbf{P}_k^f \mathbf{H}_k^{\mathrm{T}} [\mathbf{H}_k \mathbf{P}_k^f \mathbf{H}_k^{\mathrm{T}} + \mathbf{R}_k]^{-1}, \tag{9a}$$

$$\boldsymbol{x}_k^a = \boldsymbol{x}_k^f + \mathbf{K}_k(\boldsymbol{y}_k - \mathbf{H}\boldsymbol{x}_k^f), \tag{9b}$$

$$\mathbf{P}_k^a = (\mathbf{I}_k - \mathbf{K}_k \mathbf{H}_k)\mathbf{P}_k^f. \tag{9c}$$

There is  a difficulty in finding accurate solutions to equations (8-9) for realistic systems which have high dimension and are nonlinear (as is the case in weather and climate forecasting).  Within the Kalman filter class, various ensemble filter variants have been applied to tracking trajectories in nonlinear systems. The most popular are the deterministic filters (Tippett et al., 2003; Sakov and Oke, 2008; Sakov et al., 2012).

**4.1 Ensemble Kalman filtering**

Ensemble Kalman filtering methods use Monte Carlo sampling to form the approximate error statistics of a model. An ensemble of model states $\boldsymbol{x}^f \in \mathbb{R}^N$ with a finite number of ensemble members $m$ produces an approximation to the true error covariance matrix as follows. The ensemble forecast anomaly matrix $\mathbf{X}^f \in \mathbb{R}^{N \times m}$ is constructed with respect to the ensemble mean $\overline{\boldsymbol{x}}^f \in \mathbb{R}^N$:

$$\overline{\boldsymbol{x}}^f = \frac{1}{m} \sum_{n=1}^{m} \boldsymbol{x}_n^f, \tag{10a}$$

$$\mathbf{X}^f \equiv \frac{1}{\sqrt{m-1}} \left[ \boldsymbol{x}_1^f - \overline{\boldsymbol{x}}^f, \dots, \boldsymbol{x}_m^f - \overline{\boldsymbol{x}}^f \right]. \tag{10b}$$

Note that we have dropped the time subscript $k$, the subscripts used here refer to individual ensemble members. The forecast error covariance matrices $\mathbf{P}^f$ are then constructed through

$$\mathbf{P}^f = (\mathbf{X}^f)(\mathbf{X}^f)^{\mathrm{T}}. \tag{11}$$

To preserve the variance of the ensemble through the analysis step, square-root (deterministic) schemes for ensemble Kalman filtering are often used. One such scheme is the ensemble transform Kalman filter (ETKF) developed by Bishop et al. (2001) then further adapted for large, spatiotemporally chaotic systems by Hunt et al. (2007). In such schemes, the observations do not need to be perturbed to preserve the analysis covariance in equation (9c). The main idea is that a transform matrix $\mathbf{T}$ can be used to adjust the ensemble analysis anomalies matrix,

$$\mathbf{X}^a \equiv \frac{1}{\sqrt{m-1}} \left[ \boldsymbol{x}_1^a - \overline{\boldsymbol{x}}^a, \dots, \boldsymbol{x}_m^a - \overline{\boldsymbol{x}}^a \right] = \mathbf{X}^f \mathbf{T}, \tag{12}$$

which ultimately forms the analysis error covariance matrix,

$$\mathbf{P}^a = (\mathbf{X}^a)(\mathbf{X}^a)^{\mathrm{T}}. \tag{13}$$

This transform matrix $\mathbf{T}$ is recovered through calculating the ensemble perturbations in normalized observation space,

$$\underline{\mathbf{E}}\underset{\sim}{\mathbf{S}} = (\mathbf{R}^{-1/2}\mathbf{H}\mathbf{X}^f)^{\mathrm{T}}(\mathbf{R}^{-1/2}\mathbf{H}\mathbf{X}^f), \tag{14}$$

The transform matrix $\mathbf{T}$ is then defined as

$$\mathbf{T} = (\mathbf{I} + \underline{\mathbf{E}}\underset{\sim}{\mathbf{S}}^{\mathrm{T}}\underline{\mathbf{E}}\underset{\sim}{\mathbf{S}})^{-1/2}, \tag{15}$$

5    where $\mathbf{I}$ is the $m \times m$ identity matrix. See Bishop et al. (2001) for the full derivation. This leads to the update of the ensemble mean and the individual ensemble members to their analyzed state through the equations:

$$\overline{\boldsymbol{x}}^a = \overline{\boldsymbol{x}}^f + \mathbf{K}(\boldsymbol{y} - \mathbf{H}\overline{\boldsymbol{x}}^f), \tag{16a}$$

$$\boldsymbol{x}_n^a = \overline{\boldsymbol{x}}^a + (\sqrt{m-1})[\mathbf{X}^f\mathbf{T}]_{*,n}. \tag{16b}$$

 Following Bishop et al. (2001) we define the Kalman gain $\mathbf{K}$  through equation (9a), and the

10   subscript $*, n$ denotes taking the $n$-th column of the matrix.

Another deterministic scheme for ensemble Kalman filtering which uses a left-multiplied transform matrix was shown by Tippett et al. (2003) to be equivalent to ETKF:

$$\mathbf{X}^a = \mathbf{T}\mathbf{X}^f, \tag{17a}$$

$$\mathbf{T} = (\mathbf{I} - \mathbf{K}\mathbf{H})^{1/2}. \tag{17b}$$

15   We will refer to this left-multiplied transform filter as the ensemble square-root filter (ESRF). The ensemble mean is updated through (16a) and the individual ensemble members are then updated through:

$$\boldsymbol{x}_n^a = \overline{\boldsymbol{x}}^a + (\sqrt{m-1})[\mathbf{T}\mathbf{X}^f]_{*,n}. \tag{18}$$

When using ensemble Kalman filtering methods like the ones introduced here, sampling errors can often occur. For nonlinear models in particular, there is a systematic underestimation of analysis error covariances which eventually leads to filter

20   divergence (Anderson and Anderson, 1999; Bocquet et al., 2015; Raanes et al., 2019). This is commonly avoided through the use of inflation. In other words, after each analysis step the ensemble anomalies are inflated through

$$\boldsymbol{x}_n^a = \overline{\boldsymbol{x}}^a + \lambda(\boldsymbol{x}_n^a - \overline{\boldsymbol{x}}^a), \qquad \lambda > 1, \tag{19}$$

where $(\lambda - 1)/100$ is the percentage inflation. Grudzien et al. (2018b) recently showed that the need for inflation tuning could

25   potentially be compensated by including the asymptotic stable modes which produce transient instabilities. The following section introduces one way to account for such transient instabilities through a projection of the forecast error covariance matrix onto a subset of CLVs.

**4.2 Ensemble filtering in reduced subspace**

Here we define the error covariance matrix $\mathbf{P}^f$ based on the directions of growth and decay of  errors associated with different timescales at the given analysis time. Specifically, we construct $\mathbf{P}^f$ using the CLVs computed at each data assimilation time step where the number of CLVs is determined by the local attractor dimension $\dim_{KY}$. This allows for the inclusion of unstable, neutral and stable directions dependent on the local dynamics of the system. This differs to past approaches where the subspace was determined in terms of the asymptotic Lyapunov exponents, and therefore the rank of the error covariance matrix was kept fixed (Trevisan and Uboldi, 2004; Carrassi et al., 2008; Trevisan and Palatella .

To determine the number of CLVs required to form the basis for $\mathbf{P}^f$, we use the time dependent or local $\dim_{KY}$ rounded up to an integer value. To determine how to weight the individual CLVs, we deconstruct the ensemble anomalies matrix defined in (10b), $\mathbf{X}^f$, into

$$\mathbf{X}^f = \mathbf{\Phi}\mathbf{W}, \tag{20}$$

where $\mathbf{\Phi}$ is a matrix with columns equal to the CLVs ($\phi_i$) and $\mathbf{W}$ is a matrix of weights. The columns of $\mathbf{\Phi}$ are ordered according to the corresponding FTLEs in descending order (the first being the direction corresponding to fastest unstable growth). In this formulation $\mathbf{\Phi}$ need not be square, i.e. the CLVs used do not need to span the entire space. We compute the CLVs at the assimilation step using Algorithm 1. Equation (20) can then be solved for $\mathbf{W}$ in a least squares sense through

$$\mathbf{W} = (\mathbf{\Phi}^{\mathrm{T}}\mathbf{\Phi})^{-1}\mathbf{\Phi}^{\mathrm{T}}\mathbf{X}^f. \tag{21}$$

The weights in $\mathbf{W}$ combined with the directions in $\mathbf{\Phi}$ now define an object with dimension equal to the chosen number of CLVs whose covariance matrix is defined by

$$\mathbf{P}^f = \mathbf{\Phi}\mathbf{W}\mathbf{W}^{\mathrm{T}}\mathbf{\Phi}^{\mathrm{T}}. \tag{22}$$

We can then use the formulation of $\mathbf{P}^f$ above in conjunction with the ensemble schemes of Section 4.1. We use the modified forecast covariance matrix (22) in the calculation of the Kalman gain (9a) which then also alters any subsequent calculations.

It is important that we span the local dimension of the attractor within the ensemble DA framework in order to avoid ensemble collapse. As mentioned in Section 3, the Kaplan-Yorke dimension computed from the FTLEs is an upper bound to the true local dimension. Figure 7 shows the alignment of the CLVs at two time instances of a model run where the leading FTLEs behave similarly and consequently the Kaplan-Yorke dimensions are approximately the same. In the first case ($t = 306.24$), the leading 5 CLVs are strongly aligned as well as the two most stable (8 and 9). The dimension based on alignment would then intuitively be around 4, and one would need to select the set of CLVs that are not aligned in order to avoid ensemble collapse. According to our method, since the local Kaplan-Yorke dimension is greater than 7 we would retain the leading 8 CLVs, therefore retaining all necessary directions to maintain spread. The second case ($t = 705.04$) shows very different alignment

[Figure]

**Figure 7.** We compare the alignment of the CLVs at two time steps of a model run. The two time steps have similar Kaplan-Yorke measure and distribution of leading FTLEs, shown in the bottom panels. We observe that the behaviour of the alignment can be vastly different for similar FTLE behaviour, however the method of retaining CLVs based on the Kaplan-Yorke measure gives a reliable way to reflect the true local dimension regardless of alignment.

behaviour. Here the leading CLVs are not strongly aligned, but there is strong alignment of CLVs 4-6 and pairs 3,7 and 8,9. This would give an alignment-based dimension around 5, but again we need to retain up to the 8th CLV to span the independent directions. While one could create a method based on alignment for selecting directions, we point out that the actual criteria for "strong alignment" is arbitrary and one could risk excluding a significant direction. For this reason we argue that the local Kaplan-Yorke measure gives a conservative estimate for the number of CLVs to retain.

**5 Results**

We perform a collection of data assimilation experiments for system (1) using a control run as observations (computed using a Runga-Kutta 4th order scheme with $\Delta t = 0.01$). Here we emphasize that we are interested in exploring the dynamical attributes of data assimilation across multiple timescales. In all cases we are using standard strong CDA, meaning that the cross-covariances are used amongst all components regardless of the observation set (Laloyaux et al., 2016; O'Kane et al., 2019). This allows for the analysis increment of any unobserved subsystems to be influenced by the observations, even in cases of weak coupling between the subsystems. We experiment with different observation sets to explore the applicability and

performance of the variable-rank strong CDA with incomplete or temporally correlated observations. Such observation sets are arguably more realistic representations of the types of observations used in climate applications.

The initialisation settings for the DA experiments are as follows, unless otherwise stated. We use the settings from Yoshida and Kalnay (2018):  assimilation window 0.08, inflation factor 1% (optimal for all experiments unless noted otherwise), and 10 ensemble members. We run the model for 75000 time steps (9375  assimilation windows) and use 50000 time steps (6250  assimilation windows) for calculating analysis error statistics. The model is let spin-up for 400 time steps before starting the assimilation cycles as we are using a window of $\tau = 4$ for the calculation of the FTLEs and CLVs. We note that for the CLV method, the system must be sufficiently tracking the control to accurately calculate the initial CLVs. For this reason we start the assimilation before there is significant ensemble divergence. The ensemble members are initialised as perturbations from the control initial condition, taken from a uniform distribution defined on $[-0.025, 0.025]$.

The dynamical properties of the experiments are calculated with respect to the ensemble mean trajectory. The FTLEs are computed using the QR decomposition over the previous 400 time steps leading up to the assimilation time step. The local Kaplan-Yorke dimension is then calculated from the FTLEs. The CLVs are then calculated using a slight modification to Algorithm 1- due to the absence of an accurate future trajectory of the ensemble mean, we do not perform the reorthogonalization to the eigenvectors of $\mathcal{A}(x^{i-n}, N)^* \mathcal{A}(x^{i-n}, N)$ (step 4). This is equivalent to using Algorithm 2.1 of Froyland et al. (2013). We use the previous 400 time steps of the ensemble mean trajectory leading up to the assimilation time step to compute the matrix cocycle and subsequently the CLVs.

**5.1 Constructing the observations**

[revised manuscript text omitted]

20 authors discuss the number of ensemble members which is equivalent to rank of covariance matrix). These fixed numbers are 5 and 6 CLVs, respectively. Finally we analyse our novel reduced subspace method which uses a variable number of CLVs based on the local Kaplan-Yorke dimension. We note that all experiments perform similarly when using the BLVs instead of the CLVs, however BLVs do not provide the same local phase-space information. While we focus on the performance of the CLV method in this work, we include a comparison using BLVs for the experiment utilising local dimension.

25 The error statistics of all the experiments are listed in Table 2. The analysis RMSE is calculated for each subsystem individually at every  assimilation step and then averaged over the  steps, in line with the error statistics produced in Yoshida and Kalnay (2018). We also calculate the average RMSE of the full system. The RMSE is defined as

$$\mathrm{RMSE} = \sqrt{\frac{1}{N}(\overline{\boldsymbol{x}}^a - \boldsymbol{x})^{\mathrm{T}}(\overline{\boldsymbol{x}}^a - \boldsymbol{x})}, \tag{23}$$

where $N$ is the number of states in the analysed system (either 3 or 9) and $\boldsymbol{x}$ is the truth (control run in our case).

$$\text{spread} = \sqrt{\sum_{n=1}^{m}(\boldsymbol{x}_n^f - \overline{\boldsymbol{x}}^f)^2}, \qquad\qquad \text{average increment} = \frac{1}{m}\sum_{n=1}^{m}(\boldsymbol{x}_n^a - \boldsymbol{x}_n^f),$$

$$\text{MAD} = \frac{1}{m}\sum_{n=1}^{m}|\boldsymbol{y} - \mathbf{H}\boldsymbol{x}_n^f|, \qquad\qquad \text{bias} = \frac{1}{m}\sum_{n=1}^{m}(\boldsymbol{y} - \mathbf{H}\boldsymbol{x}_n^f).$$

5   We also calculate the spread and average increment for each state variable through

$$\text{spread} = \sqrt{\text{diag}(\mathbf{P}^f)}, \qquad\qquad \text{average increment} = \overline{(\boldsymbol{x}^a - \boldsymbol{x}^f)}. \qquad\qquad (24)$$

Finally, we calculate the bias with respect to our observations $\boldsymbol{y}$,

$$\text{bias} = \overline{(\boldsymbol{y} - \mathbf{H}\boldsymbol{x}^f)}. \qquad\qquad (25)$$

[revised manuscript text omitted]
. In their study the authors find that the forecast error covariance is often underestimated in such highly nonlinear systems, particularly when the model is in a region of phase space subject to transitions. Subsequently, the Kalman gain is underestimated. The authors account for this by including the third and fourth moments of anomalies in the Kalman gain calculation.

Here we introduce a new method for adaptive scaling of the Kalman gain.  Rather than explicitly calculating higher moments of the anomalies, we account for the underestimation through a spread-dependent  factor which balances our forecast error covariance $\mathbf{P}^f$ and observation error variance $\mathbf{R}$ accordingly. The idea being that larger spread implies we have underestimated the covariances, and vice versa. We note that

this adaptive scaling is different to traditional inflation as it does not directly adjust the underlying ensemble spread. We use this in conjunction with the static background inflation of $1\%$ to avoid total ensemble collapse.

We scale the  Kalman gain in the following way:

$$\hat{\mathbf{K}} = \frac{||\mathbf{P}^f||\mathbf{P}^f\mathbf{H}^{\mathrm{T}}}{||\mathbf{P}^f||\mathbf{H}\mathbf{P}^f\mathbf{H}^{\mathrm{T}} + \mathbf{R}}, \tag{27}$$

or equivalently

$$\hat{\mathbf{K}} = \mathbf{P}^f\mathbf{H}^{\mathrm{T}}\left[\mathbf{H}\mathbf{P}^f\mathbf{H}^{\mathrm{T}} + \frac{\mathbf{R}}{||\mathbf{P}^f||}\right]^{-1}. \tag{28}$$

Here the scaling factor is the Frobenius norm of  $(\mathbf{X}^f)(\mathbf{X}^f)^{\mathrm{T}}$, where $\mathbf{X}^f$ is the ensemble spread matrix defined by (10b). This

$$\hat{\mathbf{P}}^f = ||(\mathbf{X}^f)(\mathbf{X}^f)^{\mathrm{T}}||(\mathbf{X}^f)(\mathbf{X}^f)^{\mathrm{T}},$$

$$\hat{\mathbf{P}}^f = ||(\mathbf{X}^f)(\mathbf{X}^f)^{\mathrm{T}}||(\mathbf{\Phi}\mathbf{W}\mathbf{W}^{\mathrm{T}}\mathbf{\Phi}^{\mathrm{T}}),$$

 rescaling factor is mathematically similar to the K-factor adaptive quality control procedure introduced by Sakov and Sandery (2017) and the $\beta$-factor rescaling of the background error covariances introduced by Bowler et al. (2013). The K-factor method was used to account for inconsistencies in observations and therefore uses an adaptive observation error covariance $\mathbf{R}$ that takes into account innovation size at each analysis step, while the $\beta$-factor is a  deflation to the forecast error covariance matrix to avoid the underestimation of the ensemble spread ($0 < \beta < 1$). Both the K-factor procedure and the $\beta$-factor multiplier can be shown to have the same scaling effect on the Kalman gain $\mathbf{K}$ defined by (9a) as the adaptive scaling presented here, with the difference in that the modified  $\hat{\mathbf{K}}$ in (28) takes into account both effects: small $||(\mathbf{X}^f)(\mathbf{X}^f)^{\mathrm{T}}||$ behaves like the $\beta$-factor and large $||(\mathbf{X}^f)(\mathbf{X}^f)^{\mathrm{T}}||$ behaves like the K-factor. We discuss the limiting behaviour of this adaptive scaling method in terms of the increment size and analysis error covariance in Appendix A.

Due to the fact that only the Kalman gain is being adjusted, for ease of implementation we use ESRF method introduced in Section 4.1. This allows for the left-transform matrix to be calculated with the modified Kalman gain,

$$\hat{\mathbf{K}} = \hat{\mathbf{P}}^f\mathbf{H}^{\mathrm{T}}[\mathbf{H}\hat{\mathbf{P}}^f\mathbf{H}^{\mathrm{T}} + \mathbf{R}]^{-1},$$

$$\mathbf{T} = (\mathbf{I} - \hat{\mathbf{K}}\mathbf{H})^{1/2}.$$

$$\mathbf{T} = (\mathbf{I} - \hat{\mathbf{K}}\mathbf{H})^{1/2}. \tag{29}$$

[Figure]

**Figure 12.** Trajectories of DA experiments using  variable CLVs, left-transform matrix (17b), and perfect observations from the extratropical subsystem of a control run ($x_e, y_e, z_e$), with (a-c) the standard Kalman gain (9a) and (d-f) the adaptive Kalman gain (28). Trajectories shown are control run (red), ensemble mean (blue), and individual ensemble members (multicoloured). For conciseness we only show the results for the x-coordinate of each subsystem. The other two coordinates behave similarly. Parameters:  assimilation window 0.02, inflation factor 1%, 10 ensemble members, observations error covariance $\mathbf{R} = I_4$.

~~We also note that the variable CLV method will be less effective than using the full rank CLV matrix (or equivalent standard ESRF). The reduction in dimension is related to the unobserved subsystems being unconstrained. By using the variable CLV method, we are *a priori* setting the rank of the ensemble member matrix and covariance matrix. This can also have the effect of reducing some cross-covariances further, and thus the scaling implemented here may be ineffective. Therefore, in both methods~~
5     We apply the adaptive gain to both the full rank (9 CLV) and variable rank formulation of the covariance matrix. The results of  the variable rank experiments with and without adaptive gain are shown in Figure 12 and the error statistics of all four experiments are listed in Table 6.

[revised manuscript text omitted]
  should be tested on additional systems with weak coupling in order to assess its general applicability, although care may need to be taken in the choice of the norm.

We now turn to the implications on more realistic high dimensional systems. It has been shown that when using a finer model resolution (increasing dimension) there is an increase in near-zero asymptotic Lyapunov exponents (De Cruz et al., 2018). We observed through the examination of the dynamical properties of the coupled Lorenz system that the stable yet near-zero exponents have the largest temporal variability which affect the local dimension. As the number of near-zero exponents increase, we may expect that the temporal variability in dimension will increase further. This would have strong implications on the necessary rank of the forecast error covariance and the subsequent number of ensemble members. It is not implausible that the number of ensemble members could vary significantly in time. In such a case where the model degrees of freedom

[revised manuscript text omitted]

$$\mathbf{P}^a = \left(\mathbf{I} - \mathbf{P}^f\mathbf{H}^{\mathrm{T}}(\mathbf{H}\mathbf{P}^f\mathbf{H}^{\mathrm{T}})^{-1}\mathbf{H}\right)\mathbf{P}^f. \tag{A9}$$

Recall that $\boldsymbol{y}$ are the observations defined by $\boldsymbol{y} = \mathbf{H}\boldsymbol{x}$ where $\boldsymbol{x}$ is the truth (the observation error has implicitly been set as zero in the case of large spread). For a function $\mathcal{F}$ with a Taylor series expansion operating on two arbitrary matrices $\mathbf{A}$ and $\mathbf{B}$, we have the following identity:

$$\mathcal{F}(\mathbf{A}\mathbf{B})\mathbf{A} = \mathbf{A}\mathcal{F}(\mathbf{B}\mathbf{A}) \tag{A10}$$

[revised manuscript text omitted]